# Decoding Causal Structure: End-to-End Mediation Pathways Inference

**Yulong Li[1*], Xiwei Liu[1,*], Feilong Tang[1,4,*], Ming Hu[1,4]**
**Jionglong Su[3], Zongyuan Ge[4], Imran Razzak[1,†], Eran Segal[1,2,†]**

[1]Mohamed bin Zayed University of Artificial Intelligence
[2] Weizmann Institute of Science
[3]Xi'an Jiaotong-Liverpool University
[4]Monash University
[*]Equal contribution
[†]Corresponding authors

## Abstract

Causal mediation analysis is crucial for deconstructing complex mechanisms of action. However, in current mediation analysis, complex structures derived from causal discovery lack direct interpretation of mediation pathways, while traditional mediation analysis and effect estimation are limited by the reliance on pre-specified pathways, leading to a disconnection between structure discovery and causal mechanism understanding. Therefore, a unified framework integrating structure discovery, pathway identification, and effect estimation systematically quantifies mediation pathways under structural uncertainty, enabling automated identification and inference of mediation pathways. To this end, we propose Structure-Informed Guided Mediation Analysis (**SIGMA**), which guides automated mediation pathway identification through probabilistic causal structure discovery and uncertainty quantification, enabling end-to-end propagation of structural uncertainty from structure learning to effect estimation. Specifically, SIGMA employs differentiable Flow-Structural Equation Models to learn structural posteriors, generating diverse Directed Acyclic Graphs (DAGs) to quantify structural uncertainty. Based on these DAGs, we introduce the Path Stability Score to evaluate the marginal probability of pathways, identifying high-confidence mediation paths. For identified mediation pathways, we integrate Efficient Influence Functions with Bayesian model averaging to fuse within-structure estimation uncertainty and between-structure effect variation, propagating uncertainty to the final effect estimates. In synthetic data experiments, SIGMA achieves state-of-the-art performance in pathway identification accuracy and effect quantification precision under structural uncertainty, concurrent multiple pathways, and nonlinear scenarios. In real-world applications using Human Phenotype Project data, SIGMA identifies mediation effects of sleep quality on cardiovascular health through inflammatory and metabolic pathways, uncovering previously unspecified multiple mediation paths.

## 1   Introduction

Causal Mediation Analysis (CMA) has emerged as a critical framework bridging causal inference and machine learning. Recent studies [47, 27] emphasize the evolution of CMA from identifying causal effects to elucidating underlying mechanisms, a transition essential for developing machine learning systems capable of intervention reasoning, cross-domain knowledge transfer, and interpretable explanations [10, 2, 52]. In particular, analyzing causal mediation pathways involving multiple mediators arranged in parallel, sequential, or networked structures [58, 9, 28] provides deeper mechanistic

39th Conference on Neural Information Processing Systems (NeurIPS 2025).

insights in high-dimensional contexts commonly encountered in biology, social networks, and behavioral sciences [21, 26, 40]. For instance, elucidating how sleep quality impacts cardiovascular health in Human Phenotype Projects (HPP) [36] requires modeling its interplay with multiple mediators (*e.g.,* inflammation, metabolic) across interrelated pathways [30, 61, 7]. Therefore, quantifying the effects of mediation pathways under causal structural uncertainty remains a challenge.

Although analyzing complex mediation pathways is both theoretically and practically significant, current methods face several limitations: *(i) Structure Discovery (SD):* While mainstream causal discovery algorithms [56, 55, 72] can automatically learn causal structures, they lack principled mechanisms for quantifying and propagating structural uncertainty, undermining the reliability of downstream analyses. *(ii) Pathway Identification (PI):* Current structure discovery methods [1, 29] fail to explicitly identify specific mediation pathways. Meanwhile, classical mediation analysis approaches [3, 27] rely on pre-specified paths, which become infeasible in high-dimensional data due to the exponential growth in the number of candidate paths. *(iii) Effect Estimation (EE):* Semi-parametric methods like Efficient Influence Function (EIF) [6] offer theoretical guarantees (*e.g.,* multiple robustness) [59, 71, 41] but typically assume that the mediation structure is known and correctly specified, which creates a contradiction with the structural uncertainty faced in the previous two stages. Consequently, a gap between current methods becomes evident in high-dimensional settings like HPP, where discovery yields causal structure without pathway interpretation, while traditional mediation is confined to predefined paths, missing underlying causal mechanisms. *In this work, we argue that a unified framework integrating SD, PI, and EE can quantify the mediating pathways under structural uncertainty in real-world settings.*

To this end, we propose an end-to-end **S**tructure-**I**nformed **G**uided **M**ediation **A**nalysis (SIGMA) framework, which integrates probabilistic causal structure discovery with uncertainty quantification to guide automatic mediation pathway identification, and propagates structural uncertainty to robust effect estimation, thus enabling integrated inference of causal structures and mediation effects. Specifically, differentiable Flow-Structural Equation Models [64] are leveraged to learn edge parameters encoding structural uncertainty, from which an ensemble of directed acyclic graphs (DAGs) is subsequently sampled. Building on this DAG ensemble, we introduce the Path Stability Score which performs Bayesian inference by evaluating the marginal probability paths across DAGs to select high-confidence mediation pathways, thereby circumventing manual path pre-specification. For each identified mediation pathway, SIGMA employs Efficient Influence Functions to estimate pathway-specific mediation effects, using nuisance functions tailored to each DAG structure. Subsequently, these pathway-specific estimates are aggregated across the DAG ensemble via Bayesian model averaging (BMA) [48], propagating structural uncertainty into the final causal effect estimates. Furthermore, SIGMA integrates a variational autoencoder-based imputation module to explicitly-model latent dependencies among variables, enabling accurate imputation of missing values. We empirically demonstrate SIGMA's effectiveness on high-dimensional, heterogeneous HPP data[1], identifying and quantifying mediation pathways connecting sleep quality to cardiovascular health through inflammatory and metabolic mediators. The contributions are summarized as follows:

- We propose an end-to-end framework, SIGMA, unifying probabilistic structure discovery with uncertainty quantification, high confidence path identification, and effect estimation incorporating structural uncertainty, to address mediation analysis under uncertainty structures.

- We introduce the Path Stability Score to identify mediation by quantifying their stability across posterior directed acyclic graph samples, thus eliminating manual pre-specification.

- We develop a structure-guided estimation approach, employing Efficient Influence Functions (EIF), combined with Bayesian model averaging, enabling propagation of structural uncertainty and inference by leveraging EIF properties.

- Extensive experiments validate SIGMA on real-world HPP data, revealing complex and previously unspecified mediation pathways linking sleep quality to cardiovascular health through inflammatory and metabolic, demonstrating its interpretability in real-world.

## 2  Dataset

**Human Phenotype Project (HPP)**[1]**.** We conduct our real-world evaluation using the HPP dataset, a deeply phenotyped cohort comprising over 6,000 individuals with multi-night home sleep apnea

---

[1]`https://knowledgebase.pheno.ai/`

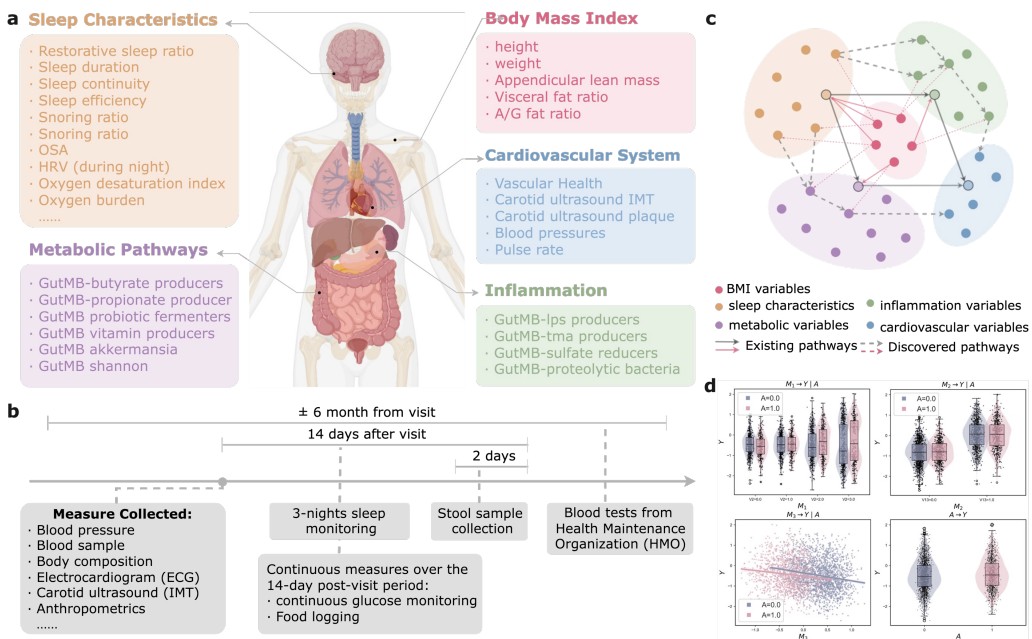

Figure 1: **(a)** Illustration of five body system-level categories from the HPP used in this work. **(b)** Timeline of data acquisition in the HPP cohort. **(c)** Conceptual schematic of the SIGMA framework, showing its capacity to identify both known and previously unrecognized mediation pathways under structural uncertainty. **(d)** Distributions and dependencies among variables in a representative synthetic dataset (MidDim-S), illustrating the indirect effects of mediators $M_1, M_2, M_3$ on outcome $Y$ under different treatment $A$ and the direct effect of $A$ on $Y$. Refer Appendix E.3 for details.

testing (HSAT) data across 16,000+ nights. Beyond sleep, HPP includes multimodal data covering 16 other body systems that represent a wide array of physiological functions and environmental exposures [34]. This coordinated collection process, as shown in Figure 1(b), provides temporally aligned, longitudinal measurements across multiple physiological systems. The integration of sleep monitoring, clinical biomarkers, imaging, and omics data within a ±6-month window requires considerable logistical effort and sustained participant engagement. These characteristics make the HPP dataset a particularly valuable testbed for evaluating SIGMA's ability to recover complex mediation pathways under structural uncertainty. In total, we utilize data from 11 body systems, covering over 500 raw clinical and physiological attributes. To support tractable and clinically meaningful mediation analysis, we aggregated these into 62 domain-informed features by grouping related variables across systems. Each feature was designed to reflect a coherent physiological process suitable for pathway-level causal interpretation. Among these, five representative categories analyzed in our work: **Sleep Characteristics**, **Body Mass Index (BMI)**, **Cardiovascular System**, **Inflammation** and **Metabolic Pathways**, are illustrated in Figure 1(a). Detailed data descriptions can be found in the Appendix D. As depicted in Figure 1(c), SIGMA framework is capable of validating previously reported mediation pathways [19] while also uncovering novel, clinically meaningful causal mediation mechanisms spanning different systems. We provide results to demonstrate this in Section 5.3, where we quantify these pathways and assess their natural effect.

**Synthetic Data.** To evaluate SIGMA's ability to identify mediation pathways under controlled yet realistic conditions, we generate synthetic datasets that mirror key statistical and structural characteristics of real-world data (Figure 1(d)). Specifically, we simulate missing data patterns with both missing completely at random (MCAR) [50] and missing at random (MAR) mechanisms to reflect non-random measurement gaps common in clinical studies, and introduce heterogeneous variable types (continuous, binary, categorical) and nonlinear relationships to mimic biological complexity. Each synthetic dataset is constructed from a randomly sampled DAG using an **Erdős-Rényi (ER)** [20] model, followed by structural equation modeling (SEM) [46] with configurable nonlinearity and noise. We explicitly embed mediation structures—parallel, chain, or hybrid—between a designated treatment and outcome, with tunable effect strengths to ensure identifiable direct and indirect effects.

Functional mechanisms include both linear and nonlinear mappings (e.g., polynomial, sinusoidal), and exogenous noise is drawn from mixed distributions to induce non-Gaussianity. Natural Direct/Indirect Effects (NDE/NIE) [45, 49] are estimated via Monte Carlo simulations to facilitate quantitative benchmarking. This synthetic setup enables systematic evaluation of SIGMA's robustness across varied structural regimes, pathway configurations, and data quality levels. See Appendix E.1 for detailed generation procedures and E.2 for embedded mediation pathway structures.

## 3 Causal Mediation: Definitions, Notation, and Identification

We consider the problem of causal mediation analysis in observational data. Let $A$ denote the treatment variable, $Y$ the outcome variable, $M = \{M_1, \ldots, M_k\}$ a set of $k$ potential mediators, and $C$ a set of covariates. The variables may be continuous, discrete, or mixed-type. Let $V = \{A, Y\} \cup M \cup C$ be the full set of observed variables and assume the underlying causal structure is a DAG $G = (V, E)$, where $\mathrm{Pa}_G(V_i)$ denotes the parent set of node $V_i$. We assume that the true graph $G^*$ satisfies the Causal Markov and Faithfulness assumptions, which enable conditional independencies to be inferred from the graph. Appendix A.1 provides a detailed discussion of these assumptions, including Causal Sufficiency.

We adopt the Potential Outcomes Framework [50, 57] to define causal mediation effects. Let $M(a)$ be the potential mediator value under treatment $A = a$, and $Y(a, m)$ the potential outcome if $A = a$ and $M = m$. We focus on two core mediation effects comparing treatment level $a$ to reference level $a^*$: (1) The **NDE** [45] is defined as $\mathrm{NDE}(a, a^*) = \mathbb{E}[Y(a, M(a^*)) - Y(a^*, M(a^*))]$, quantifying the effect of changing $A$ from $a^*$ to $a$ on $Y$ while holding $M$ fixed at its natural level under $A = a^*$. (2) The **NIE** [49] is defined as $\mathrm{NIE}(a, a^*) = \mathbb{E}[Y(a, M(a)) - Y(a, M(a^*))]$, capturing the effect on $Y$ mediated through $M$ when $A$ change from $a^*$ to $a$. When the true causal structure is unknown, our goal is to estimate these effects under structural uncertainty. Under certain assumptions [45], the Total Effect, $\mathrm{TE}(a, a^*) = \mathbb{E}[Y(a) - Y(a^*)]$ can be decomposed as $\mathrm{TE} = \mathrm{NDE} + \mathrm{NIE}$.

Identifying NDE and NIE from observational data requires several key assumptions [8, 45]. We use $\perp\!\!\!\perp$ to denote conditional independence. Appendix A.3 discuss more about following assumptions, including strategies for practical consideration and the theoretical implications of violation. Throughout, *identification* denotes recovery of NDE and NIE under ignorability, consistency, and positivity; *path identification* denotes that a path attains high posterior support as quantified by the PSS. Estimation then proceeds via influence-function–based and plug-in estimators under these assumptions.

**Consistency**: if $A = a'$, then $M = M(a')$, and if $A = a'$ and $M = m$, then $Y = Y(a', m)$, for all relevant treatment levels $a'$ (e.g., $a' \in \{a, a^*\}$) and all $m$ in the support of $M$.

**Ignorability**: The following conditional independence relations hold almost surely for all relevant values $a, a^*, m, m'$: (i) $A \perp\!\!\!\perp \{Y(a^*, m), M(a)\} \mid C$; (ii) $M \perp\!\!\!\perp Y(a, m') \mid A = a, C$; and (iii) $Y(a^*, m) \perp\!\!\!\perp M(a) \mid C$.

**Positivity**: For all $a$ and $c$ in the support of $A$ and $C$, the treatment propensity score $\pi(a \mid C) \equiv \Pr(A = a \mid C = c) \in (c_1, c_2)$ for some constants $0 < c_1 \leq c_2 < 1$. The conditional density function for discrete $M = m$ given $A$ and $C = c$ is bounded between $[\rho_1, \rho_2]$ for some constants $0 < \rho_1 \leq \rho_2 < \infty$, almost surely for all relevant $a, c$, and $m$.

These assumptions collectively allow identification of NDE and NIE from observational data, forming the foundation of our mediation analysis under unknown causal graph structure.

## 4 Methodology

### 4.1 SIGMA Framework Overview

We propose Structure-Informed Guided Mediation Analysis (SIGMA), a unified framework for causal mediation analysis under structural uncertainty. SIGMA operates in an end-to-end fashion, integrating causal discovery, pathway identification, and effect estimation within a probabilistic graphical modeling paradigm. Given observational data over treatment, outcome, mediators, and confounders, SIGMA performs probabilistic structure discovery via a flow-based structural equation model (Flow-SEM), generating a posterior ensemble of DAGs that capture causal uncertainty (Section 4.2). It then identifies stable mediation pathways using a path stability score (PSS), which quantifies the marginal support for each path across sampled DAGs (Section 4.3). For each retained pathway, SIGMA estimates natural direct and indirect effects using structure-guided nuisance estimation and

a hybrid influence-function/plugin strategy, with final estimates aggregated via Bayesian model averaging to propagate structural uncertainty (Section 4.4). This unified design enables automated, uncertainty-aware mediation analysis in complex high-dimensional systems.

## 4.2 Probabilistic Structure Discovery

To model structural uncertainty in causal mediation, SIGMA employs a Flow-SEM to learn a distribution over DAGs from observational data. Let $V = \{V_1, \ldots, V_p\}$ denote the set of variables, the goal is to infer a posterior distribution $P(G \mid D)$ over DAGs $G = (V, E)$ that encode the conditional dependencies among these variables. The validity of this structure discovery step relies on several core assumptions, including the Causal Markov, Faithfulness, and Causal Sufficiency assumptions. Refer to Appendix A for a detailed discussion.

### 4.2.1 Flow-SEM Model Formulation

In Flow-SEM, each variable $V_i \in V$ is modeled as a stochastic function of its parents in a candidate DAG $G$, with the structural equation:

$$V_i = f_i(\mathrm{Pa}_G(V_i); \phi_i) + \sigma_i(\mathrm{Pa}_G(V_i); \phi_i) \cdot U_i, \quad U_i \sim \mathcal{N}(0, 1), \tag{1}$$

where $\mathrm{Pa}_G(V_i)$ denotes the parent set of $V_i$, and $f_i, \sigma_i$ are flexible neural network functions parameterized by $\phi_i$. The collection $\Phi = \{\phi_i\}_{i=1}^p$ defines the global functional parameters. This formulation accommodates nonlinearities and heteroskedasticity in the data-generating process.

To enable differentiable structure learning, SIGMA introduces a continuous weight matrix $\mathbf{W}_\theta \in \mathbb{R}^{p \times p}$, where each entry $[\mathbf{W}_\theta]_{ji}$ reflects the strength or existence of a directed edge from $V_j$ to $V_i$. A zero-diagonal $[\mathbf{W}_\theta]_{ii} = 0$ is imposed to prevent self-loops. The parent set $\mathrm{Pa}_G(V_i)$ is inferred from the sparsity pattern of $\mathbf{W}_\theta$. Under this, the joint distribution $P(V | \mathbf{W}_\theta, \Phi)$ can be derived via a change-of-variables formulation using normalizing flows. Given an observed data sample $V^{(j)}$, its log-likelihood is computed as:

$$\log P(V^{(j)} | \mathbf{W}_\theta, \Phi) = \sum_{i=1}^p \log p_U(u_i^{(j)}) - \sum_{i=1}^p \log \sigma_i(\mathrm{Pa}_G(V_i^{(j)}); \phi_i) \tag{2}$$

where $u_i^{(j)} = (V_i^{(j)} - f_i(\mathrm{Pa}_G(V_i^{(j)}); \phi_i))/\sigma_i(\mathrm{Pa}_G(V_i^{(j)}); \phi_i)$ represents the standardized noise for variable $V_i^{(j)}$. This formulation enables likelihood-based training of both structural and functional components via stochastic optimization.

### 4.2.2 Model Learning and Optimization

Model parameters $(\mathbf{W}_\theta, \Phi)$ are optimized by minimizing a regularized loss that balances data fit, structural sparsity, and acyclicity:

$$\mathcal{L}(\mathbf{W}_\theta, \Phi; D) = -\frac{1}{n} \sum_{j=1}^n \log P(V^{(j)} | \mathbf{W}_\theta, \Phi) + \lambda_{dag} h(\mathbf{W}_\theta) + \alpha \|\mathbf{W}_\theta\|_1 \tag{3}$$

where $h(\mathbf{W}_\theta) = \mathrm{tr}(\exp(\mathbf{W}_\theta \odot \mathbf{W}_\theta)) - p$ serves as a continuous relaxation for enforcing acyclicity, $\exp(\cdot)$ is the matrix exponential and $\odot$ denotes element-wise multiplication; $\|\mathbf{W}_\theta\|_1 = \sum_{i \neq j} |[\mathbf{W}_\theta]_{ji}|$ encourages sparsity. Further technical details and acyclicity properties are discussed in Appendix A. Gradient-based optimization is used to train the model. Once trained, $\mathbf{W}_\theta$ captures both the functional and structural dependencies among variables.

### 4.2.3 Approximate Structural Posterior and DAG Sampling

To approximate the structural posterior $P(G \mid D)$, SIGMA interprets each entry in $\boldsymbol{W}_\theta$ as defining a marginal edge inclusion probability via:

$$p_{ij} = \sigma(|[\boldsymbol{W}_\theta]_{ji}|), \quad \text{with } p_{ii} = 0, \tag{4}$$

where $\sigma(\cdot)$ is the sigmoid function. Candidate DAGs $\{G_s\}_{s=1}^{N_{\mathrm{DAG}}}$ are generated by sampling each edge independently as Bernoulli($p_{ij}$), followed by cycle removal to ensure graph validity.

This ensemble of sampled DAGs provides a Monte Carlo approximation of the posterior over causal structures, capturing both edge-level and pathway-level uncertainty. These samples are used downstream for pathway selection and mediation effect estimation (see Sections 4.3 and 4.4).

### 4.3 Automated Mediation Pathway Identification

Given a posterior ensemble of sampled DAGs $\{G_s = (V, E_s)\}_{s=1}^{N_{\text{DAG}}}$, SIGMA identifies mediation pathways that are robust to structural uncertainty by quantifying the posterior support for each candidate path. This is achieved through the **Path Stability Score** (PSS), which estimates the marginal probability of a directed path being present in the true causal structure.

Formally, let $\pi = (V_0 \to V_1 \to \cdots \to V_k)$ denote a directed acyclic path from treatment node $V_0 = A$ to outcome node $V_k = Y$. The PSS of path $\pi$ is defined as:

$$\text{PSS}(\pi) = \frac{1}{N_{\text{valid}}} \sum_{s=1}^{N_{\text{DAG}}} \mathbb{I}(\pi \subseteq G_s), \tag{5}$$

where $\mathbb{I}(\pi \subseteq G_s) = 1$ if all directed edges in $\pi$ are present in graph $G_s$, and 0 otherwise. The term $N_{\text{valid}}$ denotes the number of acyclic graphs retained after cycle removal. As shown in Appendix B.1, PSS is an unbiased and consistent estimator of the posterior inclusion probability $P(\pi \subseteq G \mid D)$.

To construct the candidate set of mediation paths, SIGMA enumerates all simple directed paths from $A$ to $Y$ in each valid DAG $G_s$, subject to a maximum length constraint for computational tractability. These paths are aggregated across DAGs, and their frequencies are normalized to obtain the empirical PSS scores. Paths with stability above a user-specified threshold $\tau \in [0, 1]$ are retained as high-confidence mediation candidates, forming the set: $\Pi_{\text{stable}} = \{\pi \mid \text{PSS}(\pi) \geq \tau\}$. This selection process effectively propagates structural uncertainty from edge-level variation to the pathway level, allowing downstream effect estimation to focus on statistically supported mediation mechanisms. A theoretical justification for using PSS as a Monte Carlo estimator and its asymptotic properties is provided in Appendix B.1 and B.2.

### 4.4 Mediation Effect Estimation

Once a set of stable mediation pathways $\Pi_{\text{stable}}$ is identified from the DAG ensemble, SIGMA proceeds to estimate path-specific NDE and NIE for each retained pathway $\pi \in \Pi_{\text{stable}}$. Given that each pathway is supported by a subset of DAGs $\{G_s \mid \pi \subseteq G_s\}$, effect estimation must account for both within-structure statistical uncertainty and between-structure causal ambiguity. SIGMA achieves this by combining structure-specific nuisance estimation with hybrid estimators and Bayesian model averaging (BMA). Appendix A.3 details the assumptions required for valid estimation, including ignorability, consistency, and positivity conditions.

#### 4.4.1 Structure-Guided Nuisance Estimation

For each stable pathway $\pi \in \Pi_{\text{stable}}$, SIGMA estimates the nuisance functions $\hat{\eta}_s$ by leveraging structure-specific information from each DAG $G_s$ in the ensemble. The conditioning sets for all components of $\hat{\eta}_s$ are defined strictly by the parent sets $\text{Pa}_{G_s}(\cdot)$ within the corresponding DAG. Neural networks are used to fit the conditional relationships implied by these parent sets. To reduce estimation bias and enhance robustness, we use K-fold cross-fitting to compute out-of-sample predictions across the dataset. This procedure yields all necessary components for downstream effect estimation via both the EIF and plugin estimators, including the required counterfactual outcome predictions.

#### 4.4.2 DAG Effect Estimation

After obtaining nuisance function estimates $\hat{\eta}_s$ for a specific DAG $G_s$ (as detailed in Section 4.4.1), the SIGMA framework calculates path-specific NDE and NIE for a path $\pi$. These constitute the single-DAG effect vector $\hat{\theta}_s(\pi) = \left( \widehat{\text{NDE}}_s(\pi), \widehat{\text{NIE}}_s(\pi) \right)$. SIGMA employs a hybrid estimation strategy: for paths of length $L(\pi)$ (number of nodes) equal to 3 (*i.e.,* $A \to M \to Y$), an influence function based estimation method is utilized; for paths with $L(\pi) > 3$, a plugin estimator is adopted. All effect quantifications are based on a comparison between a pre-specified exposure level $a$ and a reference level $a^*$. For a binary exposure $A$, these levels are 1 and 0, respectively. For a continuous exposure $A$, $a^*$ is its estimated mean $\tilde{\mu}_A$ and $a$ is $\tilde{\mu}_A + k \cdot \tilde{\sigma}_A$ (with $k = 1$ in this study), where $\tilde{\mu}_A$ and $\tilde{\sigma}_A$ are estimated from the first K-Fold training set for the specific (path, DAG) combination.

**Influence Function-based Estimation** ($L(\pi) = 3$): The point estimate $\hat{\theta}_s(\pi)$ is the sample mean of the corresponding influence function scores $\psi_i(Z_i; \hat{\eta}_s, G_s)$, i.e., $\hat{\theta}_s(\pi) = \frac{1}{n} \sum_{i=1}^{n} \psi_i(Z_i; \hat{\eta}_s, G_s)$.

For a **binary exposure** $A$, the influence functions $\psi_{\text{NDE}}(Z_i; \hat{\eta}_s)$ and $\psi_{\text{NIE}}(Z_i; \hat{\eta}_s)$ are constructed within the SIGMA framework by combining cross-fitted estimates of several nuisance functions. These include the propensity score $\hat{P}(A_i = a | C_{A,i})$, outputs from the mediator model $\mathcal{M}_{M|A,C_M}$, which characterizes the conditional behavior of $M_i$ given $A_i = a$ and covariates $C_{M,i}$, and the conditional expectation of the outcome $\hat{E}[Y_i | A_i = a, M_i, C_{Y,i}]$. These IFs are formulated to provide efficient, bias-corrected estimates for the target counterfactual expectations (e.g., $\mathbb{E}[Y(a', M(a''))]$) that comprise NDE and NIE. The specific mathematical expressions for these influence functions as utilized within SIGMA are detailed in Appendix A.3.4.

For a **continuous exposure** $A$, SIGMA constructs and applies a specific influence function. This function aims to correct plugin estimates of counterfactual expectations (e.g., $\gamma_{aa^*} = \mathbb{E}[Y(a, M(a^*))]$), which form the NDE and NIE, by integrating cross-fitted nuisance components using the data-driven intervention levels $(a, a^*)$. These key nuisance components include: (i) the conditional expectation of $A$, $\hat{\mu}_A(C_{A,i})$; (ii) an estimate of the global homoscedastic residual variance of $A$, $\hat{\sigma}_{A|C_A,res}^2$; (iii) the conditional expectation of the outcome $\hat{\mu}_{Y|A=\text{level},M_{obs},C_Y}^{(i)}$; and (iv) individual plugin predictions of the target counterfactual expectation $\hat{\gamma}_{aa^*,i}$ and their sample mean $\hat{\gamma}_{aa^*}$. The influence function score $\psi_{\gamma_{aa^*}}(Z_i)$ used in SIGMA to estimate a counterfactual expectation (such as $\gamma_{aa^*}$) is defined as:

$$\psi_{\gamma_{aa^*}}(Z_i; \hat{\eta}_s) = \frac{A_i - \hat{\mu}_A(C_{A,i})}{\hat{\sigma}_{A|C_A,res}^2} \left( \hat{\mu}_{Y|A=a,M_{obs},C_Y}^{(i)} - \hat{\gamma}_{aa^*} \right) + \left( \hat{\gamma}_{aa^*,i} - \hat{\gamma}_{aa^*} \right), \qquad (6)$$

The overall influence functions for NDE and NIE are derived from linear combinations of these component scores. The theoretical motivation and property analysis of this SIGMA-specific influence function are detailed in Appendix A.3.5. For all influence function-based estimates, the single-DAG variance $\hat{V}(\hat{\theta}_s(\pi))$ is estimated by $\frac{1}{n^2} \sum_{i=1}^{n} (\psi_i - \bar{\psi})^2$, where $\bar{\psi} = \hat{\theta}_s(\pi)$.

**Plugin Estimation ($L(\pi) > 3$):** For paths longer than three nodes, SIGMA utilizes a plugin estimator. NDE and NIE are computed by substituting the relevant cross-fitted counterfactual outcomes (i.e., individualized $\hat{\mu}_Y^{(i)}(a, M(a'))$ predictions from Section 4.4.1) into their defining expressions, followed by averaging over the sample. Specifically, $\widehat{\text{NIE}}_s(\pi) = \frac{1}{n} \sum_{i=1}^{n} \left( \hat{\mu}_Y^{(i)}(a, M(a)) - \hat{\mu}_Y^{(i)}(a, M(a^*)) \right)$. Its variance $\hat{V}(\hat{\theta}_s(\pi))$ is calculated as $\frac{1}{n} \left( \frac{1}{n-1} \sum_{i=1}^{n} (\Delta_{\text{CF}}^{(i)} - \overline{\Delta_{\text{CF}}})^2 \right)$, where $\Delta_{\text{CF}}^{(i)}$ denotes the individual-level counterfactual difference.

### 4.4.3 Aggregation and Uncertainty Propagation

To generate final mediation effect estimates that account for structural uncertainty, SIGMA aggregates single-DAG estimates $\hat{\theta}_s(\pi)$ and their variances $\hat{V}(\hat{\theta}_s(\pi))$ for a path $\pi$ across the $N_{\text{path}}$ supporting DAGs. The Bayesian Model Averaged point estimate is $\hat{\theta}_{\text{BMA}}(\pi) = \frac{1}{N_{\text{path}}} \sum_{s=1}^{N_{\text{path}}} \hat{\theta}_s(\pi)$, using uniform weights for each supporting DAG $G_s$. The total BMA variance, $\hat{V}(\hat{\theta}_{\text{BMA}}(\pi))$, combines the average within-DAG variance (reflecting estimation uncertainty for each structure) and the between-DAG variance (capturing structural uncertainty across different structures) via the law of total variance:

$$\hat{V}(\hat{\theta}_{\text{BMA}}(\pi)) = \frac{1}{N_{\text{path}}} \sum_{s=1}^{N_{\text{path}}} \hat{V}(\hat{\theta}_s(\pi)) + \frac{1}{N_{\text{path}}} \sum_{s=1}^{N_{\text{path}}} \left( \hat{\theta}_s(\pi) - \hat{\theta}_{\text{BMA}}(\pi) \right)^2. \qquad (7)$$

Here, $\hat{V}(\hat{\theta}_s(\pi))$ is the single-DAG variance estimate from Section 4.4.2. Confidence intervals are then constructed using the standard normal approximation: $\hat{\theta}_{\text{BMA}}(\pi) \pm z_{1-\alpha/2} \sqrt{\hat{V}(\hat{\theta}_{\text{BMA}}(\pi))}$. This aggregation framework ensures that both estimation uncertainty (within-DAG) and structural uncertainty (between-DAG) are rigorously propagated into the final mediation effect estimates.

## 5 Experiments

We evaluate SIGMA across two key tasks: (1) causal structure discovery and (2) causal mediation analysis, covering both synthetic datasets with known ground-truth effects and a real-world medical cohort from the HPP. For structure discovery, we assess the accuracy of learned causal graphs in

various settings with different graph topologies, nonlinearity levels, and data types. For mediation analysis, we test SIGMA's ability to estimate direct and indirect effects of identified mediation pathways. The synthetic data allows controlled benchmarking, while the HPP cohort provides a complex, high-dimensional setting with clinically relevant mediation structures. A full list of results and details of the experimental setup are in Appendix F.1.

**Experimental setup** Our experimental evaluation comprises two primary tasks: (1) performance assessment of causal structure discovery, and (2) end-to-end causal mediation analysis evaluation. For synthetic data experiments, we construct 10 benchmark datasets that systematically vary feature dimensionality, degree of nonlinearity, and pathway structural complexity to assess SIGMA's robustness across different complexity regimes. Each dataset contains 6,000 samples with mixed variable types (60% continuous, 30% binary, 10% categorical) and 10% injected missingness to simulate realistic scenarios.

The SIGMA framework is implemented through a three-stage pipeline: (1) learning the structural posterior via Flow-SEM and sampling 1,000 DAGs; (2) identifying high-confidence mediation pathways based on Path Stability Score (threshold $\tau = 0.15$); (3) estimating natural direct and indirect effects for identified pathways, with estimates aggregated across multiple DAGs via Bayesian model averaging to propagate structural uncertainty. Effect estimation employs a 5-fold cross-fitting strategy, utilizing EIF for paths of length 3 and plug-in estimators for longer paths.

## 5.1 Causal Discovery Evaluation

**Datasets.** We generate 10 synthetic datasets with varying characteristics, including different graph sizes from low to high dimensional $p \in \{20, 50, 100, 200\}$, nonlinearity level ($\rho_{nonlin} \in \{0, 0.5\}$), graph structures (parallel $\kappa \in \{2, 3, 6\}$ and chain $\ell \in \{1, 2\}$), and variable types variables (continuous, binary, categorical). Dataset-specific configurations are provided in Appendix E.3.

Figure 2 shows that SIGMA demonstrates consistently strong performance across datasets. On datasets (HighDim-D/P), DECI has a slight advantage in adjacency F1 (0.3814/0.4869 vs. 0.3741/0.4754), potentially due to differences between SIGMA's Flow-SEM continuous optimization framework in capturing weak connections under high-dimensional sparse conditions and DECI's global variational inference approach. In contrast, SIGMA shows comprehensive leadership in orientation determination (average improvement of 42.1%), a capability crucial for accurate identification of mediation pathways. Experimental results confirm that SIGMA can recover directed structures under diverse conditions, providing a robust foundation for downstream causal effect estimation.

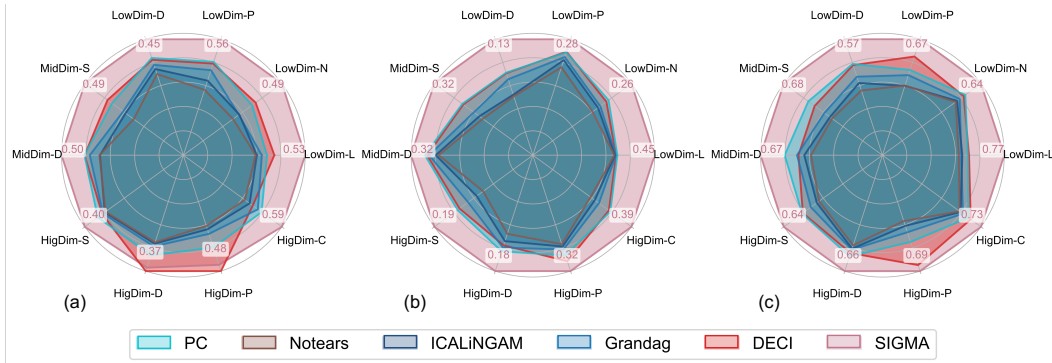

Figure 2: **Causal discovery on benchmark datasets.** We compare SIGMA against PC [33], (linear) Notears [72], ICALiNGAM [55], Grandag [20] and DECI [22] on ten synthetic Datasets. (a) shows adjacency F1 [23], (b) reports orientation F1 [60], and (c) evaluates causal accuracy [15]. Refer Table 5 in Appendix F for detailed results.

## 5.2 End-to-end Causal Mediation Analysis

We evaluate SIGMA's effect estimation performance on synthetic data by comparing estimates against ground truth values generated via structural equation models (see Appendix F.2). For comparison, we

use the Lasso, random forest (RF), gradient boosted machine (GBM) and DeepMed [65]. Two high-confidence mediation pathways were selected from the synthetic dataset *LowDim-L* for evaluation. As shown in Table 1, SIGMA consistently produces low bias and narrow confidence intervals across both natural direct and indirect effects. These results indicate that by sampling multiple DAGs from the structural posterior and applying BMA, SIGMA effectively incorporates both within-graph estimation uncertainty and across-graph structural variation. This leads to robust and stable inference of mediation effects under structural uncertainty.

Table 1: The empirical biases, empirical standard errors (SE) and the lower and upper bounds of the corresponding 95% confidence intervals (CI) of the estimated $\widehat{\text{NDE}}$, $\widehat{\text{NIE}}$ and $\widehat{\text{TE}}$ across two mediation pathways on synthetic dataset *LowDim-L*. (Pathway 1: V5→V19→V16; Pathway 2: V12→V10→V6). Bias is the Monte-Carlo mean deviation from the ground truth.

| | | $\widehat{\text{NDE}}$ | | | | | $\widehat{\text{NIE}}$ | | | | | $\widehat{\text{TE}}$ | | | | |
| | Method | DeepMed | Lasso | RF | GBM | SIGMA | DeepMed | Lasso | RF | GBM | SIGMA | DeepMed | Lasso | RF | GBM | SIGMA |
|---|---|---|---|---|---|---|---|---|---|---|---|---|---|---|---|---|
| Pathway 1 | Empirical Bias | -0.0637 | -0.2098 | -0.2263 | 0.0713 | -0.0066 | 0.0777 | 0.0339 | 0.4347 | -0.2074 | 0.0811 | 0.1414 | 0.2084 | -0.1360 | -0.1361 | 0.0746 |
| | SE | 0.0497 | 0.0139 | 0.0082 | 0.0058 | 0.0063 | 0.0575 | 0.0151 | 0.0095 | 0.0152 | 0.0078 | 0.1008 | 0.0207 | 0.0064 | 0.0143 | 0.0078 |
| | CI_Upper | 0.0258 | 0.2292 | 0.2345 | -0.0677 | 0.0112 | 0.2850 | 0.2458 | -0.1661 | 0.4064 | 0.1873 | 0.2497 | 0.4587 | 0.0463 | 0.4064 | 0.1873 |
| | CI_Lower | -0.1688 | 0.1748 | 0.2025 | -0.0905 | -0.0136 | 0.0596 | 0.1865 | -0.2033 | 0.3503 | 0.1480 | -0.0481 | 0.3776 | 0.0213 | 0.3503 | 0.1479 |
| Pathway 2 | Empirical Bias | 0.0185 | -0.0114 | -0.0653 | 0.0158 | 0.0007 | 0.1397 | 0.0643 | 0.1542 | 0.0942 | -0.0231 | 0.0530 | 0.0889 | -0.0573 | | -0.0224 |
| | SE | 0.0306 | 0.0061 | 0.0081 | 0.0371 | 0.0057 | 0.0759 | 0.0090 | 0.0091 | 0.0328 | 0.0139 | 0.0844 | 0.0103 | 0.0123 | 0.1629 | 0.0122 |
| | CI_Upper | 0.0677 | 0.0494 | 0.0366 | 0.1072 | 0.0829 | 0.2857 | 0.2300 | 0.3270 | 0.1403 | 0.2662 | 0.3128 | 0.2700 | 0.2379 | 0.6795 | 0.3491 |
| | CI_Lower | -0.0524 | 0.0255 | 0.0142 | 0.0757 | -0.0623 | -0.0118 | 0.1947 | 0.2727 | 0.1047 | 0.0689 | -0.0183 | 0.2296 | 0.1899 | 0.0408 | 0.3014 |

Table 2 reports the performance metrics of SIGMA across different dimensionalities. As feature dimensionality increases from 500 to 2000, all three core evaluation metrics exhibit a declining trend: Adjacency F1 decreases by 22.5% (from 0.603 to 0.467), Orientation F1 decreases by 25.9% (from 0.541 to 0.401), and Path F1 decreases by 26.9% (from 0.587 to 0.429). Despite this performance degradation, SIGMA maintains an edge recovery F1 of 0.467 and a pathway identification F1 of 0.429 in the 2000-dimensional configuration, with all standard errors controlled within 0.083. These results confirm the scalability of the SIGMA framework for high-dimensional mediation analysis.

Table 2: SIGMA high-dimensional scalability analysis. Setup: three synthetic configurations with $p \in \{500, 1000, 2000\}$; nonlinearity degree $\rho = 0.5$; parallel ($\kappa = 2$) and chain ($\ell = 2$) structures; $n = 6000$; 10 independent runs. Metrics are reported as mean (SE).

| Dimension ($p$) | Adj F1 (Mean) | Adj F1 (SE) | Orient F1 (Mean) | Orient F1 (SE) | Path F1 (Mean) | Path F1 (SE) |
|---|---|---|---|---|---|---|
| 500 | 0.603 | 0.044 | 0.541 | 0.051 | 0.587 | 0.047 |
| 1000 | 0.548 | 0.058 | 0.479 | 0.066 | 0.521 | 0.062 |
| 2000 | 0.467 | 0.074 | 0.401 | 0.083 | 0.429 | 0.079 |

## 5.3   Real Data HPP Validation

Validation on the HPP dataset shows that the SIGMA framework successfully identified multiple physiologically meaningful mediation pathways (Figure 3). Specifically, the pathway (sleep HRV index → pulse wave velocity → carotid IMT) verified a known mechanism whereby sleep quality affects cardiovascular health via vascular function. Previous studies have reported associations between reduced sleep quality and autonomic dysfunction, indicated by decreased heart rate variability (HRV), which is linked to increased arterial stiffness (PWV) and subsequently greater carotid IMT [11, 32, 51]. Additionally, SIGMA also identified a pathway where sleep snoring severity affects carotid IMT via body fat percentage and blood pressure regulation. This suggest an indirect effect of sleep quality on vascular health mediated through metabolism and blood pressure. Existing clinical studies have linked severe snoring and obstructive sleep apnea (OSA) to obesity via endocrine disturbances [24]. Obesity, particularly increased visceral fat, has been linked to elevated blood pressure [73], and sustained high blood pressure is associated with increased carotid IMT [37]. Moreover, as illustrated in Figure 3, the analysis reveals associations among nocturnal hypoxic burden, OSA, body fat composition, carotid IMT, and gut microbiota abundance. This newly identified mediation pathway suggests connections among sleep-disordered breathing, metabolism, cardiovascular health, and gut microbiota ecology. This indicates that abnormal body fat distribution induced by sleep disturbances may further affect cardiovascular health through elevated blood pressure and autonomic dysfunction. Classical medical regression mediation analysis further validated these pathways, with an overall TE error of 0.2734 and consistent NIE and NDE estimation directions. Numerical results and visualization details are provided in the Appendix F.4.

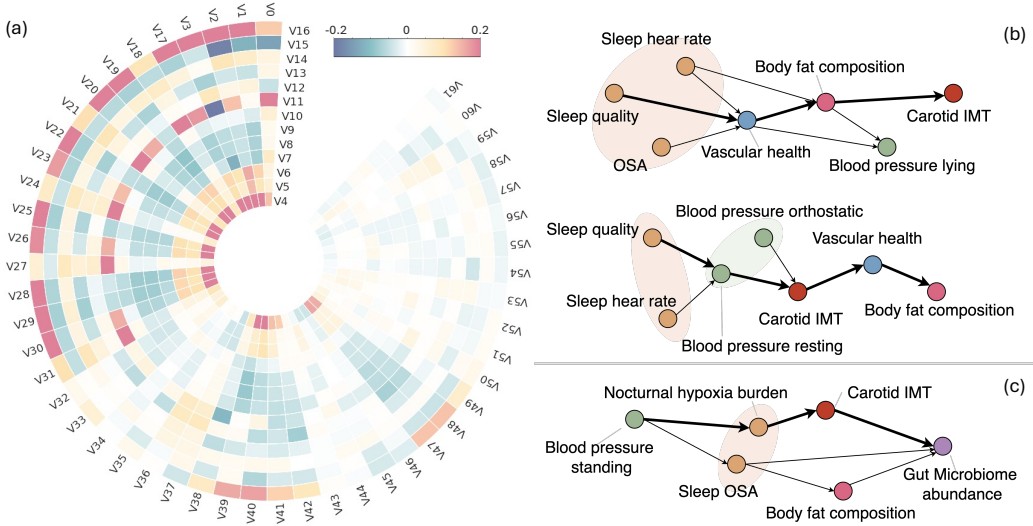

Figure 3: (a) Correlations of key sleep-derived features with features derived from other body systems. See Table 3 for the corresponding features description of V0-V61. (b) The mediation pathways reproduce a previously reported mechanism; (c) Novely discovered pathways by SIGMA, showing potential links across sleep, metabolic state, vascular function, and gut microbiome composition.

## 6 Related Work

CMA has evolved from estimating total treatment effects to decomposing pathways into direct and indirect components [27, 45, 49]. Semiparametric methods leveraging EIFs have been widely adopted for this purpose, offering multiple robustness and asymptotic efficiency [59, 71]. Recent work such as DeepMed [65] advances this line by using deep neural networks to estimate high-dimensional nuisance functions without sparsity constraints, achieving semiparametric efficiency under weaker smoothness assumptions. However, these methods generally assume a known mediation structure. Parallelly, causal discovery frameworks like NOTEARS [72], GES [56], and DAG-GNN [69] aim to recover causal graphs from observational data but do not target mediation-specific pathways. DECI [22] proposes a variational approach that jointly infers causal structure and structural equations, supporting downstream inference via interventional queries. While powerful, DECI and similar methods [15, 60] focus on recovering global structure and interventional distributions rather than quantifying pathway-specific mediation effects. Other works in high-dimensional mediation [7, 18] have explored multi-mediator identification but typically rely on predefined structures or linear assumptions. This disconnect between causal discovery and effect estimation limits their applicability in complex, uncertain systems. Please refer Appendix C for detailed literature review.

## 7 Conclusion, Limitations and Future

The SIGMA framework we proposed provides an end-to-end solution for causal mediation analysis under structural uncertainty by unifying structure discovery, pathway identification, and effect estimation. The core innovation of this framework lies in: quantifying causal graph uncertainty through probabilistic structural posterior learning, automatically identifying high-confidence mediation pathways via the Path Stability Score, and systematically propagating structural uncertainty to final effect estimates through Bayesian model averaging. In validation experiments on HPP data, SIGMA confirmed known sleep-cardiovascular mediation relationships and discovered previously unspecified cross-system mediation pathways, revealing how sleep quality affects cardiovascular health through inflammatory and metabolic pathways, providing new targets for clinical intervention. However, as an observational research method, SIGMA remains limited by unmeasured confounders and selection bias. To address these challenges, we plan to conduct clinical intervention experiments, validating the inflammatory and metabolic mediation mechanisms discovered by SIGMA through sleep quality interventions (e.g., CPAP treatment, sleep hygiene improvement), providing stronger causal evidence, and developing sensitivity analysis tools to systematically assess the robustness of inference results, thereby facilitating translation from computational discoveries to clinical applications.

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

# A Detailed Discussion of Causal Assumptions

## A.1 Core Assumptions for Causal Structure Discovery and Identification

The validity of causal structures inferred by Flow-SEM—and, by extension, the SIGMA framework—relies on a set of core assumptions that connect observed statistical dependencies to underlying causal mechanisms. In particular, the following three foundational assumptions enable reliable identification and interpretation of causal graphs from observational data.

### A.1.1 The Causal Markov Assumption

**Formal Definition.** A probability distribution $P(V)$ is said to satisfy the **Causal Markov Condition** with respect to a DAG $G = (V, E)$ if each variable $X_i \in V$ is conditionally independent of its non-descendants, given its parents in the graph. Formally, for all $X_i \in V$:

$$X_i \perp\!\!\!\perp \mathrm{NonDesc}_G(X_i) \setminus \mathrm{Pa}_G(X_i) \mid \mathrm{Pa}_G(X_i), \tag{8}$$

where $\mathrm{Pa}_G(X_i)$ denotes the set of parent nodes of $X_i$, and $\mathrm{NonDesc}_G(X_i)$ denotes its non-descendants in $G$. For DAGs, this local condition is equivalent to the global Markov condition defined via *d-separation*: for disjoint subsets $X, Y, Z \subseteq V$, if $Z$ d-separates $X$ and $Y$ in $G$, then $X \perp\!\!\!\perp Y \mid Z$ in $P(V)$.

**Role in SIGMA.** The Causal Markov Assumption justifies the structural factorization used by Flow-SEM in SIGMA. Specifically, it enables the modeling of the joint distribution $P(V \mid \boldsymbol{W_\theta}, \boldsymbol{\Phi})$ as the product of conditionals:

$$P(V \mid \boldsymbol{W_\theta}, \boldsymbol{\Phi}) = \prod_{i=1}^{p} P(V_i \mid \mathrm{Pa}_G(V_i), \phi_i), \tag{9}$$

which underpins the Structural Equation Model (SEM) formulation:

$$V_i = f_i(\mathrm{Pa}_G(V_i); \phi_i) + \sigma_i(\mathrm{Pa}_G(V_i); \phi_i) \cdot U_i, \tag{10}$$

where $f_i$ and $\sigma_i$ are flexible functions (e.g., neural networks), and $U_i \sim \mathcal{N}(0, 1)$ represents exogenous noise. This factorization supports the likelihood-based training objective used in Flow-SEM (see Section 4.2.1). In particular, Equation 3 optimizes the negative log-likelihood while enforcing acyclicity and sparsity constraints. Without the Causal Markov assumption, the inferred structure may not reflect the true causal dependencies, even when the model achieves high likelihood under the observed data distribution.

### A.1.2 The Faithfulness Assumption

**Formal Definition.** A distribution $P(V)$ is **faithful** to a DAG $G = (V, E)$ if every conditional independence observed in $P(V)$ corresponds to a d-separation in $G$. Formally, for any disjoint subsets $X, Y, Z \subseteq V$:

$$X \perp\!\!\!\perp Y \mid Z \text{ in } P(V) \iff Z \text{ d-separates } X \text{ from } Y \text{ in } G. \tag{11}$$

The forward implication ($\Rightarrow$) corresponds to the Causal Markov Assumption, while the reverse ($\Leftarrow$) constitutes the Faithfulness Assumption.

**Role in SIGMA.** Faithfulness is critical to ensuring that Flow-SEM accurately infers the presence or absence of causal edges. Violations occur when distinct causal pathways counteract each other such that their net effect cancels out. For instance, in competing mediation paths:

$$\beta_{AM_1} \cdot \beta_{M_1Y} = -\beta_{AM_2} \cdot \beta_{M_2Y}, \tag{12}$$

which can induce conditional independencies not reflected by any d-separation in the true DAG. While such exact cancellations are measure-zero events in parameter space, they can still arise in practical scenarios.

Violations of faithfulness in SIGMA result in biased structure learning. Specifically, the learned posterior $P(G_s \mid D)$ may diverge from the true posterior over the causal graph $G^*$:

$$P(G_s \mid D) \neq P(G^* \mid D), \tag{13}$$

leading Flow-SEM to favor incorrect structural hypotheses. This has downstream effects on mediation pathway identification, particularly through the computation of the Path Stability Score (PSS), defined as:

$$\text{PSS}(\pi) = \frac{1}{N_{\text{valid}}} \sum_{s=1}^{N_{\text{DAGs}}} \mathbb{I}(\pi \subseteq G_s), \tag{14}$$

where $\pi$ denotes a candidate mediation path, and the indicator function returns 1 if all directed edges in $\pi$ exist in the sampled DAG $G_s$. As a result, true causal pathways may be assigned artificially low PSS values due to systematically missing edges, reducing their chances of being selected for downstream effect estimation.

### A.1.3 The Causal Sufficiency Assumption

**Formal Definition.** A variable set $V_{\text{obs}}$ is causally sufficient if for every pair $X, Y \in V_{\text{obs}}$, all common causes are also in $V_{\text{obs}}$. Equivalently, no unmeasured variable $U \notin V_{\text{obs}}$ is a direct cause of two or more variables in $V_{\text{obs}}$.

**Role in SIGMA.** Causal sufficiency plays a pivotal role across all components of SIGMA. In the structure discovery phase, unmeasured confounders can distort edge estimation, leading to divergence between the learned and true graph posteriors:

$$P(G \mid D) \neq P(G^* \mid D), \tag{15}$$

where $G^*$ denotes the true underlying causal graph.

In the effect estimation phase (Section 4.4), causal sufficiency affects the correctness of nuisance function estimation. These functions, which are essential for computing NDE and NIE, depend on the appropriate conditioning sets defined by the graph:

$$\hat{\eta}_s = f(\text{Pa}_{G_s}(\cdot)). \tag{16}$$

When $G_s$ is misspecified due to unmeasured confounding, the adjustment sets are invalid, thereby violating key identification assumptions such as:

$$A \per\!\!\!\perp Y(a^*, m), M(a) \mid C, G_s \quad \text{and} \quad M \per\!\!\!\perp Y(a, m') \mid A = a, C, G_s. \tag{17}$$

SIGMA assumes that the full variable set $V = \{A, Y\} \cup M \cup C$ is causally sufficient, or that any substantial unobserved confounding is accounted for within $C$. When this assumption is questionable, violations can be partially mitigated through sensitivity analysis.

### A.1.4 SIGMA Framework to Challenges in Core Causal Assumptions

The efficacy of inferring causal structures $G = (V, E)$ and quantifying mediational pathway effects from observational data $D$ is intrinsically linked to a set of foundational causal assumptions. The SIGMA framework incorporates specific mechanisms in its design to consider and respond to the challenges these assumptions may face in complex real-world systems.

Considerations regarding the Causal Sufficiency Assumption

The Causal Sufficiency assumption, requiring that all common causes $U$ affecting any two variables $V_i, V_j \in V$ within the system are included in the observed set $V$ (i.e., $U \subseteq V$), is an ideal yet often challenging prerequisite for unbiased causal effect estimation. When this condition is potentially violated, implying the existence of unmeasured confounders $U \notin V$, SIGMA offers a more robust analytical pathway through the following design considerations:

**Structured Integration of High-Dimensional Covariates:** SIGMA mandates the explicit inclusion of a set of covariates $C$ within the observed variable set $V = \{A, Y\} \cup M \cup C$ (see Section 3 of the

main text). The Flow-SEM module (Section 4.2.1 of the main text) incorporates these covariates $C$ into the structural learning process, treating them equivalently to other variables ($A, M_k \in M, Y$). This allows for the data-driven discovery of relationships between $C$ and $A, M, Y$. By employing flexible functions $f_i(\text{Pa}_G(V_i); \phi_i)$ (Equation 1 of the main text), SIGMA can capture complex, non-linear confounding patterns mediated by the observed covariates $C$, thereby adjusting for such confounding at a structural level, which surpasses traditional methods that often treat $C$ merely as linear control variables.

**Exploration of Structural Posterior Distribution:** Rather than relying on a single, potentially misspecified, predefined causal graph, SIGMA leverages Flow-SEM to learn a parameterized posterior distribution $P(G|D, \boldsymbol{W_\theta}, \boldsymbol{\Phi})$ over causal structures. From this distribution, an ensemble of DAGs $\mathcal{G} = \{G_s\}_{s=1}^{N_{\text{DAG}}}$ is sampled (Section 4.2.3 of the main text). This ensemble $\mathcal{G}$ represents a multiplicity of causal structure hypotheses compatible with the observed data $D$ and the given variable set $V$. This strategy acknowledges that, even conditional on $V$, the true causal structure may be uncertain. Different $G_s \in \mathcal{G}$ may account for complex dependencies arising from $C$ in various ways, thus providing a more comprehensive examination of confounding effects mediated by *observed* variables.

**Structure-Guided Bayesian Model Averaging:** For each candidate pathway $\pi$ identified via the Path Stability Score (PSS) (Section 4.3 of the main text), SIGMA estimates pathway-specific effects $\hat{\theta}_s(\pi)$ under each supporting DAG $G_s \in \mathcal{G}_\pi \subseteq \mathcal{G}$. The nuisance functions $\hat{\eta}_s$ are parameterized based on the parent sets $\text{Pa}_{G_s}(\cdot)$ (including relevant covariates from $C$) defined by that specific $G_s$ (Sections 4.4.1, 4.4.2 of the main text). The final effect estimate $\hat{\theta}_{\text{BMA}}(\pi)$ and its variance $\hat{V}(\hat{\theta}_{\text{BMA}}(\pi))$ are obtained by BMA over $\mathcal{G}_\pi$ (Section 4.4.3, Equation 7 of the main text). This aggregation ensures that the final inference incorporates not only the statistical uncertainty $\hat{V}(\hat{\theta}_s(\pi))$ within each specific structure but also the structural uncertainty, $\text{Var}_{G_s \in \mathcal{G}_\pi}[\hat{\theta}_s(\pi)]$, arising from the heterogeneity of causal hypotheses consistent with the observed data $V$.

Although SIGMA cannot intrinsically detect or fully neutralize biases from truly unmeasured confounders (i.e., those not included in $V$), its design aims to maximize the utility of all available information within $V$ (especially covariates $C$). Through structured modeling of observed covariates, explicit modeling of structural uncertainty, and aggregation over multiple structural hypotheses, SIGMA provides a more systematic and data-driven framework than analyses relying on a single, fixed causal model to mitigate biases stemming from confounding by *observed* variables. For potential unmeasured confounding beyond the scope of $V$, sensitivity analysis, as recommended in Appendix A.1.3 of the main paper, remains an important complementary tool for assessing the robustness of conclusions.

Considerations regarding the Causal Faithfulness Assumption

The Causal Faithfulness assumption posits that all conditional independence relations present in the probability distribution $P(V)$ are entailed by d-separations in the true causal graph $G^*$, and vice-versa (Sections 3, A.1.2 of the main text). This implies the absence of statistical independencies arising from exact cancellations of path effects (e.g., $\sum_{\pi \in \Pi_{A \rightsquigarrow Y}} \text{effect}(\pi) \approx 0$ while $\Pi_{A \rightsquigarrow Y} \neq \emptyset$), where underlying causal pathways exist. SIGMA's mechanisms offer the following considerations when facing challenges of approximate or exact unfaithfulness:

**Continuous Edge Weight Learning and Probabilistic Representation:** Flow-SEM (Section 4.2.3 of the main text) learns a continuous adjacency weight matrix $\boldsymbol{W_\theta}$, where entries $|[\boldsymbol{W_\theta}]_{ji}|$ reflect the strength of a potential edge $V_j \to V_i$. These weights are transformed into edge inclusion probabilities $p_{ji} = P(e_{ji} \in E|D)$ via $\sigma(|[\boldsymbol{W_\theta}]_{ji}|)$ (Section 4.2.3 of the main text). In scenarios of approximate unfaithfulness, where the net effect of a path is close to zero, diminishing the signal for its constituent edges, the corresponding $p_{ji}$ values may be low but not necessarily zero (unless strong sparsity priors are imposed). This allows edges with weak signals a non-zero probability of being included in sampled DAGs, thus mitigating premature exclusion based on hard independence thresholds.

**Statistical Filtering via Path Stability Score (PSS):** The $\text{PSS}(\pi)$ (Section 4.3, Equation 5 of the main text) is defined as the frequency of path $\pi$'s appearance across $N_{\text{valid}}$ sampled DAGs, $G_s$, serving as a Monte Carlo estimate, $E_{G \sim P(G|D)}[\mathbb{I}(\pi \subseteq G)]$, of its marginal inclusion probability under the learned posterior $P(G|D)$. If a true path's signal is attenuated due to approximate unfaithfulness affecting one of its constituent edges $e_{ji}$ (resulting in a low $p_{ji}$), the joint probability of observing $\pi$ (and thus its $\text{PSS}(\pi)$) will be reduced. By thresholding $\text{PSS}(\pi)$ with $\tau$, SIGMA selects paths that

exhibit sufficient statistical consistency across the ensemble, effectively focusing on pathways whose structural evidence remains discernible despite data noise and potential complexities in parameter configurations.

**Diversity of Structural Ensemble and BMA Weighting:** SIGMA generates a diverse ensemble of DAGs $\mathcal{G}$. Even if certain parameter configurations lead to the apparent weakness or absence of a specific path in some $G_s \in \mathcal{G}$ due to approximate unfaithfulness, such a path might still manifest in other $G_s$ corresponding to different plausible parameterizations, provided some evidential support exists in $D$. While the BMA procedure in this work employs uniform weights for selected paths, the PSS acts as a preceding filter. Paths consistently undermined by strong approximate unfaithfulness, resulting in PSS values below $\tau$, are excluded from the final effect aggregation. This prioritizes mediational mechanisms with more robust and identifiable signals in the observed data.

Consequently, while SIGMA cannot resolve fundamental unidentifiability stemming from perfect parameter cancellations, its probabilistic and ensemble-based framework allows for a more nuanced, data-driven assessment of pathways whose signals might be weakened by approximate unfaithfulness. It quantifies the evidential strength for each path via PSS, thereby focusing inferential resources on mediational mechanisms that are statistically stable and discernible under the learned posterior distribution of structures.

Considerations regarding the Causal Markov Condition

The Causal Markov Condition, a cornerstone of graphical causal models, states that any variable $V_i$ is conditionally independent of its non-descendants (excluding its parents) given its direct causes $\text{Pa}_G(V_i)$ in a DAG $G$ (Section A.1.3 of the main text). SIGMA's design inherently relies on and leverages this assumption:

**DAG-based Parametrization:** The Flow-SEM module (Section 4.2.1 of the main text), central to SIGMA's structure discovery, models inter-variable relationships using a Directed Acyclic Graph (DAG). The structural equations $V_i = f_i(\text{Pa}_G(V_i); \phi_i) + \sigma_i(\text{Pa}_G(V_i); \phi_i) \cdot U_i$ (Equation 1 of the main text) explicitly encode that the distribution of $V_i$ is determined solely by its parents $\text{Pa}_G(V_i)$ and an exogenous noise term $U_i$.

**Factorization of Likelihood Function:** The log-likelihood function $\log P(V^{(j)} | \boldsymbol{W_\theta}, \boldsymbol{\Phi})$ in Flow-SEM (Equation 2 of the main text) is expressed as a sum over terms corresponding to each variable $V_i$ conditioned on its parents $\text{Pa}_G(V_i^{(j)})$, i.e., $\sum_{i=1}^{p} \left( \log p_U(u_i^{(j)}) - \log \sigma_i(\text{Pa}_G(V_i^{(j)}); \phi_i) \right)$. This factorization is a direct mathematical consequence of the Causal Markov Condition, rendering the learning of high-dimensional joint distributions computationally tractable.

**Flexible Functional Forms and Structural Exploration:** By employing neural networks for $f_i$ and $\sigma_i$, SIGMA accommodates complex non-linear dependencies. The learning of $\boldsymbol{W_\theta}$ and subsequent sampling of the DAG ensemble $\mathcal{G}$ allow SIGMA to explore a rich space of DAG structures that are compatible with the data and inherently satisfy the Markov condition. This combination enables SIGMA to seek the best description of observed data dependencies within the class of Markovian models representable by DAGs.

## A.2 Robustness Considerations and Sensitivity Analysis of SIGMA

The theoretical guarantees for SIGMA framework validity rely on a series of core causal assumptions, including the Causal Markov condition, Faithfulness, and Causal Sufficiency defined in Appendices A.1, as well as effect identification conditions such as Ignorability and Positivity specified in Section 3 of the main text. In practical application contexts, these assumptions may be violated to varying degrees, threatening the validity and reliability of the estimated mediational pathway effects. Therefore, systematic analysis of the robustness and sensitivity of results obtained under the SIGMA framework becomes particularly necessary. Next, we discuss potential methodological frameworks for sensitivity analysis and mathematical prospects for future research.

### A.2.1 Sensitivity Analysis under Deviations from Causal Sufficiency

Consider an unobserved potential confounding variable $U$ that simultaneously affects exposure $A$, the set of mediator variables $M$, and the outcome variable $Y$. In the presence of such unobserved confounding, we define a confounding sensitivity analysis framework modulated by $U$ as follows:

Assume the confounding variable $U$ is simultaneously associated with exposure and mediators, as well as outcomes and mediators, with their association strengths described by the following two sensitivity parameters $\Gamma_{AU}, \Gamma_{UY}$:

$$\Gamma_{AU} = \frac{\sup_u \Pr(A = a|U = u, C)}{\inf_u \Pr(A = a|U = u, C)}, \quad \Gamma_{UY} = \frac{\sup_u E[Y|M = m, U = u, A = a, C]}{\inf_u E[Y|M = m, U = u, A = a, C]}$$

Here $\Gamma_{AU}, \Gamma_{UY} \geq 1$, with larger values indicating stronger influence of the unobserved confounding variable. To quantify the robustness of observed effects to the strength of the above unobserved confounding, the E-value [62] can be used, defined as:

$$\text{E} - \text{value}(\hat{\theta}_{\text{BMA}}(\pi)) = \inf\{\Gamma_{AU}, \Gamma_{UY} \geq 1 : 0 \in \text{CI}_{\Gamma_{AU}, \Gamma_{UY}}(\hat{\theta}_{\text{BMA}}(\pi))\}$$

where $\text{CI}_{\Gamma_{AU}, \Gamma_{UY}}(\hat{\theta}_{\text{BMA}}(\pi))$ represents the confidence interval for the effect estimate adjusted for $U$ given the sensitivity parameters. A larger E-value indicates that the conclusion is more robust to unobserved confounding; conversely, a smaller E-value indicates weaker robustness. Future research could consider further integrating the above sensitivity analysis automatically into the SIGMA process, using simulation studies to quantify its sensitivity performance under various data-generating mechanisms.

### A.2.2 Robustness Analysis under Deviations from Causal Faithfulness

Suppose the data-generating mechanism has approximate effect cancellations, where the true effect of some path $\pi$ in the path set $\Pi_{A \to Y}$ is $\theta(\pi) \approx 0$, despite this path existing in the true causal structure (i.e., approximate unfaithfulness). This scenario leads to the estimated Path Stability Score (PSS, Section 4.3) tending toward lower values. Sensitivity analysis considers a path stability threshold $\tau \in [0, 1]$, defining a stable path set as:

$$\Pi_\tau = \{\pi : \text{PSS}(\pi) \geq \tau\}$$

By studying the impact of changes in $\tau$ on the final Bayesian Model Averaging (BMA) estimate $\hat{\theta}_{\text{BMA}}(\pi|\tau)$, we can examine the sensitivity of $\hat{\theta}_{\text{BMA}}(\pi|\tau)$ to different $\tau$:

$$\frac{\partial \hat{\theta}_{\text{BMA}}(\pi|\tau)}{\partial \tau}, \quad \pi \in \Pi_\tau$$

In applications, if low sensitivity is found (the above derivative is small), it indicates that the SIGMA estimation conclusion is relatively robust; conversely, if high sensitivity is found, it may suggest that approximate unfaithfulness significantly affects the analysis, requiring cautious interpretation. Future research could deeply explore the properties of this derivative and its theoretical and empirical performance under different structural assumptions.

### A.2.3 Robustness Analysis of Model Specification Errors in Effect Estimation

When estimating path-specific effects $\hat{\theta}_{\text{BMA}}(\pi)$, SIGMA involves multiple nuisance functions (e.g., propensity scores and conditional expectation functions). Setting the bias of these nuisance function estimates as $\delta$, the following form of bias sensitivity analysis can be considered:

Define the sensitivity function of estimator bias to nuisance function estimation bias:

$$\frac{\partial \hat{\theta}_{\text{BMA}}(\pi)}{\partial \delta}$$

Lower sensitivity values imply higher robustness of the estimator to model specification errors, and vice versa. The Cross-Fitting strategy adopted in SIGMA provides a certain degree of robustness guarantee, but future research should systematically compare the sensitivity function characteristics of this estimation strategy with other semiparametric doubly robust estimators.

### A.2.4 Interplay

The three assumptions—Causal Markov, Faithfulness, and Causal Sufficiency—jointly define the identification conditions for reliable causal discovery in SIGMA. The Causal Markov condition links the graphical structure to conditional independence relations. The Faithfulness assumption ensures that all observed independencies reflect actual graphical separations rather than coincidental parameter cancellations. Causal Sufficiency guarantees that all relevant common causes are observed, preventing bias from unmeasured confounding. Together, these assumptions enable SIGMA to learn meaningful causal structures from observational data and propagate this structure into consistent mediation effect estimation.

### A.3 Identification Assumptions for NDE and NIE

TThe nonparametric identification of Natural Direct Effects (NDE) and Natural Indirect Effects (NIE) requires a set of conditional independence assumptions that account for confounding. These effects are defined in the potential outcomes framework (see Section 3) as: $\text{NDE}(a, a^*) = \mathbb{E}[Y(a, M(a^*)) - Y(a^*, M(a^*))]$ and $\text{NIE}(a, a^*) = \mathbb{E}[Y(a, M(a)) - Y(a, M(a^*))]$, These assumptions ensure that observed statistical associations admit a valid causal interpretation. In the context of SIGMA, this is particularly important because the underlying causal graph $G^*$ is unknown and approximated via a sampled ensemble $\{G_s\}$.

#### A.3.1 Ignorability Assumptions for NDE and NIE Identification

A common set of sufficient conditions for the identification of NDE and NIE, often referred to as "sequential ignorability" or "no unmeasured confounding" assumptions, can be stated conditional on a set of pre-treatment covariates $C$ [45, 8, 49]. For treatment levels $a$ and $a^*$:

**1. Conditional independence of treatment and potential outcomes/mediators (Controlling for A-M and A-Y confounding):** There is no unmeasured confounding of the effect of $A$ on $M$, and no unmeasured confounding of the effect of $A$ on $Y$ that is not mediated by $M$. Formally:

$$\forall a' \in \mathcal{A}: \quad A \perp\!\!\!\perp M(a') \mid C, \quad \forall a'', \forall m \in \mathcal{M}: \quad A \perp\!\!\!\perp Y(a'', m) \mid C. \tag{18}$$

Together, these imply that $C$ blocks all back-door paths from $A$ to $M$, and from $A$ to $Y$ that do not go through $M$. A common shorthand formulation is:

$$(Y(a^*, m), M(a)) \perp\!\!\!\perp A \mid C. \tag{19}$$

**2. Conditional independence of mediator and potential outcomes, given treatment and covariates (Controlling for M-Y confounding):** This assumption requires that, conditional on treatment $A$ and covariates $C$, there is no unmeasured confounding between the mediator $M$ and the outcome $Y$:

$$\forall a \in \mathcal{A}, \ \forall m' \in \mathcal{M}: \quad M \perp\!\!\!\perp Y(a, m') \mid A = a, C. \tag{20}$$

Under this assumption, among units with the same treatment and covariates, the mediator behaves as if it were randomly assigned with respect to the counterfactual outcome $Y(a, m')$.

**3. Conditional independence of treatment and potential outcomes, given mediator and covariates (No treatment-induced M-Y confounding that also depends on A directly):** This assumption rules out any variable that is affected by treatment and also confounds the relationship between $M$ and $Y$. In some formulations, this condition is written as:

$$A \perp\!\!\!\perp Y(a^*, m) \mid M = m, C. \tag{21}$$

However, this expression involves *cross-world counterfactuals*, which are often avoided in practice. Instead, most identification strategies rely on single-world assumptions derived from the causal graph.

For example, the *mediation formula* [45] provides a means to identify NIE using the following expression:

$$\mathbb{E}[Y(a, M(a^*))] = \sum_m \mathbb{E}[Y \mid A = a, M = m, C = c] \cdot P(M = m \mid A = a^*, C = c) \cdot P(C = c). \tag{22}$$

This formula is valid under Assumptions 1 and 2, as well as the Causal Markov and Faithfulness assumptions for the underlying DAG encoding the relationships among $A, M, Y, C$. Within the SIGMA framework, pathway-specific effect estimation relies on Efficient Influence Functions (EIFs), which are computed relative to each sampled graph $G_s$. For example, to estimate the effect along a mediation pathway $A \to M_j \to Y$, SIGMA computes the following nuisance quantities:

$$\mathbb{E}[Y \mid A, \ M_j, \ \text{Pa}_{G_s}(Y) \setminus \{A, M_j\}, \ C], \tag{23}$$

$$P(M_j \mid A, \ \text{Pa}_{G_s}(M_j) \setminus \{A\}, \ C), \quad P(A \mid \text{Pa}_{G_s}(A), \ C). \tag{24}$$

These estimates rely on graph-structured adjustment sets and implicitly assume that the corresponding back-door paths are blocked. As such, the validity of EIF-based identification hinges on the correctness of the local parent sets derived from $G_s$. In cases where no variable is jointly affected by $A$ and also acts as a confounder of the $M \to Y$ link, the stronger cross-world independence condition $Y(a^*, m) \perp\!\!\!\perp M(a) \mid C$ can be avoided [49].

**Role of Graphical Structure $G_s$ in Satisfying Ignorability within SIGMA.** In SIGMA, the Flow-SEM module generates an ensemble of candidate DAGs $\{G_s\}$. For each graph $G_s = (V, E_s)$, the estimation of nuisance functions $\hat{\eta}_s$ (see Section 4.4.1), such as the conditional expectation $\mathbb{E}[Y \mid A, M, Z_Y]$ and the conditional probability $P(M \mid A, Z_M)$, relies on structure-specific conditioning sets. These sets are constructed from the local parent sets $\text{Pa}_{G_s}(\cdot)$ defined by each DAG, together with the baseline covariates $C$. Specifically, we define:

$$Z_Y = C \cup \text{Pa}_{G_s}(Y) \setminus \{A, M\}, \quad Z_M = C \cup \text{Pa}_{G_s}(M) \setminus \{A\}. \tag{25}$$

This formulation reflects the assumption that causal relationships are encoded in $G_s$, and that proper conditioning on parent sets and covariates is sufficient to block non-causal pathways. The theoretical justification rests on Pearl's d-separation criterion and the back-door adjustment principle [46].

Under ignorability Condition 1, which requires $A \perp\!\!\!\perp M(a') \mid C$ and $A \perp\!\!\!\perp Y(a'', m) \mid C$, a set of variables—denoted $C'$—must block all back-door paths from $A$ to $M$, and from $A$ to $Y$ that are not mediated through $M$. These paths must begin with an arrow into $A$, and blocking them ensures that the necessary conditional independencies hold in $G_s$. For Condition 2, which assumes $M \perp\!\!\!\perp Y(a, m') \mid A = a, C$, the relevant conditioning set must block all back-door paths between $M$ and $Y$, conditional on $A = a$. This is achieved by the set $C \cup \text{Pa}_{G_s}(Y) \setminus \{A, M\} \cup \text{Pa}_{G_s}(M) \setminus \{A\}$, where conditioning on $A = a$ accounts for any paths passing through $A$ as either a collider or a non-collider.

The central premise is that each sampled DAG $G_s$ is a candidate for the true underlying causal graph $G^*$, and that the observed variable set $V_{\text{obs}} = \{A, Y\} \cup M \cup C$ satisfies causal sufficiency. When these conditions hold, and if $G_s$ accurately reflects the structural relationships encoded in $G^*$, then using the parent sets $\text{Pa}_{G_s}(\cdot)$ in conjunction with $C$ provides a principled approach for selecting conditioning variables based on the d-separation criterion. For instance, if $\text{Pa}_{G_s}(M)$ includes all common causes of $M$ and $Y$ that are not affected by $A$ and are not in $C$, then conditioning on $A, C, \text{Pa}_{G_s}(M) \setminus \{A\}$ is sufficient to satisfy the M–Y ignorability condition:

$$M \perp\!\!\!\perp Y(a, m') \mid A = a, \ C \cup \text{Pa}_{G_s}(M) \setminus \{A\}. \tag{26}$$

However, since each $G_s$ is only a hypothesis, ignorability is not guaranteed unless $G_s = G^*$ and the covariate set $C$ is truly sufficient. If either the graph is misspecified or unmeasured confounders are omitted from both $C$ and the parent sets, the resulting conditioning set $C \cup \text{Pa}_{G_s}(\cdot)$ may be inadequate. This misalignment can lead to biased nuisance function estimates and consequently biased causal effect estimates $\hat{\theta}_s(\pi)$. While the BMA procedure in SIGMA propagates structural uncertainty by averaging across sampled DAGs, it does not inherently resolve violations of ignorability that are present across all candidate structures. Therefore, rigorous selection of covariates $C$ remains essential for ensuring the validity of effect estimation under structural uncertainty.

### A.3.2 Consistency Assumption

The consistency assumption establishes the fundamental link between potential outcomes (defined under hypothetical interventions) and observed outcomes in the data. It posits that for any individual,

the observed outcome corresponds precisely to the potential outcome associated with the treatment and mediator values that the individual actually received.

Formally, for all individuals $i$, the assumption requires:

$$A_i = a \Rightarrow M_i = M_i(a), \quad \text{and} \quad (A_i = a, \ M_i = m) \Rightarrow Y_i = Y_i(a, m), \tag{27}$$

which means that if an individual received treatment $a$, the observed mediator equals the potential mediator under $a$; and if the individual received treatment $a$ and mediator value $m$, the observed outcome equals the potential outcome under those values.

These conditions can be restated as follows. For all individuals $i$, and any treatment level $a \in \mathcal{A}$, if $A_i = a$, then $M_i = M_i(a)$. This expresses that the observed mediator is what would have been realized under the assigned treatment. Similarly, for all $a \in \mathcal{A}$ and all $m \in \mathcal{M}$, if $A_i = a$ and $M_i = m$, then $Y_i = Y_i(a, m)$. This implies that the observed outcome reflects the potential outcome under the observed treatment and mediator values. These requirements must hold for all relevant treatment levels (e.g., $a$, $a^*$ in the definition of NDE and NIE) and all values of $m$ within the support of $M$.

Consistency is essential for expressing counterfactual estimands such as $\mathbb{E}[Y(a, M(a^*))]$ in terms of observable quantities. Under this assumption, conditional expectations from the observed distribution $P(Y, A, M, C)$ can be equated to expectations over potential outcomes. For example, the observed expectation $\mathbb{E}[Y \mid A = a, M = m, C = c]$ corresponds to $\mathbb{E}[Y(a, m) \mid A = a, M = m, C = c]$, assuming that ignorability holds for the conditioning set.

In the SIGMA framework, consistency enables the interpretation of structure-specific nuisance functions as counterfactual components. These include, for each sampled DAG $G_s$:

$$\mathbb{E}[Y \mid A, M, \mathrm{Pa}_{G_s}(Y) \setminus \{A, M\}, C], \quad P(M \mid A, \mathrm{Pa}_{G_s}(M) \setminus \{A\}, C), \tag{28}$$

which are estimated from observational data and treated as valid components of EIF-based identification strategies (see Section 4.4.1). This directly affects single-DAG effect estimates $\hat{\theta}_s(\pi)$ (Section 4.4.2) and, through Bayesian model averaging, the final estimates $\hat{\theta}_{\mathrm{BMA}}(\pi)$.

Violations of consistency occur when the treatment $A = a$ or mediator $M = m$ is ambiguously defined in the data. For example, if multiple unobserved variants of $A = a$ exist (e.g., "exercise intervention" encompassing both "light jogging" and "intense weightlifting"), and these variants have distinct effects on downstream variables, then consistency does not hold, even though the data records only a single treatment label. The same logic applies if $M$ is coarsely defined.

SIGMA implicitly assumes that treatment $A$ and mediator $M$ are sufficiently well-defined and measured with adequate granularity such that their causal interpretation is meaningful. Ensuring consistency is not typically addressed at the algorithmic level, but rather requires careful study design or strong domain knowledge to define, operationalize, and measure treatments and mediators in a causally coherent way.

### A.3.3 Positivity Assumption

The positivity assumption, sometimes referred to as overlap or the experimental treatment assignment (ETA) assumption, is essential for the nonparametric or semiparametric identification of causal effects, including NDE and NIE. It states that, for any combination of observed pre-treatment covariates $C = c$, each treatment level $A = a$ and each mediator value $M = m$ must occur with non-zero probability. This ensures adequate empirical support across all necessary strata, which is required to estimate causal contrasts without extrapolation.

Let $\mathcal{A}$ denote the set of treatment values, $\mathcal{M}$ the support of the mediator(s) $M$, and $\mathcal{C}$ the support of covariates $C$. The positivity condition for NDE and NIE identification can be stated as:

$$\forall a \in \mathcal{A}, \forall c \in \mathcal{C} \ \text{s.t.} \ P(C = c) > 0 : \quad P(A = a \mid C = c) > \epsilon_A \tag{29}$$

$$\forall a \in \mathcal{A}, \forall m \in \mathcal{M}, \forall c \in \mathcal{C} \ \text{s.t.} \ P(A = a, C = c) > 0 : \quad p(M = m \mid A = a, C = c) > \epsilon_M \tag{30}$$

where $\epsilon_A > 0$ and $\epsilon_M > 0$ are small constants ensuring strict positivity. For continuous mediators, the density $p(M = m \mid A = a, C = c)$ must be bounded away from zero over the relevant support of $M$. In practical applications, especially when using estimators involving inverse probability weighting

(e.g., EIF-based estimators), stronger bounds such as $\epsilon_A \leq P(A = a \mid C = c) \leq 1 - \epsilon_A$ are often imposed to ensure numerical stability.

In SIGMA, positivity is required for the estimation of nuisance functions $\hat{\eta}_s$ (Section 4.4.1), such as the propensity score $P(A \mid \text{Pa}_{G_s}(A), C)$ and the conditional mediator distribution $P(M_j \mid A, \text{Pa}_{G_s}(M_j) \setminus \{A\}, C)$, for each sampled DAG $G_s$. When Equation 29 or Equation 30 is violated in practice—i.e., some combinations of $A, M, C$ occur with near-zero probability—the corresponding weights in EIFs or related estimators become excessively large. This can lead to inflated variance or unstable effect estimates $\hat{\theta}_s(\pi)$.

For example, if $p(M = m \mid A = a, \text{Pa}_{G_s}(M) \setminus \{A\}, C = c)$ is close to zero in some region of the covariate space, estimating expectations such as $\mathbb{E}[Y \mid A = a, M = m, \ldots]$ requires extrapolation into areas with limited or no observed data. This weakens the reliability of the learned nuisance models and introduces instability in the resulting causal effect estimates.

These issues propagate to the final pathway-specific estimates through their effect on the single-DAG estimators $\hat{\theta}_s(\pi)$, and subsequently affect the BMA-aggregated estimates $\hat{\theta}_{\text{BMA}}(\pi)$. Although SIGMA employs flexible nuisance function models (e.g., neural networks), which may tolerate near-violations better than simpler parametric approaches, such models can still produce biased or high-variance estimates when positivity is violated. This underscores the importance of verifying covariate support overlap in practice and ensuring estimation occurs within sufficient empirical support.

**Interaction with Other Assumptions.** Positivity is intrinsically linked to other core assumptions:

- **Ignorability:** Ignorability requires adjustment for a sufficient set of confounders $C'$ (e.g., $C \cup Pa_{G_s}(\cdot)$). Positivity ensures that such adjustment is empirically feasible by guaranteeing that all levels of $A$ (and $M$) are present across all strata of $C'$. Without Positivity, even if $C'$ theoretically suffices for Ignorability, the lack of common support can make the adjustment ineffective or impossible (Appendix A.3.1).

- **Causal Sufficiency:** If Causal Sufficiency is violated, the set $C$ may be inadequate. Even if Positivity holds for the observed $C$, the inability to adjust for unobserved confounders means Ignorability may still be violated, leading to biased estimates irrespective of data overlap on $C$ (Appendix A.1.3).

**Practical Diagnostics for Positivity Violations.** While SIGMA does not include automated procedures for testing positivity, several diagnostic practices are recommended to identify potential violations:

- **Examine Propensity Score Distributions:** Plot histograms or density plots of estimated propensity scores $P(A = a \mid \text{adjustment set})$ for each treatment group. Lack of overlap or scores concentrated near 0 or 1 indicates potential Positivity issues.

- **Assess Covariate Balance:** After stratification or weighting by propensity scores (if applicable to a diagnostic step), check for balance in covariate distributions across treatment groups using metrics like Standardized Mean Differences (SMDs). Large SMDs (e.g., > 0.1-0.25) can suggest poor overlap.

- **Inspect Weight Distributions:** For estimators that explicitly use inverse probability weights (even internally, like some EIFs), examine the distribution of these weights. A few extremely large weights can dominate the estimate and inflate variance, signaling Positivity problems.

- **Check for Sparse Strata:** For conditional mediator distributions $P(M \mid A, \text{adjustment set})$, particularly with discrete mediators or high-dimensional adjustment sets, tabulate or examine the number of observations in critical $(A, M, \text{adjustment set})$ strata to identify sparse data regions.

- **Sensitivity Analysis:** Evaluate how NDE/NIE estimates change if observations with extreme propensity scores are trimmed or if the analysis is restricted to regions of better covariate overlap (e.g., trimming based on propensity score tails).

Addressing severe positivity violations may require restricting the analysis to subpopulations with sufficient empirical support or applying estimation techniques designed for limited overlap (e.g., overlap weights or covariate balancing methods). These approaches can help reduce estimation

variance but may also introduce new assumptions or trade-offs, such as reduced generalizability or increased bias.

### A.3.4 Single-DAG Effect Estimation via Influence Functions and Plugin Estimators

This appendix details the specific construction of Efficient Influence Functions (EIFs) for estimating the Natural Direct Effect (NDE) and Natural Indirect Effect (NIE) within the SIGMA framework when the exposure variable $A$ is binary (taking values 0 or 1) and the path length is 3 (i.e., $A \to M \to Y$). These EIFs depend on a series of nuisance function estimates obtained through K-Fold cross-fitting (see Section 4.4.1 in the main text for details). We define the following core nuisance functions, where the superscript $(s)$ indicates that these functions are estimated for a specific DAG $G_s$, but will be omitted in the following for brevity:

$\hat{\pi}(C_A) = \hat{P}(A = 1|C_A)$: The propensity score for exposure $A$, i.e., the estimate of the conditional probability of $A = 1$ given covariates $C_A$ (parent nodes of $A$ in $G_s$). For the mediator $M$ (which can be continuous, binary, or categorical):

- $\hat{\mu}_M(a, C_M) = \hat{E}[M|A = a, C_M]$: When $M$ is continuous or binary, the estimate of its conditional expectation given $A = a$ and covariates $C_M$ (parent nodes of $M$ in $G_s$, excluding $A$).

- $\hat{f}_M(m|a, C_M) = \hat{P}(M = m|A = a, C_M)$: When $M$ is discrete categorical, the estimate of its conditional probability mass function of taking value $m$ given $A = a$ and $C_M$. For unified notation, we sometimes still use $\hat{E}[g(M)|A = a, C_M]$ to denote the expectation of some function $g(M)$ of $M$, which can be obtained by summing/integrating over $\hat{f}_M$.

$\hat{\mu}_Y(a, m, C_Y) = \hat{E}[Y|A = a, M = m, C_Y]$: The estimate of the conditional expectation of outcome $Y$ given $A = a$, $M = m$, and covariates $C_Y$ (parent nodes of $Y$ in $G_s$, excluding $A, M$).

Furthermore, we define the plug-in estimator for counterfactual expectations (obtained via G-computation using the aforementioned nuisance functions and averaged over the sample):

$\hat{\gamma}_{a'a''} = \widehat{\mathbb{E}}[Y(a', M(a''))] = \frac{1}{n}\sum_{i=1}^{n} \int \hat{\mu}_Y(a', m, C_{Y,i})d\hat{F}_M(m|a'', C_{M,i})$, where the integral sign represents summation over all possible values of the mediator $M$ (discrete $M$) or integration (continuous $M$), and $d\hat{F}_M(m|a'', C_{M,i})$ represents the estimated conditional distribution based on $\hat{\mu}_M(a'', C_{M,i})$ or $\hat{f}_M(m|a'', C_{M,i})$.

NDE and NIE can be expressed as differences of these counterfactual expectations:

- NDE $= \mathbb{E}[Y(1, M(0))] - \mathbb{E}[Y(0, M(0))] = \gamma_{10} - \gamma_{00}$
- NIE $= \mathbb{E}[Y(1, M(1))] - \mathbb{E}[Y(1, M(0))] = \gamma_{11} - \gamma_{10}$

Therefore, the influence functions for NDE and NIE are linear combinations of the influence functions of their constituent counterfactual expectation terms. We present the influence function $\psi_{\gamma_{a'a''}}(Z_i)$ for $\mathbb{E}[Y(a', M(a''))]$ (i.e., $\gamma_{a'a''}$). For observed data $Z_i = (A_i, M_i, Y_i, C_{A,i}, C_{M,i}, C_{Y,i})$, the typical form of $\psi_{\gamma_{a'a''}}(Z_i)$ is as follows:

$$\psi_{\gamma_{a'a''}}(Z_i; \hat{\eta}_s) = \frac{I(A_i = a')}{\hat{P}(A_i = a'|C_{A,i})} \frac{\hat{f}_M(M_i|a'', C_{M,i})}{\hat{f}_M(M_i|a', C_{M,i})}(Y_i - \hat{\mu}_Y(a', M_i, C_{Y,i}))$$

$$+ \frac{I(A_i = a'')}{\hat{P}(A_i = a''|C_{A,i})}\left(\hat{\mu}_Y(a', M_i, C_{Y,i}) - \int \hat{\mu}_Y(a', m, C_{Y,i})d\hat{F}_M(m|a'', C_{M,i})\right)$$

$$+ \left(\int \hat{\mu}_Y(a', m, C_{Y,i})d\hat{F}_M(m|a'', C_{M,i}) - \hat{\gamma}_{a'a''}\right). \tag{31}$$

Here, $\hat{P}(A_i = a'|C_{A,i})$ is $\hat{\pi}(C_{A,i})$ when $a' = 1$, and $1 - \hat{\pi}(C_{A,i})$ when $a' = 0$. When $M$ is continuous, $\hat{f}_M(m|a, C_M)$ is the conditional probability density function; when $M$ is discrete, it is the conditional probability mass function. If the mediator model directly estimates the conditional expectation $\hat{\mu}_M(a, C_M)$ (e.g., $M$ is binary or continuous, and some terms in the EIF are simplified for linearity in $M$), then the parts of the formula involving $\hat{f}_M$ are adjusted accordingly. For example, when $M$ is binary $(0, 1)$, $\hat{f}_M(M_i|a, C_{M,i}) = \hat{\mu}_M(a, C_{M,i})^{M_i}(1 - \hat{\mu}_M(a, C_{M,i}))^{1-M_i}$.

Based on Equation (31), the influence functions for NDE and NIE are respectively:

$$\psi_{\text{NDE}}(Z_i; \hat{\eta}_s) = \psi_{\gamma_{10}}(Z_i; \hat{\eta}_s) - \psi_{\gamma_{00}}(Z_i; \hat{\eta}_s) \tag{32}$$

$$\psi_{\text{NIE}}(Z_i; \hat{\eta}_s) = \psi_{\gamma_{11}}(Z_i; \hat{\eta}_s) - \psi_{\gamma_{10}}(Z_i; \hat{\eta}_s) \tag{33}$$

In its implementation, the SIGMA framework computes the $\psi_{\text{NDE}}(Z_i)$ and $\psi_{\text{NIE}}(Z_i)$ scores for each sample according to the above structure, using nuisance function estimates $\hat{\eta}_s$ obtained via cross-fitting.

### A.3.5 Definition and Analysis of the Influence Function for Continuous Exposure in SIGMA

For mediation paths of length $L(\pi) = 3$ (i.e., $A \rightarrow M \rightarrow Y$) involving a continuous exposure variable $A$, the SIGMA framework defines and utilizes the influence function described herein to estimate the Natural Direct Effect (NDE) and Natural Indirect Effect (NIE). This section details the mathematical construction of this influence function and the nuisance components it relies upon.

Nuisance Function Components for the AIF

The construction of this influence function is predicated on several key nuisance function components, estimated via K-Fold cross-fitting (as detailed in Section 4.4.1) for each (path $\pi$, DAG $G_s$) pair. Let $Z_i = (A_i, M_i, Y_i, C_{A,i}, C_{M,i}, C_{Y,i})$ denote the data for the $i$-th observation, where $C_{X,i}$ represents the values of the parent set (covariates) of variable $X$ under $G_s$ for sample $i$.

**Conditional Expectation Model Predictions for Exposure** $A$: The cross-fitted prediction for the conditional expectation of $A$ given its covariates $C_{A,i}$, obtained from a regression model (a multi-layer perceptron in SIGMA), is denoted as:

$$\hat{\mu}_A(C_{A,i}) \equiv \hat{E}[A_i | C_{A,i}].$$

**Global Estimate of Conditional Residual Variance of Exposure** $A$: SIGMA assumes homoscedasticity for $Var(A_i | C_{A,i})$, positing $Var(A_i | C_{A,i}) = \sigma^2_{A|C_A, res}$ for all $i$. This global residual variance, denoted $\hat{\sigma}^2_{A|C_A, res}$, is obtained by pooling the cross-fitted residuals $e_i = A_i - \hat{\mu}_A(C_{A,i})$ from all samples and calculating their sample variance (defined as $\frac{1}{n}\sum_{j=1}^{n}(e_j - \bar{e})^2$ or $\frac{1}{n-1}\sum_{j=1}^{n}(e_j - \bar{e})^2$, corresponding to the specific implementation).

**Conditional Expectation Model Predictions for Outcome** $Y$ **under Intervention on** $A$ **with Observed** $M, C_Y$: For a given intervention level $a$ on $A$ (e.g., $\tilde{\mu}_A$ or $\tilde{\mu}_A + k\tilde{\sigma}_A$ as defined in the main text), the cross-fitted prediction of the conditional expectation $\hat{E}[Y_i | A_i = a, M_i^{\text{obs}}, C_{Y,i}]$ is denoted as:

$$\hat{\mu}^{(i)}_{Y|A=a, M_{obs}, C_Y}.$$

**Individualized Plugin Predictions and Sample Means of Target Counterfactual Expectations**: For any counterfactual expectation composing NDE and NIE, such as $\mathbb{E}[Y(a', M(a''))]$ (denoted $\gamma_{a'a''}$):

The plugin prediction of the counterfactual outcome $Y_i(a', M_i(a''))$ for sample $i$ is denoted as $\hat{\gamma}_{a'a'',i}$. This prediction is obtained via G-computation, recursively combining the nuisance functions (1)-(3) above and outputs from the mediator model $\mathcal{M}_{M|A,C_M}$ (i.e., $\hat{E}[M|A = a'', C_M]$ or $\hat{P}(M = m|A = a'', C_M)$).And the sample mean of these individualized plugin predictions is denoted as $\hat{\gamma}_{a'a''} = \frac{1}{n}\sum_j \hat{\gamma}_{a'a'',j}$.

Here, $(a', a'')$ represent specific exposure intervention level combinations required for defining NDE or NIE.

Mathematical Definition of the Influence Function

Within SIGMA, the influence function score $\psi_{\gamma_{aa^*}}(Z_i)$ for estimating a target counterfactual expectation (e.g., $\gamma_{aa^*} = \mathbb{E}[Y(a, M(a^*))]$) is defined as (corresponding to Equation (6) in the main text):

$$\psi_{\gamma_{aa^*}}(Z_i; \hat{\eta}_s) = \frac{A_i - \hat{\mu}_A(C_{A,i})}{\hat{\sigma}^2_{A|C_A, res}}\left(\hat{\mu}^{(i)}_{Y|A=a, M_{obs}, C_Y} - \hat{\gamma}_{aa^*}\right) + \left(\hat{\gamma}_{aa^*,i} - \hat{\gamma}_{aa^*}\right). \tag{34}$$

This influence function comprises two main parts:

textbfFirst Part (Exposure Model Correction Term): $\frac{A_i - \hat{\mu}_A(C_{A,i})}{\hat{\sigma}^2_{A|C_A,res}} \left( \hat{\mu}^{(i)}_{Y|A=a,M_{obs},C_Y} - \hat{\gamma}_{aa^*} \right)$. The

term $\frac{A_i - \hat{\mu}_A(C_{A,i})}{\hat{\sigma}^2_{A|C_A,res}}$ is the score function of the log-conditional likelihood of $A|C_A$ with respect to its mean parameter $\hat{\mu}_A(C_{A,i})$, under the assumption that $A|C_A$ follows a Normal distribution $N(\hat{\mu}_A(C_{A,i}), \hat{\sigma}^2_{A|C_A,res})$. This score function is multiplied by a term related to the outcome model, $(\hat{\mu}^{(i)}_{Y|A=a,M_{obs},C_Y} - \hat{\gamma}_{aa^*})$, which reflects the deviation of the conditional outcome expectation (under intervened $A = a$, observed $M, C_Y$) from the mean target counterfactual expectation.

**Second Part (Centered Plugin Term)**: $(\hat{\gamma}_{aa^*,i} - \hat{\gamma}_{aa^*})$. This term is the centralization of the individualized plugin estimate $\hat{\gamma}_{aa^*,i}$ of the target parameter with respect to its sample mean $\hat{\gamma}_{aa^*}$.

The overall influence functions for NDE and NIE are then derived from linear combinations of these influence function scores for their constituent counterfactual expectations (i.e., $\gamma_{10}, \gamma_{00}, \gamma_{11}$, where the numerals 0 and 1 represent the reference level $a^*$ and target level $a$, respectively), according to the definitions of NDE and NIE (NDE $= \gamma_{10} - \gamma_{00}$, NIE $= \gamma_{11} - \gamma_{10}$). Specifically:

$$\psi_{\text{NDE}}(Z_i; \hat{\eta}_s) = \psi_{\gamma_{10}}(Z_i; \hat{\eta}_s) - \psi_{\gamma_{00}}(Z_i; \hat{\eta}_s) \tag{35}$$

$$\psi_{\text{NIE}}(Z_i; \hat{\eta}_s) = \psi_{\gamma_{11}}(Z_i; \hat{\eta}_s) - \psi_{\gamma_{10}}(Z_i; \hat{\eta}_s) \tag{36}$$

In these combinations, each $\psi_{\gamma_{a'a''}}(Z_i; \hat{\eta}_s)$ term on the right-hand side adopts the structure of Equation (34), utilizing the specific nuisance function components corresponding to its target counterfactual expectation ($\gamma_{10}, \gamma_{00}$, or $\gamma_{11}$).

Construction Rationale and Discussion

The influence function for continuous exposure defined within the SIGMA framework (Equation (34)) is motivated by the goal of providing an estimation method that offers first-order bias correction over simple plugin estimators while maintaining computational tractability. Its structure incorporates elements common in influence function theory, such as leveraging score function information from the exposure model and centering plugin estimates.

The definition of this influence function does not explicitly include direct correction terms related to the mediator model $P(M|A, C_M)$ beyond its implicit role in the G-computation of $\hat{\gamma}_{aa^*,i}$. In semiparametric theory, fully efficient influence functions for mediation effects often contain more complex terms explicitly dependent on the parameters or score functions of the mediator model(s) to ensure properties like double robustness against misspecification of either the outcome or mediator models. The influence function employed by SIGMA, while incorporating corrections related to the exposure model and centering the outcome predictions, may differ in its statistical properties, particularly double robustness and semiparametric efficiency, compared to such fully specified EIFs. Its performance is thus more reliant on the accurate estimation of all nuisance models, especially the conditional mean of the exposure $\hat{\mu}_A(C_A)$ and the conditional expectations related to the outcome $\hat{\mu}_{Y|A=a,M_{obs},C_Y}$. The adoption of a global homoscedastic approximation for the conditional variance of the exposure, $\hat{\sigma}^2_{A|C_A,res}$, is a choice made for implementational simplicity.

Despite these specific implementational choices, by incorporating corrections related to the exposure model, this influence function is expected to offer more robust performance in finite samples compared to uncorrected plugin estimators, particularly when the exposure model is well-estimated. A detailed theoretical analysis of this influence function under various data-generating mechanisms and an in-depth study of its finite-sample behavior constitute important directions for future research.

## B  VAE-based Imputation Module

To address missing data, SG-RMA incorporates a Variational Autoencoder (VAE) specifically adapted for imputing mixed-type tabular data. Formally, we consider the raw data matrix $X \in \mathbb{R}^{N \times P}$, which may contain missing entries, along with a binary mask matrix $M \in \{0, 1\}^{N \times P}$ indicating observed ($M_{ij} = 0$) or missing ($M_{ij} = 1$) values.

**Data Preprocessing.**  Raw features are preprocessed according to their data types: continuous variables undergo standardization, whereas categorical variables are transformed via one-hot encoding,

expanding the feature dimensionality from $P$ to $P'$. The resulting processed dataset is denoted as $X' \in \mathbb{R}^{N \times P'}$. Missing entries within $X'$ are initially filled with zeros to facilitate VAE training, and the corresponding mask $M$ is concatenated to guide the imputation process explicitly.

**VAE Model Specification.** The VAE consists of an encoder network $q_\phi(z|X', M)$ and a decoder network $p_\theta(X'|z)$. Given an input vector $(X'_i, M_i)$, the encoder produces parameters of a Gaussian posterior distribution in the latent space: mean vector $\mu_{z_i}$ and variance vector $\sigma^2_{z_i}$. The latent representation $z_i$ is then sampled via the reparameterization trick $z_i = \mu_{z_i} + \sigma_{z_i} \odot \epsilon$, where $\epsilon \sim \mathcal{N}(0, I)$. The decoder reconstructs data from latent representations, employing distinct output layers for each original feature type: linear layers for continuous features, logistic sigmoid activation for binary features, and softmax logits for categorical features.

The VAE optimization objective maximizes the evidence lower bound (ELBO), defined explicitly for observed entries as:

$$\mathcal{L}_{\text{VAE}}(\phi, \theta; x'_i, m_i) = \mathbb{E}_{q_\phi(z_i|x'_i, m_i)}[\log p_\theta(x'_{i,\text{obs}}|z_i)] - \beta D_{\text{KL}}(q_\phi(z_i|x'_i, m_i) \,\|\, p(z)), \qquad (37)$$

where $x'_{i,\text{obs}}$ denotes the observed elements of $x'_i$. The reconstruction likelihood $\log p_\theta(x'_{i,\text{obs}}|z_i)$ aggregates squared errors for continuous features, binary cross-entropy losses for binary features, and categorical cross-entropy losses for categorical features. The prior distribution $p(z)$ is standard normal, and $\beta$ is a hyperparameter balancing reconstruction quality and latent regularization. Training employs the AdamW optimizer, adaptive learning rate schedules, and early stopping criteria based on validation performance.

**Imputation Procedure.** After model training, missing data imputation proceeds as follows: For each sample $i$, the trained encoder maps the filled input $(X'_{\text{filled},i}, M_i)$ to the posterior mean latent representation $\mu_{z_i}$. Passing this latent vector through the decoder yields reconstructed outputs $\hat{x}'_i$. Reconstruction outputs are then transformed back into the original data space, applying inverse scaling to continuous variables, thresholding via logistic sigmoid for binary variables, and category assignment via argmax operation for categorical variables. Finally, missing entries within the original dataset $X$ are replaced by their respective reconstructions, while observed values remain unchanged. The imputation procedure is summarized in Algorithm 1.

## B.1 Path Stability Score: Theoretical Properties

### B.1.1 PSS as Posterior Probability Estimator

Section 4.3 of the main text introduces the Path Stability Score (PSS($\pi$)) as a measure to quantify the evidential support for a mediation pathway $\pi$. It is defined as $\text{PSS}(\pi) = \frac{1}{N_{\text{valid}}} \sum_{s=1}^{N_{\text{valid}}} \mathbb{I}(\pi \subseteq G_s)$, where $\{G_s\}_{s=1}^{N_{\text{valid}}}$ is an ensemble of $N_{\text{valid}}$ valid Directed Acyclic Graphs (DAGs) sampled based on the output of the Flow-SEM stage (Section 4.2.3). This section provides a rigorous justification for the interpretation of PSS($\pi$) as a Monte Carlo estimate of $P(\pi \subseteq G|D)$, the posterior probability that path $\pi$ is present in a DAG $G$ sampled from the (approximate) posterior distribution over graph structures learned by SIGMA given data $D$.

Let $P(G|D)$ denote this SIGMA-approximated posterior distribution over the space of DAGs $\mathcal{G}$, derived from the Flow-SEM model and the subsequent DAG sampling mechanism described in Section 4.2.3. The ensemble $\{G_s\}_{s=1}^{N_{\text{valid}}}$ consists of $N_{\text{valid}}$ valid DAGs treated as independent and identically distributed (i.i.d.) samples from $P(G|D)$.

Consider a specific path $\pi$. We define an indicator random variable $X_\pi(G) = \mathbb{I}(\pi \subseteq G)$, which equals 1 if path $\pi$ is a subgraph of $G$, and 0 otherwise. The posterior probability of path $\pi$ under $P(G|D)$ is the expectation of this indicator variable:

$$P(\pi \subseteq G|D) \equiv \mathbb{E}_{G \sim P(G|D)}[X_\pi(G)]. \qquad (38)$$

The Path Stability Score, $\text{PSS}(\pi) = \frac{1}{N_{\text{valid}}} \sum_{s=1}^{N_{\text{valid}}} X_\pi(G_s)$, is the sample mean of $X_\pi(G)$ based on these $N_{\text{valid}}$ i.i.d. samples.

**Theorem 1** (Statistical Properties of PSS). *The Path Stability Score PSS($\pi$) is:*

1. *An unbiased estimator of $P(\pi \subseteq G|D)$, i.e., $\mathbb{E}[PSS(\pi)] = P(\pi \subseteq G|D)$.*

---

**Algorithm 1** VAE-based Missing Value Imputation for Mixed-Type Data

---

**Require:** Trained VAE (encoder $q_\phi$, decoder $p_\theta$), Dataset $X \in \mathbb{R}^{N \times P}$ with missing values, Feature types $\{T_j\}_{j=1}^{P}$ where $T_j \in \{\text{continuous, binary, categorical}\}$, Preprocessing parameters (scaling factors $\{\mu_j, \sigma_j\}$ for continuous features, one-hot encoders for categorical features).
**Ensure:** Fully imputed dataset $X_{\text{imputed}} \in \mathbb{R}^{N \times P}$.
 1: $M \leftarrow \text{CreateMissingMask}(X)$
 2: $X'_{\text{raw}} \leftarrow \text{PreprocessFeatures}(X, \{T_j\})$
 3: $X'_{\text{filled}} \leftarrow X'_{\text{raw}}$; Fill missing entries in $X'_{\text{filled}}$ (identified by $M$) with 0
 4: $X_{\text{imputed}} \leftarrow X$
 5: **for** $i = 1$ to $N$ in batches **do**
 6: $\quad (\mu_{z_i}, \log \sigma_{z_i}^2) \leftarrow \text{Encoder}_\phi((X'_{\text{filled}})_i, M_i)$
 7: $\quad z_i \leftarrow \mu_{z_i}$
 8: $\quad \hat{X}'_i \leftarrow \text{Decoder}_\theta(z_i)$
 9: $\quad$ Initialize $\hat{X}_i \in \mathbb{R}^P$
10: $\quad$ **for** $j = 1$ to $P$ **do**
11: $\quad\quad$ **if** $T_j = \text{continuous}$ **then**
12: $\quad\quad\quad \hat{X}_{ij} \leftarrow \hat{X}'_{ij} \cdot \sigma_j + \mu_j$
13: $\quad\quad$ **else if** $T_j = \text{binary}$ **then**
14: $\quad\quad\quad \hat{X}_{ij} \leftarrow \mathbf{1}[\text{sigmoid}(\hat{X}'_{ij}) > 0.5]$
15: $\quad\quad$ **else if** $T_j = \text{categorical with } K_j \text{ categories}$ **then**
16: $\quad\quad\quad$ Let $\{\hat{X}'_{i,j_k}\}_{k=1}^{K_j}$ be the logits for all categories of feature $j$
17: $\quad\quad\quad \hat{X}_{ij} \leftarrow \arg\max_{k \in \{1, \ldots, K_j\}} \text{softmax}(\{\hat{X}'_{i,j_k}\})_k$
18: $\quad\quad$ **end if**
19: $\quad\quad$ **if** $M_{ij} = 1$ **then**
20: $\quad\quad\quad (X_{\text{imputed}})_{ij} \leftarrow \hat{X}_{ij}$
21: $\quad\quad$ **end if**
22: $\quad$ **end for**
23: **end for**
24: **if** $\text{HasRemainingMissing}(X_{\text{imputed}})$ **then**
25: $\quad$ **warning** "Imputation resulted in remaining NaNs."
26: **end if**
27: **return** $X_{\text{imputed}}$

---

    *2. A consistent estimator of $P(\pi \subseteq G | D)$, i.e., $PSS(\pi) \xrightarrow{p} P(\pi \subseteq G | D)$ as $N_{valid} \to \infty$.*

*Proof. (1) Unbiasedness:* Let $\{G_s\}_{s=1}^{N_{\text{valid}}}$ be the set of $N_{\text{valid}}$ valid DAGs sampled i.i.d. from $P(G | D)$.

$$\mathbb{E}[\text{PSS}(\pi)] = \mathbb{E}\left[ \frac{1}{N_{\text{valid}}} \sum_{s=1}^{N_{\text{valid}}} X_\pi(G_s) \right]$$

$$= \frac{1}{N_{\text{valid}}} \sum_{s=1}^{N_{\text{valid}}} \mathbb{E}[X_\pi(G_s)] \quad \text{(by linearity of expectation)}.$$

Since each $G_s \sim P(G | D)$ (i.i.d.), $\mathbb{E}[X_\pi(G_s)] = \mathbb{E}_{G \sim P(G | D)}[X_\pi(G)] = P(\pi \subseteq G | D)$. Thus,

$$\mathbb{E}[\text{PSS}(\pi)] = \frac{1}{N_{\text{valid}}} \sum_{s=1}^{N_{\text{valid}}} P(\pi \subseteq G | D) = P(\pi \subseteq G | D).$$

*(2) Consistency:* The variables $X_\pi(G_s)$ are i.i.d. Bernoulli random variables with mean $p = P(\pi \subseteq G | D)$ and finite variance $p(1 - p)$. By the Weak Law of Large Numbers (and Strong Law for almost sure convergence), their sample mean $\text{PSS}(\pi)$ converges in probability to $p$ as $N_{\text{valid}} \to \infty$. $\qquad\square$

Therefore, $\text{PSS}(\pi)$ provides a statistically sound Monte Carlo approximation of the posterior probability of path $\pi$ under the posterior distribution $P(G | D)$ approximated by SIGMA. The accuracy of

this approximation depends on two primary factors: (i) the quality of $P(G|D)$ as determined by the Flow-SEM and the DAG sampling mechanism, and (ii) the number of valid DAGs $N_{\text{valid}}$.

### B.1.2 Asymptotic Properties of PSS: Convergence Analysis with Increasing Sample Size

Theorem 1 establishes that the Path Stability Score, $\text{PSS}(\pi)$, serves as an unbiased and consistent estimator for $p_\pi \equiv P(\pi \subseteq G|D)$. The consistency, derived from the Law of Large Numbers, ensures that with a sufficiently large ensemble of $N_{\text{valid}}$ valid DAGs, $\text{PSS}(\pi)$ converges to $p_\pi$. This section further elaborates on the asymptotic properties of $\text{PSS}(\pi)$ as $N_{\text{valid}}$ increases, particularly its rate of convergence and distributional characteristics, which are fundamental for understanding its reliability and for guiding the choice of $N_{\text{valid}}$.

**Asymptotic Normality via the Central Limit Theorem (CLT).** Since $\text{PSS}(\pi)$ is a sample mean of $N_{\text{valid}}$ independent and identically distributed (i.i.d.) Bernoulli random variables $X_\pi(G_s) = \mathbb{I}(\pi \subseteq G_s)$ (each with mean $p_\pi$ and finite variance $p_\pi(1 - p_\pi)$), the Central Limit Theorem (CLT) applies. The CLT states that for a sufficiently large $N_{\text{valid}}$:

$$\sqrt{N_{\text{valid}}}(\text{PSS}(\pi) - p_\pi) \xrightarrow{d} \mathcal{N}(0, p_\pi(1 - p_\pi)), \tag{39}$$

where $\xrightarrow{d}$ denotes convergence in distribution. Equivalently, for large $N_{\text{valid}}$, $\text{PSS}(\pi)$ itself is approximately normally distributed:

$$\text{PSS}(\pi) \sim_{\text{approx}} \mathcal{N}\left(p_\pi, \frac{p_\pi(1 - p_\pi)}{N_{\text{valid}}}\right). \tag{40}$$

**Variance Analysis and Rate of Convergence.** The exact variance of the $\text{PSS}(\pi)$ estimator is:

$$\text{Var}(\text{PSS}(\pi)) = \frac{p_\pi(1 - p_\pi)}{N_{\text{valid}}}. \tag{41}$$

This shows that the variance is inversely proportional to $N_{\text{valid}}$. The standard error of $\text{PSS}(\pi)$ is thus $\text{SE}(\text{PSS}(\pi)) = \sqrt{p_\pi(1 - p_\pi)/N_{\text{valid}}}$, decreasing at a rate of $O(1/\sqrt{N_{\text{valid}}})$. This implies that to halve the standard error, $N_{\text{valid}}$ must be quadrupled. The variance is maximized when $p_\pi = 0.5$, indicating that more samples are needed for precise estimation when the true path probability is near $0.5$.

**Confidence Interval Construction and Practical Considerations for $N_{\text{valid}}$.** The asymptotic normality (Equation 40) allows for the construction of approximate Wald-type confidence intervals for $p_\pi$:

$$\text{PSS}(\pi) \pm z_{1-\alpha/2}\sqrt{\frac{\text{PSS}(\pi)(1 - \text{PSS}(\pi))}{N_{\text{valid}}}}, \tag{42}$$

where $z_{1-\alpha/2}$ is the $(1 - \alpha/2)$-quantile of the standard normal distribution, and $\text{PSS}(\pi)$ is used to estimate $p_\pi$ in the standard error term. This interval provides a range of plausible values for the true posterior probability $p_\pi$. The reliability of this interval and the PSS estimate itself depends on $N_{\text{valid}}$. For probabilities $p_\pi$ very close to $0$ or $1$, fewer samples may suffice for a given precision.

**Connection to Bayesian Model Averaging (BMA).** The statistical properties of PSS, particularly its interpretation as a posterior probability estimate, directly support the subsequent Bayesian Model Averaging (BMA) stage of SIGMA (Section 4.4.3). By using a threshold $\tau$ on $\text{PSS}(\pi)$ to identify stable paths, SIGMA focuses computational resources on pathways with substantial evidential support under $P(G|D)$. The ensemble $\{G_s\}_{s=1}^{N_{\text{valid}}}$ used to compute PSS is the same basis for BMA, ensuring a coherent propagation of structural uncertainty. For a stable path $\pi$, the BMA estimate $\hat{\theta}_{\text{BMA}}(\pi)$ and its variance $\hat{V}(\hat{\theta}_{\text{BMA}}(\pi))$ represent the model-averaged effect and its associated uncertainty. This averaging is performed over the plausible graph structures within the ensemble that support path $\pi$, effectively integrating over the structural uncertainty concerning these specific pathways as captured by $P(G|D)$.

## B.2 Discussion on path stability score threshold

The Path Stability Score (PSS) is defined as the frequency of path $\pi$ in the posterior DAG sample set:

$$\text{PSS}(\pi) = \frac{1}{N} \sum_{s=1}^{N} \mathbb{I}(\pi \subseteq G_s),$$

where $\pi$ represents the candidate mediational path, $G_s$ is the $s$-th DAG sample drawn from the structural posterior distribution, $N$ is the total number of DAG samples, and $\mathbb{I}(\cdot)$ is the indicator function. Given a threshold $\tau \in [0, 1]$ for the path stability score, we define the stable path set as:

$$\Pi_{\text{stable}}(\tau) = \{\pi : \text{PSS}(\pi) \geq \tau\}.$$

The selection of threshold $\tau$ directly determines the size and composition of $\Pi_{\text{stable}}$, thereby further influencing the Bayesian Model Averaging (BMA) estimate of path-specific effects:

$$\hat{\theta}_{\text{BMA}}(\pi|\tau) = \frac{1}{|\mathcal{G}_\pi(\tau)|} \sum_{G_s \in \mathcal{G}_\pi(\tau)} \hat{\theta}_s(\pi),$$

where $\mathcal{G}_\pi(\tau) = \{G_s : \pi \subseteq G_s, \text{PSS}(\pi) \geq \tau\}$ represents the subset of DAGs containing path $\pi$ given threshold $\tau$, and $\hat{\theta}_s(\pi)$ is the mediation effect estimated based on the DAG sample $G_s$.

A smaller value of $\tau$ may lead to the inclusion of more low-frequency paths, increasing the risk of false path discoveries (false positives); a larger value of $\tau$ may exclude paths with real effects but weaker statistical support (increasing false negatives). Therefore, in practice, the recommended strategy is to select a moderate intermediate value (e.g., $\tau = 0.15$), and then make fine adjustments based on the specific characteristics of the data and prior knowledge in the relevant domain. We explicitly adopted this strategy in the numerical experiments and practical algorithm implementation in this paper, and observed that this approach can achieve a reasonable balance between robustness and sensitivity in path selection.

## C Related Work

**Causal Structure Learning.** Differentiable neural approaches have advanced causal DAG discovery by casting it as continuous optimization. Zheng et al. (2018) introduced the seminal NOTEARS [72] method, which enforces an analytic acyclicity constraint to learn DAGs via standard gradient-based solvers. This breakthrough spurred many extensions to capture nonlinear relations and improve scalability. For example, GraN-DAG [35] extended NOTEARS to nonlinear settings by using feedforward networks to model causal mechanisms, and DAG-GNN [69] embedded a variational autoencoder with a graph neural network to learn DAG structures from data. Subsequent works proposed alternative differentiable formulations of the acyclicity constraint for stability and speed. ENCO [38] optimizes independent edge likelihoods without explicit acyclicity constraints, while Nazaret et al. introduce a spectral constraint in a Stable DCD [44] approach to improve numerical stability and scale causal discovery to thousands of variables .

Reinforcement learning (RL) has also been applied to structure learning – e.g., Zhu and Chen [75] use an RL agent to sequentially build the DAG, avoiding expensive combinatorial searches. Beyond point estimation of a single graph, recent variational methods estimate a distribution over plausible DAGs. For instance, BCD-Nets [16] perform Bayesian causal discovery via variational inference, yielding a posterior over DAG structures and quantifying uncertainty. Similarly, Differentiable Causal Discovery with Interventions (**DCDI**) [12] integrates normalizing flows to model complex causal mechanisms under interventions.

**Mediation Analysis.** Deep learning has also been applied to mediation analysis to estimate path-specific effects (natural direct and indirect effects) in complex settings. DeepMed [65] is a recent example that uses deep neural networks to flexibly estimate the necessary nuisance functions in the efficient influence function for mediation, achieving semiparametric-optimal estimation of direct and indirect effects. By cross-fitting DNNs for propensity and outcome models, DeepMed debiases mediator and outcome predictions and attains the efficiency bound without restrictive model assumptions. Other works leverage deep latent-variable models to handle mediators and hidden confounding.

CMA-VAE [13] introduced a variational autoencoder for causal mediation that accounts for latent confounders, enabling estimation of natural direct and indirect effects even with unobserved confounders. Building on this, Disentangled Mediation Analysis VAE (DMAVAE) [66] separates latent factors into mediating, confounding, and irrelevant components. This disentanglement allows DMAVAE to consistently estimate direct and indirect effects under weaker assumptions than traditional sequential ignorability, significantly improving mediation effect estimation in the presence of multiple types of latent confounders.

In parallel, deep models have been used for path-specific effect estimation in high-dimensional settings like text and fairness. CausalBERT [63] fine-tunes transformer language models to learn low-dimensional text embeddings that retain information needed for causal adjustment. This approach enables estimation of causal effects from text (e.g., the effect of a review's content on outcomes) by adjusting for textual confounders within the embedding space. Similarly, in algorithmic fairness, researchers have examined path-specific effects via deep nets to detect mediated discrimination. For example, Chiappa [14] studies counterfactual fairness by removing information about protected attributes along certain causal paths, and Nabi & Shpitser [43] develop methods to identify path-specific effects under mediation and confounding. These efforts underscore a growing theme: integrating deep representation learning with causal mediation analysis to uncover nuanced causal pathways (e.g., via stability analysis or regularization to select stable mediators) in complex, high-dimensional data.

**Deep Treatment Effect Estimation.** Estimating individualized treatment effects (ITEs) or heterogeneous causal effects has benefited greatly from deep learning, particularly through representation learning and efficient influence function (EIF) techniques. A landmark work by Johansson et al. [31] introduced learning balanced representations for treatment and control groups to improve ITE generalization. This approach, extended by Shalit et al. [53] with integral probability metric regularization, underlies methods like TARNet/CFR that reduce selection bias by aligning latent representations across treatment groups. Subsequent methods added adversarial objectives to enforce balance: Yao et al. [67] propose a SITE approach that preserves local similarity while using adversarial training to minimize covariate distribution differences, and GANITE [68] directly employs GANs to generate counterfactual outcomes for ITE estimation.

Another line of work incorporates semiparametric theory to improve estimation efficiency and robustness. DragonNet [54] is a neural architecture that adapts the network training to target the efficient influence function of the average treatment effect. By jointly learning the outcome and propensity predictor with a targeted regularization term, DragonNet yields doubly-robust estimates that empirically outperform earlier architectures. In a related vein, CEVAE [39] uses deep variational inference to handle unobserved confounders, learning a latent variable model that estimates treatment effects under a causal graphical model assumption. Extensions of CEVAE have further improved disentanglement; for instance, TEDVAE [70] factorizes latent confounders into multiple parts to better separate causal effects.

To inject inductive biases about potential outcomes, FlexTENet [17] adaptively learns what representations to share between the treated and control outcome functions, reflecting the assumption that many predictive factors are common between potential outcomes. Beyond standard observational studies, deep models have tackled more complex causal inference tasks: DeepIV [25] uses deep nets for instrumental variable regression to estimate counterfactual predictions in the presence of hidden confounding, and other works address time-series and panel data with recurrent networks for sequential treatment effects [5].

**Causal Inference under Structural Uncertainty** A distinguishing challenge for methods like SIGMA is **causal inference under structural uncertainty**, where the true causal graph is unknown or ambiguous. Recent research has begun to explicitly account for this uncertainty using Bayesian deep learning, model averaging, and ensembles of causal models. One approach is to perform Bayesian model averaging (BMA) over DAGs: rather than committing to a single estimated graph, one can weight effect estimates by each graph's posterior probability. Cundy et al. [16] implement this via their variational BCD-Nets, which output a distribution over possible DAG structures. By sampling DAGs from the learned posterior, BCD-Nets can compute expectation of causal effects across many graph realizations, inherently providing credible intervals for effects. DECI [22], a deep end-to-end causal inference framework, uses normalizing flows within a variational Bayesian scheme.

DECI simultaneously learns a DAG posterior and estimates causal quantities, essentially performing causal discovery and inference in one unified model.

Furthermore, techniques like stability selection have been adapted to causal structure learning to identify consensus edges under data perturbations, thereby informing model averaging strategies (e.g., selecting only edges that consistently appear across high-scoring DAGs) [44]. Such ideas align with the stability analysis component of SIGMA, which seeks robust mediation pathways by exploring variations in the learned structure. Overall, the literature suggests that accounting for structural uncertainty – through Bayesian posterior averaging, DAG ensembles, or stability-based model selection – can substantially improve the reliability of causal effect estimates in complex models. SIGMA builds on these insights by integrating BMA of mediation pathways with EIF-based effect estimation, ensuring that uncertainty in the causal graph is properly propagated into uncertainty in estimated direct and indirect effects.

# D   Human Phenotype Project Dataset

## D.1   Description of HPP Cohort

The Human Phenotype Project (HPP) is a deeply phenotyped, longitudinal cohort study conducted in Israel, designed to investigate the interplay between various physiological systems and environmental exposures. The present study analyzed data collected from 6,748 participants aged 40–75 years between January 2019 and December 2022. All participants provided written informed consent, and the study protocol was approved by the institutional review board of the Weizmann Institute of Science (IRB no. 1719-1). Participants underwent comprehensive profiling, including clinical, physiological, behavioral, and multi-omics assessments, which were categorized into 17 body systems [34], one of which focused on sleep. Sleep characteristics were derived from home sleep apnea testing (HSAT) using the WatchPAT 300 device (ZOLL Itamar), with each participant completing up to two monitoring series, each comprising three nights of testing over a two-week period. A total of 16,812 nights of valid HSAT recordings from 6,366 individuals (47.8% male, 52.2% female; mean age 52.4 $\pm$ 7.7 years; mean BMI 26.1 $\pm$ 4.1 kg/m$^2$) were included in the final analysis.

The cohort primarily consisted of healthy, educated individuals of European (Ashkenazi) Jewish ancestry residing in a relatively homogenous environmental context. Exclusion criteria included severe medical conditions at enrollment. Participants were scheduled for biennial follow-up assessments over a 25-year period. This extensive data resource enables a high-resolution investigation of associations between sleep and systemic health.

## D.2   Body System-Derived Features

We utilize data from 11 body systems, comprising over 500 raw clinical and physiological attributes. To facilitate tractable and clinically meaningful mediation analysis, we derived a set of 62 aggregated features by domain-informed grouping of related variables across systems. These features are constructed to capture coherent physiological constructs suitable for pathway-level causal interpretation. Please see Table 3 for the detailed description of our aggregated features.

**Sleep Characteristic**   Sleep characteristics were captured using the WatchPAT 300 home sleep monitoring device across up to three nights per participant. A total of 448 features were extracted, comprising two main subgroups: 100 sleep test measurements and 348 pulse rate variability (PRV) features. Sleep test features included metrics related to respiratory events (e.g., peripheral apnea–hypopnea index [pAHI]), oxygen desaturation (e.g., mean SpO$_2$ nadir, time below 90% saturation), snoring intensity, sleep stage distribution (light, deep, REM), sleep efficiency, and body position during sleep. PRV features were derived from the peripheral arterial tonometry signal using the NeuroKit2 library [42], spanning time- and frequency-domain indices, entropy, and nonlinear dynamics, computed across REM, NREM, wake, and full-night segments.

**Metabolic Pathways**   Metabolic pathway features were derived from metagenomic profiling of the gut microbiome using the HUMAnN3 [4] functional annotation pipeline. Relative abundances of microbial metabolic pathways were quantified based on high-throughput sequencing data, followed by taxonomic and functional mapping against a curated reference database. Only pathways present in

Table 3: Summary of physiological systems, features and medical explanations

| No. | System | Features (V0 → V61) | Medical Explanation |
|---|---|---|---|
| 1 | Body Fat Composition | body fat percentage | high values are linked to obesity and metabolic risk |
| | | ag fat ratio | Reflects android-to-gynoid fat distribution; a higher ratio suggests central obesity and cardiovascular risk |
| | | visceral fat volume | Quantifies visceral fat volume around organs; associated with insulin resistance and inflammation. |
| | | appendicular lean mass | Measures limb muscle mass; a key indicator for sarcopenia and frailty. |
| 2 | OSA-Related Phenotypes | osa severity | Indicates severity of obstructive sleep apnea (OSA); high levels suggest moderate to severe OSA. |
| | | snoring severity | Key indicator of OSA severity, based on snoring loudness and frequency. |
| | | snoring ratio | Proportion of snoring time during total sleep time; linked to OSA risk. |
| 3 | Sleep Structure Quality | sleep duration | Total sleep length; short duration is associated with various disorders. |
| | | sleep efficiency | Ratio of actual sleep time to time in bed; reflects sleep maintenance quality. |
| | | sleep continuity | Measures fragmentation; lower continuity indicates more fragmented sleep. |
| | | restorative sleep ratio | Ratio of restorative sleep stages to total sleep; reflects sleep quality. |
| 4 | Sleep HRV | night mean HR | Average nighttime heart rate; reflects autonomic balance. |
| | | hrv index | HRV index during sleep; lower HRV linked to stress and metabolic issues. |
| | | hrv stage variation | HRV variability across sleep stages; reflects sleep structure quality. |
| | | hrv wake delta | HRV difference between sleep and wake; indicates recovery capacity. |
| 5 | Nocturnal Hypoxia Burden | oxygen desaturation | Measures oxygen drop severity during sleep; indicates hypoxia level. |
| | | oxygen burden | Cumulative oxygen burden from all events; reflects physiological load. |
| 6 | Blood Pressure Lying | lying diastolic pressure | Diastolic blood pressure measured while lying. |
| | | lying pulse rate | Pulse rate measured while lying. |
| | | lying systolic pressure | Systolic blood pressure measured while lying. |
| 7 | Blood Pressure Sitting | sitting diastolic pressure | Diastolic blood pressure measured while sitting. |
| | | sitting pulse rate | Pulse rate measured while sitting. |
| | | sitting systolic pressure | Systolic blood pressure measured while sitting. |
| 8 | Blood Pressure Standing | standing 1min diastolic pressure | Diastolic blood pressure measured after standing for 1 minute. |
| | | standing 1min pulse rate | Pulse rate measured after standing for 1 minute. |
| | | standing 1min systolic pressure | Systolic blood pressure measured after standing for 1 minute. |
| | | standing 3min diastolic pressure | Diastolic blood pressure measured after standing for 3 minutes. |
| | | standing 3min pulse rate | Pulse rate measured after standing for 3 minutes. |
| | | standing 3min systolic pressure | Systolic blood pressure measured after standing for 3 minutes. |
| 9 | Blood Pressure Resting | resting systolic pressure | Systolic blood pressure at rest. |
| | | resting diastolic pressure | Diastolic blood pressure at rest. |
| | | resting pulse rate | Pulse rate at rest. |
| 10 | Blood Pressure Orthostatic | orthostatic SBP drop 1min | Drop in systolic blood pressure 1 minute after standing. |
| | | orthostatic DBP change 1min | Change in diastolic blood pressure 1 minute after standing. |
| | | orthostatic SBP drop 3min | Drop in systolic blood pressure 3 minutes after standing. |
| | | orthostatic DBP change 3min | Change in diastolic blood pressure 3 minutes after standing. |
| 11 | Blood Pressure Pulse | pulse rate increase 1min | Increase in pulse rate 1 minute after standing. |
| | | pulse rate increase 3min | Increase in pulse rate 3 minutes after standing. |
| 12 | Vascular Health | ABI min | Ankle-brachial index; values <0.9 suggest peripheral artery disease. |
| | | PWV mean | Pulse wave velocity; indicates arterial stiffness and cardiovascular risk. |
| | | SBP max | Maximum systolic blood pressure; indicates peak arterial load. |
| 13 | Carotid Ultrasound IMT | imt left | Intima-media thickness (IMT) of left carotid artery. |
| | | imt right | Intima-media thickness (IMT) of right carotid artery. |
| | | imt window width left | Window width of IMT measurement on the left side. |
| | | imt window width right | Window width of IMT measurement on the right side. |
| | | imt fit left | IMT fit value on the left side. |
| | | imt fit right | IMT fit value on the right side. |
| | | mean cimt | Average carotid IMT across measured regions. |
| | | max cimt | Maximum carotid IMT observed. |
| | | abnormal cimt | Indicates presence of abnormal IMT exceeding clinical thresholds. |
| 14 | Carotid Ultrasound Plaque | plaque presence by fit | Indicates presence of plaque based on IMT fit criteria. |
| | | plaque presence by thickness | Indicates presence of plaque based on IMT thickness threshold. |
| 15 | Gut Microbiome Abundance | butyrate producers abundance | Abundance of butyrate-producing bacteria; linked to anti-inflammatory effects and gut barrier integrity. |
| | | propionate producers abundance | Propionate-producing bacteria; involved in appetite regulation and metabolism. |
| | | probiotic fermenters abundance | Probiotic fermenting bacteria; produce lactic acid and other beneficial metabolites. |
| | | vitamin producers abundance | Bacteria capable of synthesizing vitamins such as B vitamins. |
| | | akkermansia abundance | Abundance of *Akkermansia*, associated with metabolic health. |
| | | LPS producers abundance | Bacteria producing lipopolysaccharides (LPS); may promote inflammation. |
| | | sulfate reducers abundance | Sulfate-reducing bacteria; potentially harmful. |
| | | TMA producers abundance | Bacteria producing TMA (precursor of TMAO); linked to cardiovascular disease. |
| | | proteolytic bacteria abundance | Bacteria that degrade proteins; involved in production of toxic metabolites. |
| | | shannon diversity index | Shannon index; measures richness and evenness of gut microbes — a key marker of gut health. |

at least 5% of participants were retained for analysis, and abundance values were $\log_{10}$-transformed to stabilize variance. These pathways represent a functional summary of microbial activity, covering key domains such as amino acid biosynthesis, vitamin and cofactor metabolism, carbohydrate degradation, and short-chain fatty acid production. They were used to probe host–microbiome interactions relevant to systemic metabolic health and potential associations with sleep-disordered breathing phenotypes.

**Cardiovascular System** Cardiovascular features encompassed a comprehensive set of hemodynamic and structural measurements obtained through non-invasive techniques. Peripheral and central

blood pressures were measured in sitting, supine, and standing positions. Arterial stiffness was evaluated using pulse wave velocity (PWV) recorded with the Falcon device (Viasonix), and endothelial function was indirectly assessed through carotid intima-media thickness using high-resolution ultrasonography (SuperSonic Aixplorer MACH 30). Retinal microvascular parameters were extracted via fundus imaging (iCare DRSplus) and processed with the AutoMorph [74] Python package. Cardiac electrical activity was characterized by 12-lead resting ECG (PC-ECG 1200, NORAV), from which standard and derived indices (e.g., heart rate, QT intervals) were calculated. The cardiovascular dataset thus integrates structural, functional, and electrophysiological dimensions of cardiovascular health, enabling multiscale association testing with sleep traits.

**Inflammation**    The inflammation system features were derived from standard clinical laboratory tests obtained from participants' health maintenance organizations (HMOs) as part of their routine medical care. Participants were encouraged to upload results from various tests, including those indicative of systemic inflammation. Key biomarkers relevant to the inflammation system included C-reactive protein (CRP), white blood cell (WBC) count, and differential counts such as neutrophils, lymphocytes, monocytes, eosinophils, and basophils. These markers provide insights into both acute and chronic inflammatory states. CRP, an acute-phase protein synthesized by the liver, serves as a sensitive indicator of systemic inflammation and is commonly elevated in response to infection or tissue injury. The WBC count and its differentials offer information on the immune system's cellular components, with variations potentially reflecting underlying inflammatory or immune responses. All laboratory values were standardized and stored in a structured format to facilitate downstream analyses.

**Baseline Characteristics (BMI)**    Body mass index (BMI) was calculated as weight in kilograms divided by height in meters squared ($kg/m^2$) and was treated as a baseline covariate alongside age and sex. Though BMI is not classified as a standalone body system in this analysis, it serves as a critical covariate due to its known influence on both sleep-disordered breathing and cardiometabolic risk. It was excluded from the body composition system to avoid redundancy in statistical modeling.

# E    Synthetic Datasets Details

## E.1    Generation Process

We construct synthetic datasets using a modular simulation framework that mimics realistic causal systems with configurable complexity. The generation process is parameterized to control graph structure, variable types, functional complexity, effect magnitudes, missing data patterns, and more, making the synthetic data suitable for benchmarking causal discovery and mediation analysis algorithms. The entire dataset generation procedure is outlined in Algorithm 2.

**Causal Graph Construction**    Each dataset is generated from a DAG $G$ over $p$ variables and contains $n$ observational samples. The graph is sampled from an **Erdős-Rényi (ER)** model with a specified expected average in-degree $k$, followed by a random topological sort to ensure acyclicity. The DAG defines the full set of causal relationships among the $p$ variables, with no predefined roles assigned to individual variables. To support the construction of realistic and diverse causal structures, we explicitly control two key properties of the graph: the chain path length $\ell$, which determines the depth of the longest directed chain in the DAG, and the number of structurally parallel paths $\kappa$, which specifies how many disjoint or partially overlapping causal routes exist across the graph. This parameterization allows us to generate graphs that reflect varying levels of structural complexity, enabling robust evaluation of causal discovery and inference methods under different topological regimes.

**Structural Equation Modeling**    Let $\mathbf{v} = (v_1, \cdots, v_p)$ be a collection of random variables. Each node is randomly assigned a variable type according to a multinomial distribution over three categories: continuous with probability $\pi_c$, binary with $\pi_b$, and categorical (with up to $L_{\text{cat}}$ levels) with probability $\pi_{\text{cat}}$, where $\pi_c + \pi_b + \pi_{\text{cat}} = 1$. Once the types are assigned, we generate variables according to a structural equation modeling (SEM) [46] framework based on a given DAG. Specifically, each variable $v_i$ is assigned a structural equation of the form:

$$v_i = F_i \left( \mathbf{v}_{\text{pa}(i;G)}, z_i \right), \tag{43}$$

where $z_i$ is an exogenous noise term independent of all other variables, $\mathrm{pa}(i; G)$ denotes the set of parent nodes of $i$ in the graph $G$, and $F_i$ is the deterministic function governing the influence of parent variables on $v_i$. In this work, we focus on a special class of SEMs known as additive noise models (ANMs), where the structural equations take the simplified form:

$$v_i = f_i\left(\mathbf{v}_{\mathrm{pa}(i;G)}\right) + z_i, \quad \text{or in vector form:} \quad \mathbf{v} = f_G(\mathbf{v}) + \mathbf{z}. \tag{44}$$

The functional form $f_i$ may be linear or nonlinear, including polynomial (e.g., $v^2$), sinusoidal (e.g., $\sin(v)$), or exponential (e.g., $\exp(-|v|)$) transformations. To control the proportion of nonlinear mechanisms, we define a nonlinearity ratio parameter $\rho_{\mathrm{nonlin}} \in [0, 1]$, which specifies the expected fraction of nodes governed by nonlinear functions. For nodes not selected as nonlinear, $f_i$ defaults to a linear combination of parent variables. This stochastic selection enables systematic benchmarking across datasets with different levels of structural complexity. Pairwise interactions between parent variables can also be incorporated to increase structural complexity. The exogenous noise $z_i$ is sampled from a mixture of distributions—Gaussian, exponential, or Student-$t$—with user-defined mixture weights to simulate heteroskedasticity and non-Gaussian variability. For discrete variables, logistic or softmax functions are used to generate probabilistic outcomes conditioned on parent variables.

**Causal Effect Scaling**    To enable controlled experimentation with mediation analysis, the generator includes an effect-scaling mechanism that adjusts the magnitude of causal effects along specific paths. Structural coefficients on the direct path from treatment to outcome, and on the indirect paths through mediators, are scaled such that the resulting absolute values of the natural direct effect (NDE) and natural indirect effect (NIE) fall within user-specified intervals, denoted by $[\tau_{\mathrm{NDE}}^{\min}, \tau_{\mathrm{NDE}}^{\max}]$ and $[\tau_{\mathrm{NIE}}^{\min}, \tau_{\mathrm{NIE}}^{\max}]$, respectively. This ensures that generated datasets reflect a wide spectrum of causal regimes, from dominantly direct to heavily mediated effects.

**Missingness Injection and Ground Truth Estimation**    To further enhance realism, the framework optionally injects missingness into the dataset under both missing completely at random (MCAR) and missing at random (MAR) mechanisms. The overall missingness rate $r_{miss}$ determines the proportion of missing entries. In addition, the framework allows configuration of the proportion of variables and the conditional dependencies under which missingness occurs. Ground-truth causal effects, including total effect (TE), NDE, and NIE, are estimated via Monte Carlo simulation using a large number of samples (e.g., 10,000 or more) by evaluating counterfactual outcomes under interventions on treatment and mediator variables.

---

**Algorithm 2** Synthetic Dataset Generation Procedure

---

1: **Require:** Number of variables $p$, samples $n$, average in-degree $k$
2:         Target effect ranges $\tau_{\mathrm{NDE}}, \tau_{\mathrm{NIE}}$
3:         Number of parallel paths in DAG: $\kappa$
4:         Length of the chain path in DAG: $\ell$
5:         Variable type proportions: $\pi_c, \pi_b, \pi_{\mathrm{cat}}$
6:         Nonlinearity proportion $\rho_{\mathrm{nonlin}}$
7:         Noise distribution mix $D_{\mathrm{noise}}$, missingness rate $r_{\mathrm{miss}}$
8: **Ensure:** Synthetic dataset $D$ with DAG $G$, SEM specs, and ground-truth effects
9: Generate DAG $G$ using Erdős–Rényi model with $p$ nodes and average in-degree $k$.
10: Verify that $G$ contains $\kappa$ parallel directed paths of length $\leq \ell$
11: Assign variable types to each $v_i \in \mathbf{v}$:
12:      Continuous with $\pi_c$; Binary with $\pi_b$; Categorical (up to $L_{\mathrm{cat}}$) with $\pi_{\mathrm{cat}}$.
13: **for** each node $v \in \mathbf{v}$ **do**
14:      Define: $v_i \leftarrow f_i\left(\mathbf{v}_{\mathrm{pa}(i;G)}\right) + z_i$;
15:      Choose $f_i$ as linear or nonlinear (e.g., poly2, sin, exp, interaction) with probability $\rho_{\mathrm{nonlin}}$;
16:      Sample noise $z_i \sim D_{\mathrm{noise}}$
17: **end for**
18: Generate $n$ samples by topological traversal of $G$
19: Inject missing values using $r_{\mathrm{miss}}$ and conditional rules (e.g., MAR/MCAR).
20: Estimate ground-truth effects (TE, NDE, NIE) via Monte Carlo simulation.
21: **return** $D$

---

### E.2 Mediation Pathway Structures

To evaluate mediation analysis methods under diverse structural assumptions, we incorporate a range of causal graphs with varying mediation configurations into the synthetic data generator. Figure 4 illustrates several representative DAG structures that are automatically instantiated within our framework. These examples cover key topologies relevant for causal mediation, including both simple and complex pathways, as well as potential nonlinear effects.

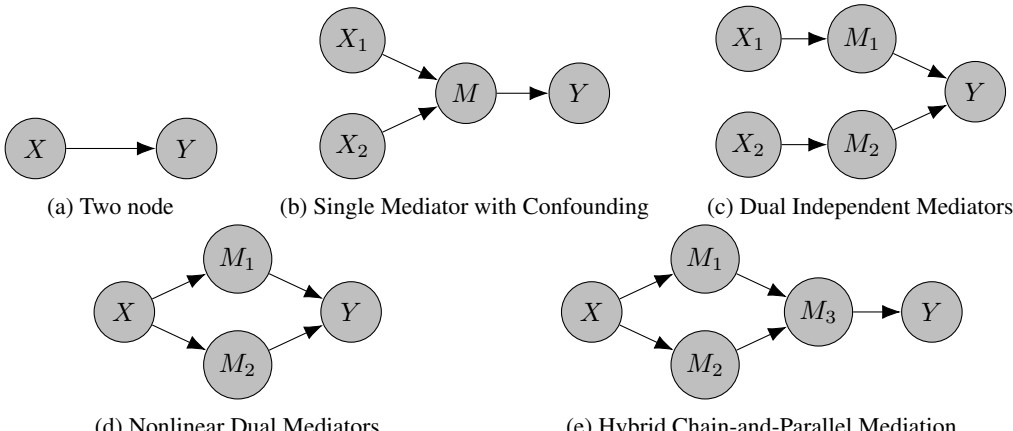

Figure 4: DAG graphs. Unless otherwise stated, we take $X$ as the treatment, $Y$ as the outcome, and $M$ as mediator variable.

1. **(a) Direct-only structure** (*"Two-Node Path"*): This minimal structure includes only a direct effect from treatment $X$ to outcome $Y$, with no mediators. It serves as a baseline for evaluating spurious mediation detection.

2. **(b) Single mediator with confounding inputs** (*"Single Mediator with Confounding"*): A single mediator $M$ lies between $X$ and $Y$, but additional variables (e.g., $X_1$, $X_2$) influence the mediator, simulating a confounded or high-dimensional mediator function.

3. **(c) Parallel mediation** (*"Dual Independent Mediators"*): Two mediators $M_1$ and $M_2$ are both affected by $X$ and influence $Y$ independently, forming parallel mediation paths.

4. **(d) Nonlinear parallel mediation** (*"Nonlinear Dual Mediators"*): Similar to (c), but the structural functions between nodes involve nonlinearities such as interactions or sine/exponential transforms.

5. **e) Mixed mediation** (*"Hybrid Chain-and-Parallel Mediation"*): A more complex topology where two mediators $M_1$ and $M_2$ form parallel channels, and a third node $M_3$ forms a chain extending toward $Y$. This structure tests the ability of algorithms to disentangle indirect effects across multiple pathways.

### E.3 Dataset Benchmarks

Table 4 summarizes the configurations used to generate representative datasets in our experiments. These settings allow us to benchmark the performance of causal inference methods across regimes with varying dimensionality, mediation complexity, and data quality. Specifically, we hold constant the overall variable type proportions ($\pi_c = 0.6$, $\pi_b = 0.3$, $\pi_{\text{cat}} = 0.1$), category levels ($L_{\text{cat}} = 5$), missingness ($r_{\text{miss}} = 0.1$), average in-degree ($k = 3$) and the target standard deviation ranges of the causal effects ($\tau_{\text{NDE}} \in [0.2, 0.4]$, $\tau_{\text{NIE}} \in [0.15, 0.3]$). This design allows for controlled benchmarking across varying causal and statistical challenges.

To further validate the quality of mediation structures embedded in the synthetic datasets, we visualize the variable relationships of sampled mediation pathways. Specifically, we examine the empirical dependencies among treatment, mediator, and outcome variables to see whether the intended causal effects are faithfully reflected in the generated data. We analyze four types of associations: (i) Treatment → Mediator, (ii) Mediator → Outcome, (iii) Treatment → Outcome, and (iv) Mediator

| Dataset ID | $p$ | $n$ | $\rho_{\text{nonlin}}$ | $\kappa$ | $\ell$ | **Configuration Focus** |
|---|---|---|---|---|---|---|
| LowDim-L | 20 | 6000 | 0 | 1 | 3 | Baseline (Low-dim, linear, simple graph) |
| LowDim-N | 20 | 6000 | 0.5 | 1 | 3 | Baseline (Low-dim, nonlinear, simple graph) |
| LowDim-P | 20 | 6000 | 0.5 | 2 | 2 | Complex path (Low-dim, parallel, nonlinear) |
| LowDim-D | 20 | 6000 | 0.5 | 1 | 6 | Complex path (Low-dim, long chain, nonlinear) |
| MidDim-S | 100 | 6000 | 0.5 | 1 | 3 | Mid-dim effect (long chain, nonlinear, simple graph) |
| MidDim-D | 100 | 6000 | 0.5 | 1 | 6 | Mid-dim effect (long chain, nonlinear, complex graph) |
| HigDim-S | 200 | 6000 | 0.5 | 1 | 3 | High-dim effect (nonlinear, simple grapgh) |
| HigDim-D | 200 | 6000 | 0.5 | 1 | 6 | High-dim effect. (long chain, nonlinear, upper bound) |
| MidDim-P | 100 | 6000 | 0.5 | 2 | 2 | Model misspecification test (nonlinear, parallel) |
| MidDim-C | 50 | 6000 | 0.5 | 1 | 6 | Mid-dim effect (nonlinear, long chain, smoothed comparison) |

Table 4: Parameter configurations for synthetic datasets used in our experiments.

$\rightarrow$ Outcome under different treatment. These visualizations provide intuitive verification that the synthetic data preserves the expected mediation mechanisms and highlight the nonlinear or conditional dependencies that may arise under the configured data-generating process.

**LowDim-L:** *LowDim-L* serves as the baseline configuration in the benchmark suite, featuring low dimensionality ($p = 20$), a fully linear structural specification ($\rho_{\text{nonlin}} = 0.0$), and a simple graph topology with a single short causal path ($\kappa = 1$, $\ell = 3$). The graph structure is minimal and interpretable, offering a clean testbed for evaluating algorithmic correctness in effect estimation under idealized, statistically well-behaved conditions. The causal graph consists of a binary treatment variable $V8$, a continuous mediator $V2$, a categorical mediator $V18$, and a binary outcome $V13$. This setup enables clean interpretability of causal effects and serves as a reference for assessing performance under idealized conditions. *LowDim-L* provides a controlled environment for validating the correctness of pathway identification and effect estimation algorithms under structurally simple and statistically well-behaved conditions.

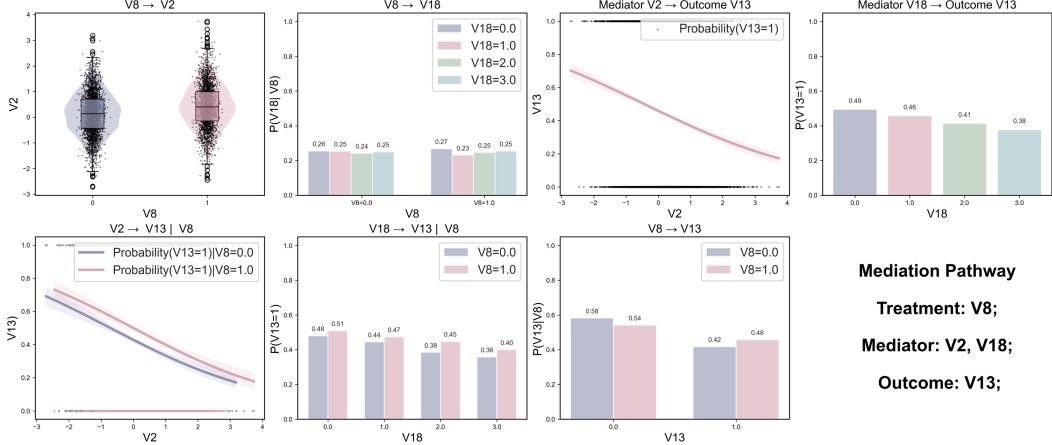

Figure 5: **Visualization of variable relationships of a sampled mediation pathway in LowDim-L**. The pathway comprises a binary treatment variable $V8$, a continuous mediator $V2$, a categorical mediator $V18$, and a binary outcome $V13$.

**LowDim-N:** *LowDim-N* mirrors the low-dimensional structure of *LowDim-L* ($p = 20, n = 6000$), but introduces nonlinear functional mechanisms ($\rho_{\text{nonlin}} = 0.5$) while maintaining a structurally simple graph with one short directed path ($\kappa = 1, \ell = 3$). This setting incorporates nonlinear mappings within the graph's structural equations, allowing us to evaluate whether causal inference methods remain robust to nonlinearity. The variable types are consistent: treatment $V8$ is binary, mediators $V2$ and $V18$ are continuous and categorical respectively, and outcome $V13$ is binary. In contrast to *LowDim-L*, the functional dependencies here include non-additive effects and interactions, enabling the examination of empirical recoverability under nonlinear data-generating processes. In particular, the direction of the mediator–outcome relationship is reversed: $V2$ now has a positive rather than negative effect on $V13$, consistent with the design of its generating function. Conditional plots further highlight interactions modulated by treatment levels, providing visual confirmation of nonlinear mediation behavior.

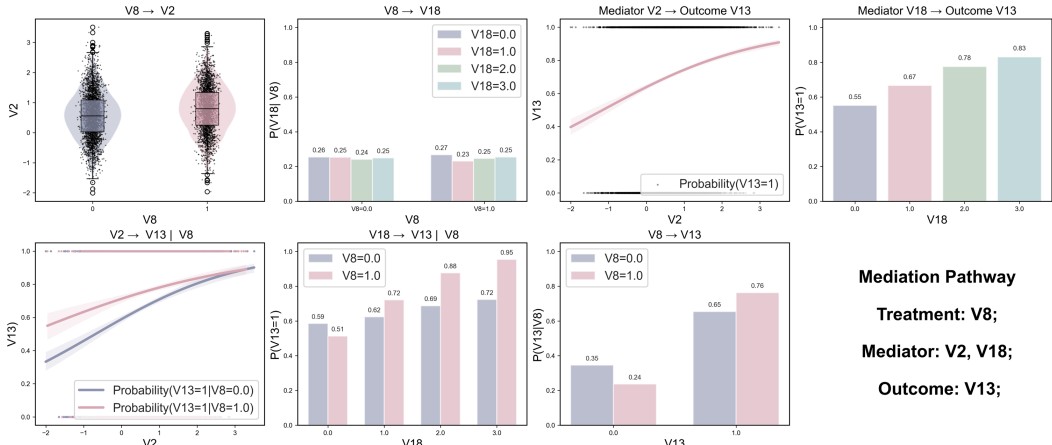

Figure 6: **Visualization of variable relationships of a sampled mediation pathway in LowDim-N**. The pathway comprises a binary treatment variable $V8$, a continuous mediator $V2$, a categorical mediator $V18$, and a binary outcome $V13$.

**LowDim-P:** *LowDim-P* is designed to introduce structural complexity through multiple causal pathways, while maintaining low dimensionality ($p = 20, n = 6000$) and moderate nonlinearity ($\rho_{\text{nonlin}} = 0.5$). The graph contains $\kappa = 2$ parallel paths of moderate depth ($\ell = 3$), representing concurrent channels of influence. The variable configuration includes a binary treatment V8, continuous mediator V2, categorical mediator V18, binary mediator V7, and binary outcome V13. This setup is designed to evaluate the interaction and combined influence of multiple concurrent mediators under nonlinear conditions.

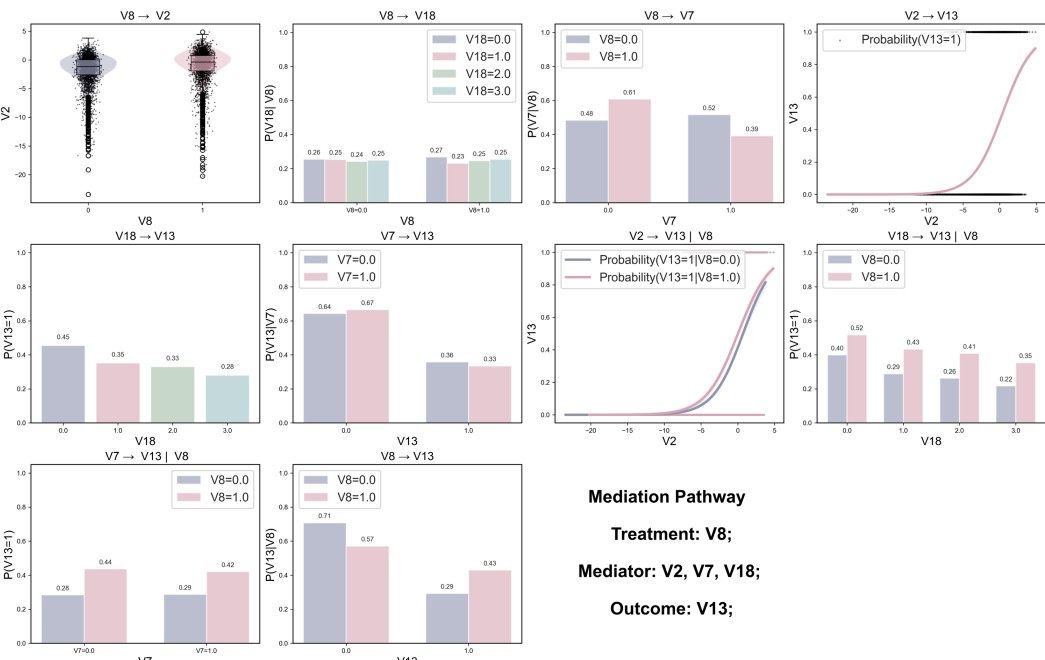

Figure 7: **Visualization of variable relationships of a sampled mediation pathway in LowDim-P.** The pathway comprises binary treatment variable $V8$, a continuous mediator $V2$, a categorical mediator $V18$, a binary mediator $V7$ and a binary outcome $V13$.

**LowDim-D:** *LowDim-D* represents a more structurally complex synthetic setting with $p = 20$, $n = 6000$, moderate nonlinearity ($\rho_{\text{nonlin}} = 0.5$), and a long-chain causal structure characterized by a single deep path ($\kappa = 1, \ell = 6$). It includes a binary treatment variable $V8$, continuous mediators $V2$, $V16$, and $V19$, a categorical mediator $V18$, a binary mediator $V11$, and a binary outcome $V13$. This configuration captures extended sequential dependencies, where indirect effects propagate through multiple intermediate variables. It provides a testbed for assessing robustness in effect estimation under deeper, multi-step causal pathways.

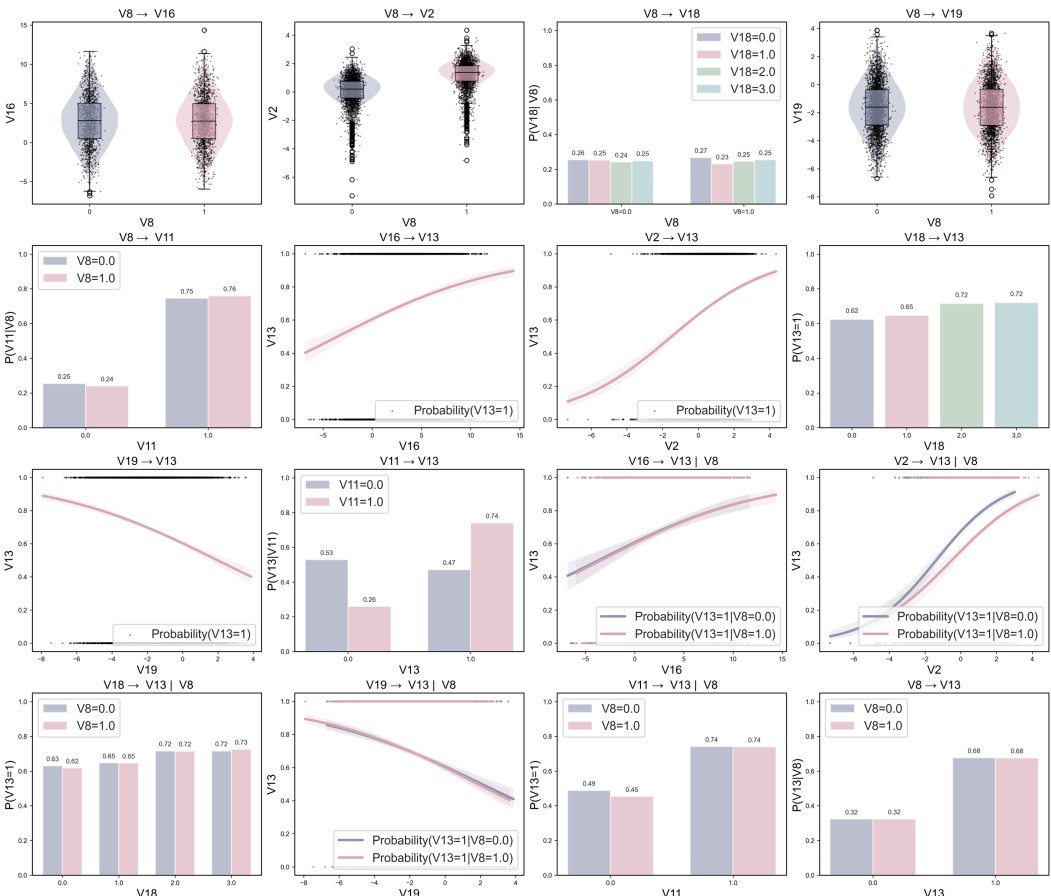

Figure 8: **Visualization of variable relationships of a sampled mediation pathway in LowDim-D**. The pathway comprises binary treatment variable $V8$, continuous mediators $V2$, $V16$ and $V19$, a categorical mediator $V18$, a binary mediator $V11$ and a binary outcome $V13$.

**MidDim-S:** *MidDim-S* increases both dimensionality and structural complexity, with $p = 100$, $n = 6000$, and a moderate level of nonlinearity ($\rho_{\text{nonlin}} = 0.5$), while preserving a simple causal topology with a single short directed path ($\kappa = 1$, $\ell = 3$). The treatment variable $V35$ is binary, mediators include a categorical variable $V2$, a binary variable $V13$, and a continuous variable $V14$, with $V17$ serving as the continuous outcome. This setup is designed to assess algorithm performance in mid-dimensional regimes where the causal structure remains sparse and shallow, isolating the effect of dimensional scaling under otherwise controlled conditions.

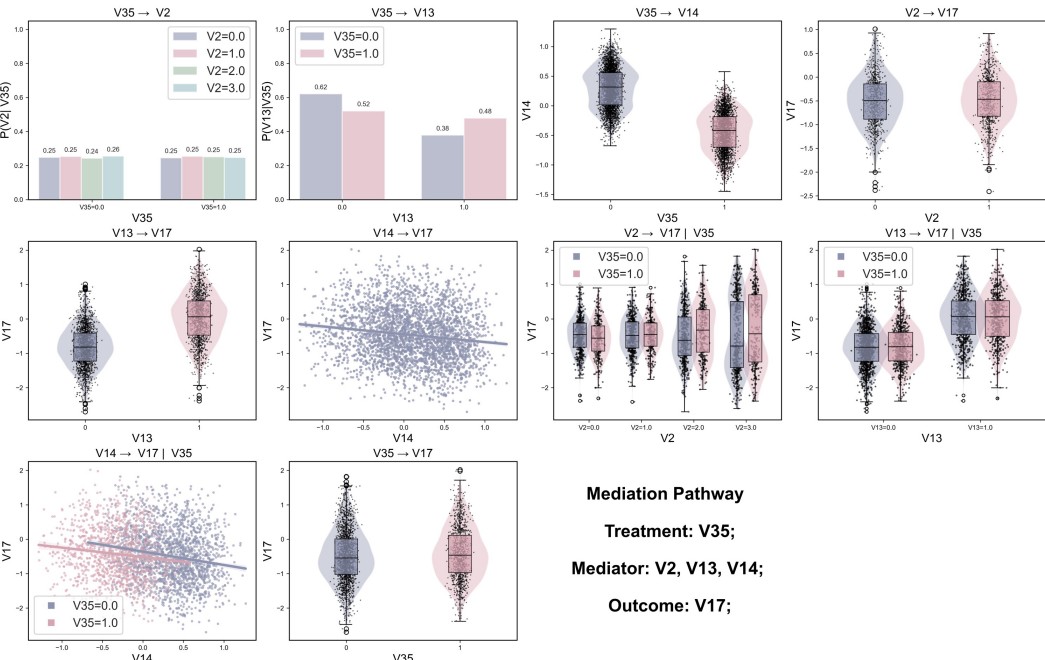

Figure 9: **Visualization of variable relationships of a sampled mediation pathway in MidDim-S.** The pathway comprises binary treatment variable $V35$, a continuous mediator $V14$, a categorical mediator $V2$, a binary mediator $V13$ and a continuous outcome $V17$.

**MidDim-D:** *MidDim-D* extends the structural complexity of the mid-dimensional setting with $p = 100$, $n = 6000$, a nonlinearity ratio of $\rho_{\mathrm{nonlin}} = 0.5$, and a long-chain causal configuration characterized by a single extended path ($\kappa = 1$, $\ell = 6$). The DAG includes multiple layers of intermediate variables connected through a deep sequential structure, challenging algorithms to recover indirect effects that accumulate across heterogeneous transformations. The treatment variable $V35$ is binary, and the mediation pathway involves heterogeneous mediators: a categorical variable $V2$, binary variables $V13$, $V15$ and $V89$, and continuous variables $V14$ and $V40$, with a continuous outcome $V17$. This dataset is well-suited for evaluating a method's ability to capture layered, nonlinear, and high-dimensional mediation dynamics.

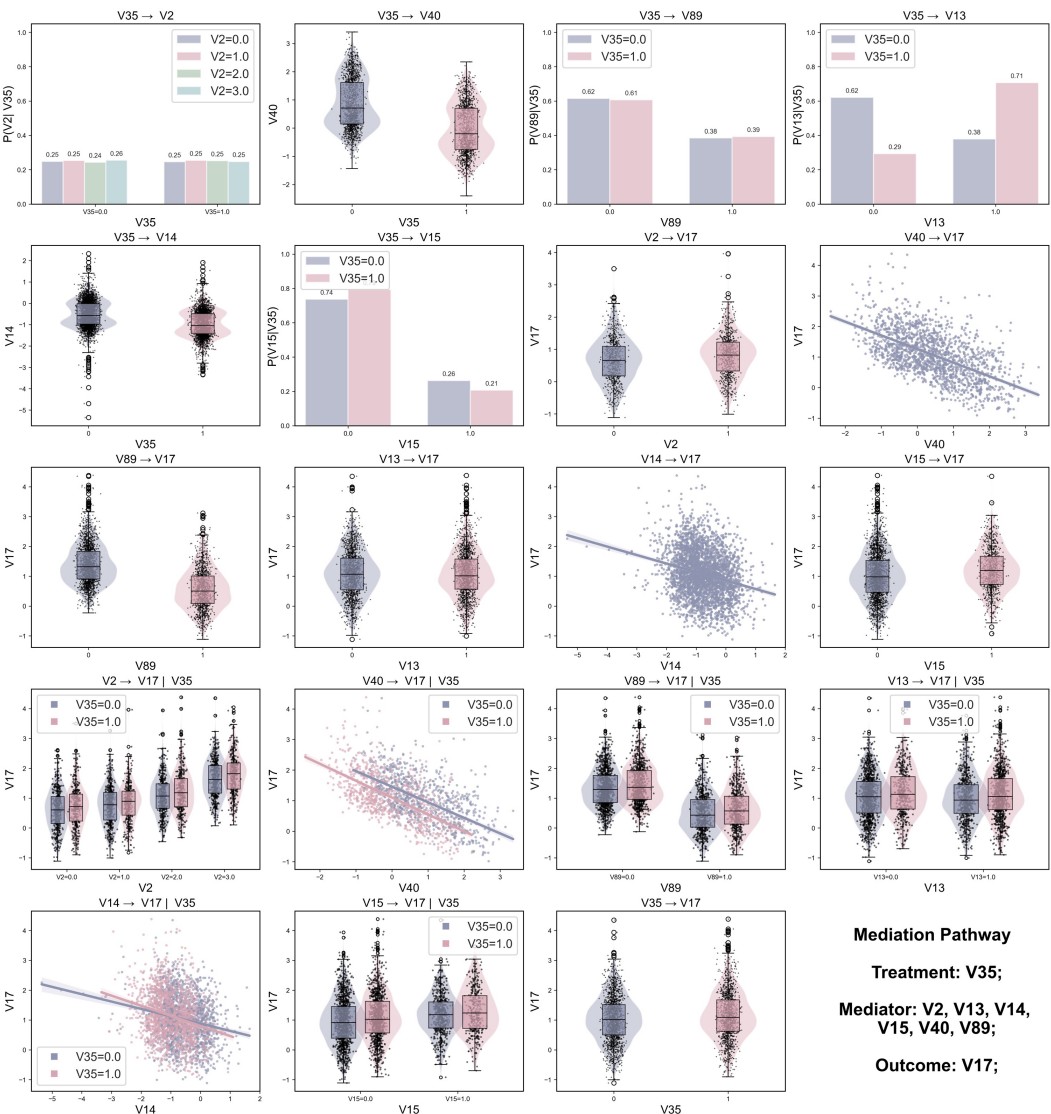

Figure 10: **Visualization of variable relationships of a sampled mediation pathway in MidDim-D**. The pathway comprises a binary treatment variable $V35$, continuous mediators $V14$ and $V40$, a categorical mediator $V2$, binary mediators $V13$, $V15$ and $V89$, and a continuous outcome $V17$.

**HigDim-S:** *HighDim-S* explores a high-dimensional setting with $p = 200$, $n = 6000$, moderate nonlinearity ($\rho_{\text{nonlin}} = 0.5$), and a sparse causal structure characterized by a single short path ($\kappa = 1$, $\ell = 3$). The causal pathway consists of a binary treatment variable $V71$, a continuous mediator $V35$, a binary mediator $V28$, and a binary outcome $V83$. Despite the limited path depth, the high-dimensional background introduces substantial noise and potential confounding, making this configuration well-suited for testing robustness in pathway recovery under low-signal regimes. It is designed to assess performance when precise signal extraction is required in the presence of many irrelevant or weakly related variables.

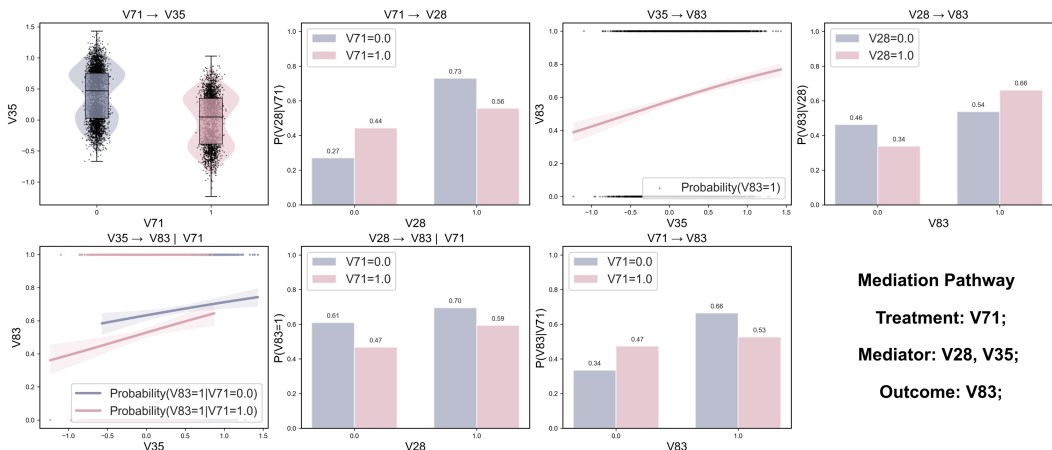

Figure 11: **Visualization of variable relationships of a sampled mediation pathway in HigDim-S**. The pathway comprises a binary treatment variable $V71$, a continuous mediator $V35$, a binary mediator $V28$, and a binary outcome $V83$.

**HigDim-D:** *HighDim-D* represents the most structurally complex setting in the benchmark, with high dimensionality ($p = 200$), moderate nonlinearity ($\rho_{\text{nonlin}} = 0.5$), and a deep causal pathway characterized by a single extended chain ($\kappa = 1, \ell = 6$). The underlying DAG contains multiple layers of variables with diverse data types, connected through a long directed path. The causal structure includes a binary treatment $V71$, continuous mediators $V35$ and $V176$, categorical mediators $V5$ and $V57$, binary mediators $V28$, $V15$, and $V89$, and a binary outcome $V83$. This design models rich causal hierarchies and nonlinear dependencies, mimicking the structure of intricate biological or behavioral systems.

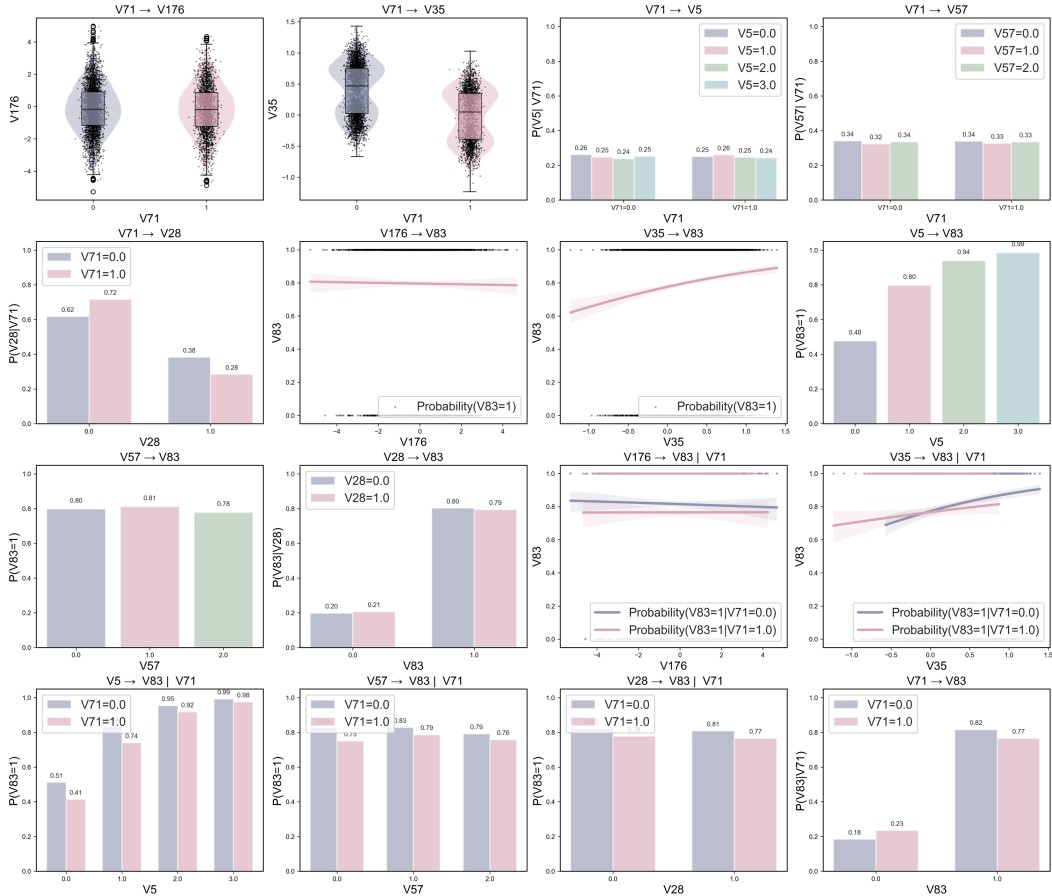

Figure 12: **Visualization of variable relationships of a sampled mediation pathway in HigDim-D**. The pathway comprises a binary treatment variable $V71$, continuous mediators $V35$ and $V176$, categorical mediators $V5$ and $V57$, binary mediators $V28$, $V15$ and $V89$, and a binary outcome $V83$.

**MidDim-P:** *MidDim-P* is designed to evaluate robustness under potential model misspecification, featuring moderate dimensionality ($p = 100$), nonlinearity ($\rho_{\text{nonlin}} = 0.5$), and a parallel causal structure with overlapping pathways ($\kappa = 2, \ell = 3$). The causal graph includes a binary treatment $V35$, continuous mediators $V14$, $V68$, and $V72$, a categorical mediator $V2$, and a continuous outcome $V17$. Unlike prior configurations emphasizing either deep chains or sparse connections, this setting introduces partially redundant, nonlinear paths that may obscure structural signals and challenge estimation accuracy when assumptions are violated.

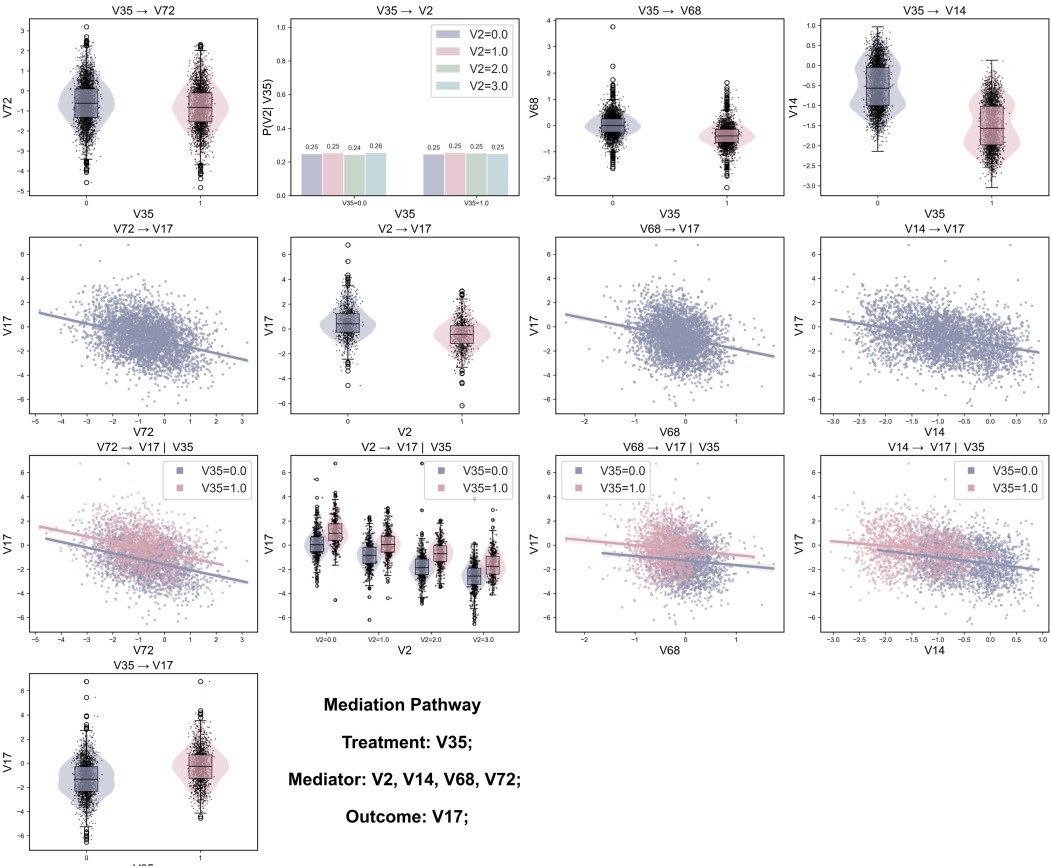

Figure 13: **Visualization of variable relationships of a sampled mediation pathway in MidDim-P**. The pathway comprises a binary treatment variable $V35$, continuous mediators $V14$, $V68$ and $V72$, a categorical mediator $V2$, and a continuous outcome $V17$.

**MidDim-C:**   *MidDim-C* explores mid-dimensional, high-complexity causal dynamics under nonlinear transformations, with $p = 50$, $n = 6000$, and a deep chain structure defined by a single extended path ($\kappa = 1$, $\ell = 6$). The treatment variable $V35$ is binary, the mediators—$V6$, $V14$, $V33$, $V42$, and $V48$—are all continuous, and the binary outcome is $V45$. This configuration emphasizes smooth, continuous interactions across multiple intermediate nodes, where causal influence accumulates gradually along the chain. It is designed to test methods' ability to detect and quantify multi-step effects under structurally rich yet low-parallelism conditions.

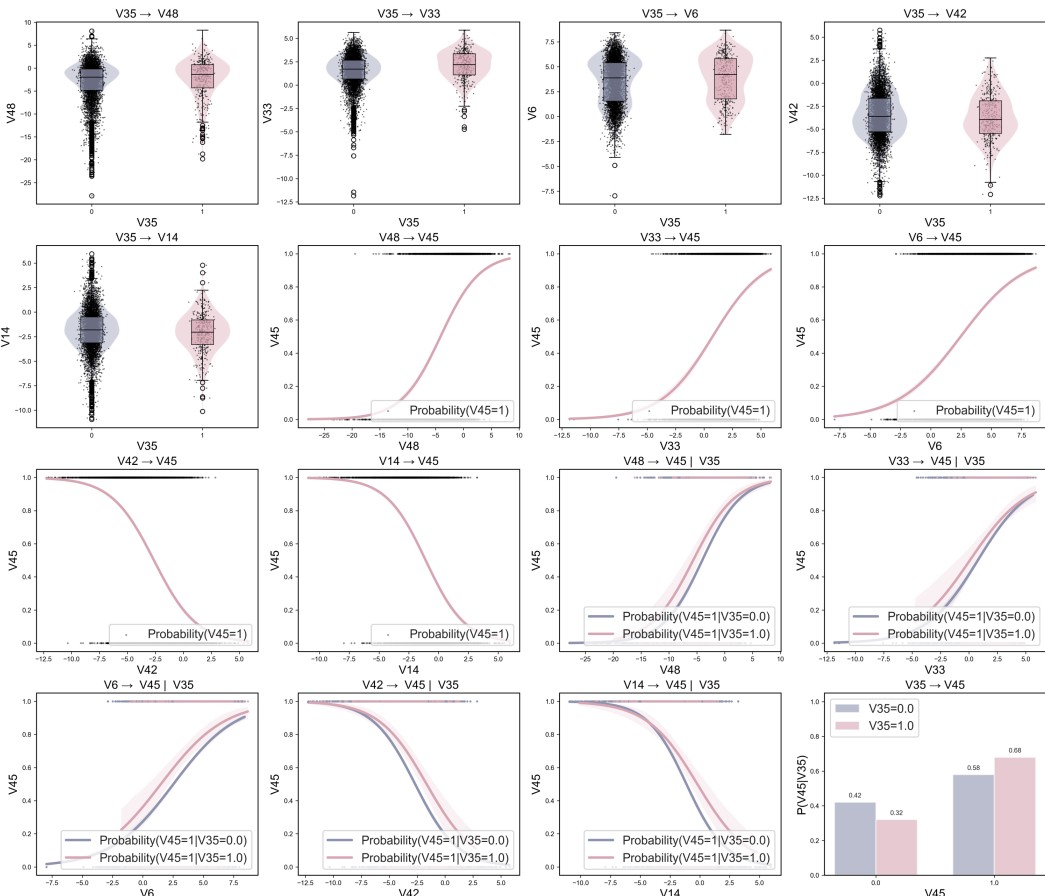

Figure 14: **Visualization of variable relationships of a sampled mediation pathway in MidDim-C**. The pathway comprises a binary treatment variable $V35$, continuous mediators $V6$, $V14$, $V33$, $V42$ and $V48$, and a binary outcome $V45$.

# F  Experiment

## F.1  Experiment Setup

In the structure discovery phase, we configure the Flow-SEM model with a two-layer MLP (hidden_dim_multiplier=2) as the conditional distribution model. The model is trained using the Adam optimizer with a learning rate of 0.001, an initial DAG constraint penalty coefficient $\lambda = 0.1$, which increases by a factor of 10 when the acyclicity constraint is violated (h_threshold=$10^{-8}$), up to a maximum of $10^5$. Sparsity is controlled through L1 regularization ($\alpha = 0.01$), with a maximum gradient norm limit of 1.0. The training iterates for 1000 epochs, with early stopping if no improvement occurs for 100 consecutive epochs.

In the posterior sampling phase, we extract 1000 DAG samples from the learned structure. During sampling, each node retains only the top 3 highest probability incoming edges to control sparseness, and a sigmoid function converts the weight matrix to edge existence probabilities. Graph structures are randomly sampled according to a Bernoulli distribution, validated for acyclicity through topological sorting, and cyclic graphs are discarded.

For path identification, we set the frequency threshold to 5%, meaning paths appearing with frequency above this threshold in the DAG sample set are identified as stable paths. The maximum path length is limited to 10, and multi-processor parallel computing is employed to accelerate the identification process.

The mechanism modeling phase implements 5-fold cross-validation. Each model uses uniform parameters: 50 iterations, batch size of 128, learning rate of $10^{-4}$, and hidden layer dimension of 128. For continuous treatment variables, intervention levels are set at the mean ($\mu$) and the mean plus one standard deviation ($\mu + \sigma$), while binary variables use intervention levels {0,1}.

Effect estimation employs a mixed approach: paths of length 3 use Efficient Influence Function estimation, while longer paths use plugin estimation. Multiple DAG effects are aggregated through Bayesian Model Averaging, considering both within-DAG estimation uncertainty (average variance) and between-DAG structural uncertainty (effect variation), constructing 95% confidence intervals.

We test SIGMA on 10 synthetic datasets with feature dimensions $p \in \{20, 50, 100, 200\}$, nonlinearity degree $\rho \in \{0, 0.5\}$, containing parallel ($\kappa \in \{2, 3, 6\}$) and chain ($\ell \in \{1, 2\}$) structures.

## F.2  Computation of Ground Truth Mediation Effects

To validate SIGMA's estimation accuracy, we compute the true NDE and NIE values for the two mediation pathways through counterfactual simulation. For a mediation pathway $X \to M \to Y$, we define three counterfactual data sets: $D_{ref} = \{(X_i = x_{ref}, M_i(x_{ref}), Y_i(x_{ref}, M_i(x_{ref})))\}_{i=1}^n$, $D_{comp} = \{(X_i = x_{comp}, M_i(x_{comp}), Y_i(x_{comp}, M_i(x_{comp})))\}_{i=1}^n$, and $D_{cf} = \{(X_i = x_{comp}, M_i(x_{ref}), Y_i(x_{comp}, M_i(x_{ref})))\}_{i=1}^n$, where $M_i(x)$ denotes the potential mediator value when treatment $X_i = x$, and $Y_i(x, m)$ represents the potential outcome when $X_i = x$ and $M_i = m$.

Based on these counterfactual sets, the true mediation effects are calculated as: NDE $= \mathbb{E}[Y(x_{comp}, M(x_{ref}))] - \mathbb{E}[Y(x_{ref}, M(x_{ref}))]$ and NIE $= \mathbb{E}[Y(x_{comp}, M(x_{comp}))] - \mathbb{E}[Y(x_{comp}, M(x_{ref}))]$. In Monte Carlo simulation, these expectations are estimated via sample means: $\widehat{\text{NDE}} = \frac{1}{n}\sum_{i=1}^n Y_i(x_{comp}, M_i(x_{ref})) - \frac{1}{n}\sum_{i=1}^n Y_i(x_{ref}, M_i(x_{ref}))$ and $\widehat{\text{NIE}} = \frac{1}{n}\sum_{i=1}^n Y_i(x_{comp}, M_i(x_{comp})) - \frac{1}{n}\sum_{i=1}^n Y_i(x_{comp}, M_i(x_{ref}))$.

We compute these effects using structural equation model specifications with 5000 simulation samples, setting intervention levels for pathway 1 (V5→V19→V16) at $x_{ref} = 0.012$ and $x_{comp} = 1.253$, and for pathway 2 (V12→V10→V6) at $x_{ref} = 0.243$ and $x_{comp} = 0.926$. To generate $D_{cf}$, we extract $M_i(x_{ref})$ values from $D_{ref}$, then set $X_i = x_{comp}$ and fix $M_i = M_i(x_{ref})$, employing topological sorting to ensure proper variable generation order in the DAG.

## F.3 Causal Discovery Evaluation

Table 5: Causal discovery on benchmark datasets.

| Method | Metrics | Dataset 1 | Dataset 2 | Dataset 3 | Dataset 4 | Dataset 5 | Dataset 6 | Dataset 7 | Dataset 8 | Dataset 9 | Dataset 10 |
|---|---|---|---|---|---|---|---|---|---|---|---|
| PC [33] | Adjacency F1 | 0.4082 | 0.439 | 0.5217 | 0.4091 | 0.4427 | 0.4479 | 0.3498 | 0.3475 | 0.4436 | 0.5333 |
| | Orientation F1 | 0.2326 | 0.2286 | 0.2564 | 0.1143 | 0.2869 | 0.2929 | 0.1771 | 0.1605 | 0.2823 | 0.3551 |
| | Causal Accuracy | 0.577 | 0.5735 | 0.5849 | 0.5286 | 0.6268 | 0.6243 | 0.5893 | 0.5806 | 0.6247 | 0.6478 |
| Notears [72] | Adjacency F1 | 0.3728 | 0.4061 | 0.4741 | 0.3727 | 0.4018 | 0.4141 | 0.32 | 0.3194 | 0.4039 | 0.4849 |
| | Orientation F1 | 0.2158 | 0.2065 | 0.2313 | 0.1056 | 0.2636 | 0.2695 | 0.1601 | 0.1448 | 0.2578 | 0.3261 |
| | Causal Accuracy | 0.5291 | 0.5301 | 0.5407 | 0.4828 | 0.5745 | 0.5692 | 0.5388 | 0.5244 | 0.577 | 0.594 |
| Grandag [20] | Adjacency F1 | 0.3922 | 0.4161 | 0.5075 | 0.3935 | 0.4216 | 0.436 | 0.3296 | 0.3283 | 0.4186 | 0.5225 |
| | Orientation F1 | 0.2247 | 0.2172 | 0.2485 | 0.1119 | 0.2738 | 0.2859 | 0.1701 | 0.1571 | 0.2693 | 0.3384 |
| | Causal Accuracy | 0.5424 | 0.5485 | 0.5703 | 0.5066 | 0.5915 | 0.5985 | 0.5633 | 0.5471 | 0.6026 | 0.614 |
| ICANLiGAM [55] | Adjacency F1 | 0.377 | 0.4168 | 0.4897 | 0.384 | 0.4171 | 0.413 | 0.3228 | 0.3226 | 0.4102 | 0.4987 |
| | Orientation F1 | 0.218 | 0.2141 | 0.2432 | 0.1062 | 0.2676 | 0.2747 | 0.1644 | 0.1514 | 0.2628 | 0.3323 |
| | Causal Accuracy | 0.5476 | 0.5371 | 0.5401 | 0.4957 | 0.5792 | 0.5833 | 0.5456 | 0.5417 | 0.5879 | 0.5998 |
| DECI [22] | Adjacency F1 | 0.431 | 0.4453 | 0.5188 | 0.4049 | 0.4512 | 0.4477 | 0.3326 | **0.3814** | **0.4869** | 0.5077 |
| | Orientation F1 | 0.2252 | 0.2256 | 0.2576 | 0.1141 | 0.2858 | 0.2891 | 0.1758 | 0.1551 | 0.295 | 0.353 |
| | Causal Accuracy | 0.5833 | 0.5661 | 0.623 | 0.5266 | 0.6129 | 0.5955 | 0.5872 | 0.5643 | 0.6776 | 0.6625 |
| **SIGMA (Ours)** | Adjacency F1 | **0.5263** | **0.4909** | **0.5602** | **0.4512** | **0.4939** | **0.4998** | **0.3998** | 0.3741 | 0.4754 | **0.5918** |
| | Orientation F1 | **0.4474** | **0.2601** | **0.2784** | **0.1266** | **0.3185** | **0.3163** | **0.1925** | **0.1782** | **0.3162** | **0.386** |
| | Causal Accuracy | **0.7695** | **0.643** | **0.6719** | **0.5712** | **0.6808** | **0.6747** | **0.6437** | **0.6561** | **0.6914** | **0.7259** |

## F.4 Real Data HPP Validation

Table 6: Estimated $\widehat{\text{NDE}}$, $\widehat{\text{NIE}}$ and $\widehat{\text{TE}}$ with the corresponding 95% confidence intervals (CI) across all the sub-pathways within the validated mediation pathways shown in Figure 15(b). Estimates are obtained using the proposed SIGMA framework with structural uncertainty propagation.

| Pathways | $\widehat{\text{TE}}$ Estimate | CI Lower | CI Upper | $\widehat{\text{NDE}}$ Estimate | CI Lower | CI Upper | $\widehat{\text{NIE}}$ Estimate | CI Lower | CI Upper |
|---|---|---|---|---|---|---|---|---|---|
| $4 \to 1 \to 6$ | 0.1880 | 0.1292 | 0.2432 | 0.1422 | 0.0902 | 0.1995 | 0.0458 | 0.0112 | 0.0823 |
| $4 \to 1 \to 12 \to 6$ | 0.1880 | 0.1306 | 0.2111 | 0.1636 | 0.0818 | 0.2006 | 0.0243 | 0.0036 | 0.0597 |
| $4 \to 1 \to 2 \to 6$ | 0.0462 | 0.0072 | 0.0737 | -0.0076 | -0.0614 | 0.0502 | 0.0538 | 0.0127 | 0.0921 |
| $4 \to 1 \to 2 \to 12 \to 6$ | 0.0554 | -0.0756 | -0.0310 | 0.0244 | 0.0006 | 0.0604 | 0.0310 | -0.0034 | 0.0790 |
| $4 \to 1 \to 13 \to 6$ | 0.0554 | -0.0625 | -0.0313 | 0.0391 | -0.0077 | 0.0758 | 0.0163 | -0.0149 | 0.0580 |
| $3 \to 1 \to 12 \to 6$ | 0.0965 | 0.0638 | 0.1384 | 0.0821 | -0.0024 | 0.1100 | 0.0144 | -0.0075 | 0.0543 |
| $3 \to 1 \to 2 \to 12 \to 6$ | 0.0965 | 0.0472 | 0.1198 | 0.0192 | -0.0033 | 0.0575 | 0.0773 | 0.0502 | 0.1023 |
| $2 \to 1 \to 6$ | 0.1584 | 0.0853 | 0.2267 | -0.0252 | -0.0884 | 0.0522 | 0.1836 | 0.0777 | 0.2409 |
| $2 \to 1 \to 12 \to 6$ | 0.0323 | -0.0580 | 0.0811 | 0.1314 | 0.0816 | 0.1402 | -0.0991 | -0.1346 | -0.0124 |
| $2 \to 1 \to 2 \to 12 \to 6$ | -0.0397 | -0.0653 | -0.0036 | -0.0462 | -0.0809 | -0.0093 | 0.0065 | 0.0132 | 0.0502 |
| $2 \to 1 \to 2 \to 6$ | 0.1584 | 0.0834 | 0.1839 | 0.1410 | 0.0023 | 0.2262 | 0.0174 | 0.0000 | 0.0408 |

Table 7: Estimated $\widehat{\text{NDE}}$, $\widehat{\text{NIE}}$ and $\widehat{\text{TE}}$ with the corresponding 95% confidence intervals (CI) across all the sub-pathways within the validated mediation pathways shown in Figure 15(b). Estimates are obtained using classical regression-based mediation analysis without structural uncertainty modeling.

| Pathways | $\widehat{\text{TE}}$ Estimate | CI Lower | CI Upper | $\widehat{\text{NDE}}$ Estimate | CI Lower | CI Upper | $\widehat{\text{NIE}}$ Estimate | CI Lower | CI Upper |
|---|---|---|---|---|---|---|---|---|---|
| $4 \to 12 \to 6$ | 0.1880 | 0.1624 | 0.2133 | 0.1875 | 0.1618 | 0.2126 | 0.0005 | -0.0063 | 0.0017 |
| $4 \to 12$ | 0.1880 | 0.1623 | 0.2124 | 0.1684 | 0.1414 | 0.1950 | 0.0196 | 0.0115 | 0.0285 |
| $4 \to 2 \to 12 \to 6$ | -0.0554 | -0.0871 | -0.0302 | -0.0502 | -0.0767 | -0.0249 | -0.0052 | -0.0097 | -0.0035 |
| $4 \to 2 \to 13$ | -0.0554 | -0.0800 | -0.0273 | -0.0626 | -0.0866 | -0.0381 | 0.0072 | 0.0031 | 0.0118 |
| $4 \to 12 \to 13$ | 0.0462 | 0.0210 | 0.0713 | 0.0340 | 0.0074 | 0.0614 | 0.0123 | 0.0042 | 0.0203 |
| $3 \to 1 \to 12$ | 0.0964 | 0.0736 | 0.1240 | 0.0482 | 0.0204 | 0.0752 | 0.0483 | 0.0334 | 0.0638 |
| $3 \to 1 \to 6$ | 0.0858 | 0.0624 | 0.1100 | 0.0853 | 0.0574 | 0.1123 | -0.0005 | -0.0147 | 0.0157 |
| $3 \to 1 \to 12 \to 6$ | -0.0730 | -0.1087 | -0.0545 | -0.0821 | -0.1091 | -0.0553 | 0.0000 | -0.0002 | 0.0002 |
| $3 \to 1 \to 2 \to 12 \to 6$ | 0.0964 | 0.0745 | 0.1188 | 0.0956 | 0.0612 | 0.1297 | 0.0004 | -0.0011 | 0.0040 |
| $3 \to 1 \to 2 \to 6$ | 0.1853 | 0.1537 | 0.2165 | 0.1835 | 0.1519 | 0.2150 | 0.0019 | -0.0051 | 0.0078 |
| $2 \to 1 \to 6$ | 0.0323 | -0.0100 | 0.0706 | -0.0183 | -0.0578 | 0.0198 | 0.0506 | 0.0195 | 0.0863 |
| $2 \to 1 \to 12 \to -13$ | -0.0397 | -0.0655 | -0.0124 | -0.0365 | -0.0459 | -0.0285 | 0.0077 | 0.0029 | 0.0131 |
| $2 \to 1 \to 12 \to 6$ | 0.1584 | 0.1338 | 0.1828 | 0.1839 | 0.1430 | 0.2250 | 0.0008 | -0.0004 | 0.0023 |

Table 8: Estimated $\widehat{\text{NDE}}$, $\widehat{\text{NIE}}$ and $\widehat{\text{TE}}$ with the corresponding 95% confidence intervals (CI) across all the sub-pathways within the novely discovered mediation pathways shown in Figure 15(c). Estimates are obtained using the proposed SIGMA framework with structural uncertainty propagation.

| | $\widehat{\text{TE}}$ | | | $\widehat{\text{NDE}}$ | | | $\widehat{\text{NIE}}$ | | |
|---|---|---|---|---|---|---|---|---|---|
| Pathways | Estimate | CI Lower | CI Upper | Estimate | CI Lower | CI Upper | Estimate | CI Lower | CI Upper |
| 3 → 9 → -13 | -0.0720 | -0.0997 | -0.0499 | 0.0534 | 0.0014 | 0.0815 | -0.1254 | -0.1774 | -0.0917 |
| 4 → 9 → -13 | -0.0858 | -0.1084 | -0.0624 | 0.0756 | 0.0315 | 0.1028 | -0.1614 | -0.2148 | -0.1172 |
| 4 → 3 → -13 | -0.0554 | -0.0719 | -0.0310 | 0.0823 | 0.0544 | 0.1036 | -0.1377 | -0.1845 | -0.1036 |
| 10 → -13 | -0.0359 | -0.0528 | -0.0128 | 0.0306 | 0.0008 | 0.0526 | -0.0665 | -0.0904 | -0.0206 |
| 10 → -12 → -13 | -0.0270 | -0.0442 | -0.0072 | 0.0425 | 0.0127 | 0.0568 | -0.0695 | -0.0917 | -0.0326 |
| 10 → -12 → -14 → -13 | -0.0441 | -0.0652 | -0.0209 | 0.0387 | 0.0095 | 0.0652 | -0.0828 | -0.1152 | -0.0411 |
| 8 → -11 → -13 | -0.0333 | -0.0508 | -0.0126 | 0.0315 | 0.0011 | 0.0452 | -0.0648 | -0.0842 | -0.0221 |
| 8 → -11 → -12 → -13 | -0.0550 | -0.0770 | -0.0291 | 0.0506 | 0.0018 | 0.0814 | -0.1056 | -0.1515 | -0.0633 |
| 8 → -13 → -15 | 0.0763 | 0.0501 | 0.1011 | -0.0516 | -0.1002 | -0.0173 | 0.1279 | 0.0814 | 0.1919 |

Table 9: Estimated $\widehat{\text{NDE}}$, $\widehat{\text{NIE}}$ and $\widehat{\text{TE}}$ with the corresponding 95% confidence intervals (CI) across all the sub-pathways within the novely discovered mediation pathways shown in Figure 15(c). Estimates are obtained using classical regression-based mediation analysis without structural uncertainty modeling.

| | $\widehat{\text{TE}}$ | | | $\widehat{\text{NDE}}$ | | | $\widehat{\text{NIE}}$ | | |
|---|---|---|---|---|---|---|---|---|---|
| Pathways | Estimate | CI Lower | CI Upper | Estimate | CI Lower | CI Upper | Estimate | CI Lower | CI Upper |
| 3 → 9 → -13 | -0.0723 | -0.0955 | -0.0500 | -0.0764 | -0.0987 | -0.0533 | 0.0030 | 0.0014 | 0.0054 |
| 4 → 9 → -13 | 0.0858 | 0.0614 | 0.1094 | 0.0983 | 0.0734 | 0.1221 | -0.0096 | 0.0002 | -0.0088 |
| 4 → 9 → -13 → -12 | -0.0554 | -0.0800 | -0.0306 | -0.0576 | -0.0841 | -0.0326 | 0.0022 | 0.0007 | 0.0041 |
| 10 → -13 | 0.0462 | 0.0212 | 0.0712 | 0.0554 | 0.0312 | 0.0809 | -0.0083 | 0.0010 | -0.0063 |
| 10 → -13 → -12 | 0.0360 | 0.0132 | 0.0584 | 0.0306 | 0.0080 | 0.0531 | 0.0053 | 0.0010 | 0.0092 |
| 10 → -13 → -12 → -1 | -0.0270 | -0.0501 | -0.0042 | -0.0271 | -0.0494 | -0.0043 | 0.0001 | -0.0010 | 0.0005 |
| 13 → -12 → -1 | -0.0441 | -0.0681 | -0.0205 | -0.0521 | -0.0748 | -0.0295 | 0.0077 | 0.0054 | 0.0116 |
| 2 → -1 | -0.0167 | -0.0391 | 0.0049 | -0.0254 | -0.0464 | -0.0036 | 0.0090 | 0.0025 | 0.0156 |
| 2 → -1 → -15 | -0.0325 | -0.0594 | -0.0085 | -0.0113 | -0.0330 | 0.0104 | -0.0212 | -0.0393 | -0.0051 |
| 8 → -2 → -1 → -15 | -0.0325 | -0.0694 | 0.0028 | -0.0087 | -0.0315 | 0.0140 | -0.0239 | -0.0510 | 0.0032 |
| 8 → -13 → -15 | -0.0325 | -0.0665 | 0.0015 | 0.0014 | -0.0225 | 0.0246 | -0.0340 | -0.0702 | 0.0019 |
| 8 → -5 → -13 | -0.0547 | -0.0812 | -0.0281 | -0.0151 | -0.0778 | 0.0477 | -0.0035 | -0.0065 | -0.0017 |
| 3 → -15 | 0.0763 | -0.1011 | 0.0515 | -0.0765 | -0.1014 | -0.0477 | 0.0002 | -0.0013 | 0.0019 |

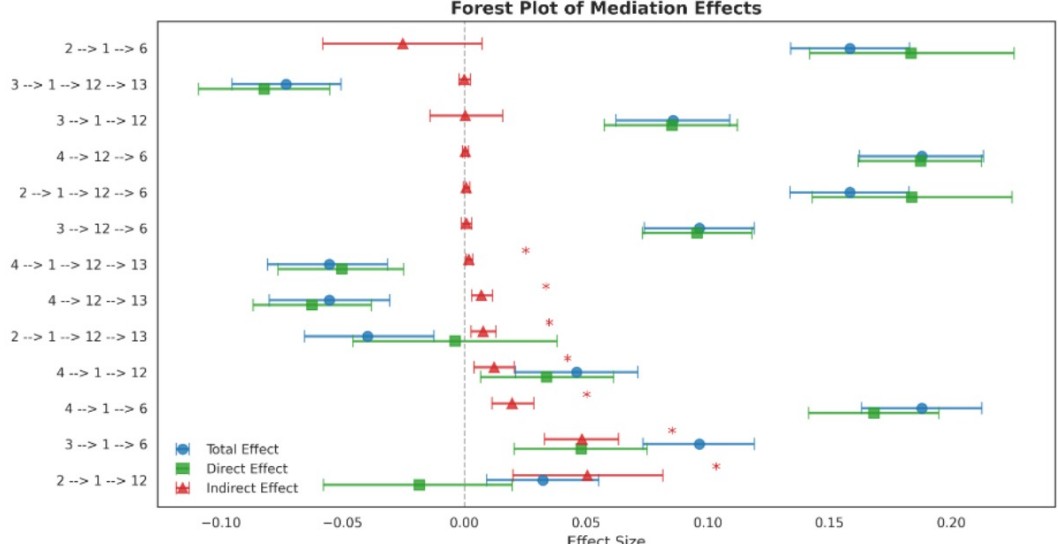

Figure 15: Visualization of the effect estimation on HPP data.

