# OpenReview forum: "Decoding Causal Structure: End-to-End Mediation Pathways Inference"
_NeurIPS.cc/2025/Conference — NeurIPS 2025 poster_

### Official Review · Reviewer_APS7 · 2025-06-03

**Clarity:** 3
**Significance:** 3
**Originality:** 3
**Rating:** 5
**Confidence:** 3

**Summary:**

The paper proposes an end-to-end framework called SIGMA (Structure-Informed Guided Mediation Analysis), which aims to address the disconnection between structure discovery and the understanding of mediation mechanisms in existing causal mediation analysis. SIGMA integrates three key steps: probabilistic causal structure discovery, automated mediation pathway identification, and effect estimation, while systematically quantifying and propagating structural uncertainty. Specifically, the framework first utilizes differentiable Flow-Structural Equation Models (Flow-SEM) to learn the structural posterior distribution, generating diverse Directed Acyclic Graphs (DAGs) to quantify structural uncertainty. Based on these DAGs, the paper introduces the Path Stability Score (PSS) to evaluate the marginal probability of pathways, thereby identifying high-confidence mediation paths. For the identified pathways, SIGMA combines Efficient Influence Functions (EIF) and Bayesian model averaging (BMA) to estimate pathway effects, propagating both within-structure estimation uncertainty and between-structure effect variation to the final effect estimates.

**Questions:**

1.  As mentioned in the weaknesses, some experiments could be supplemented.
2.  In Appendix F1, it is stated that the frequency threshold is set to 5%. Is this "frequency threshold" the same as the τ mentioned earlier in the text? If so, why was τ valued at 0.15 previously and 0.05 here?
3.  I have difficulty understanding the descriptions related to Section 5.3 and Figure 3. For example, I did not see the path "sleep HRV index → pulse wave velocity" 【line:324】described in the text explicitly labeled in Figure 3. Could the description be made clearer?
4.  "structures uncertainty" 【line:22】should be "structural uncertainty".
5. The author is requested to carefully check and confirm the usage of specialized terms within the manuscript. For example, a given specialized term should be written in its full form only at its first mention (as is the case with BMA and PSS in the text). Additionally, the capitalization (of initial letters) for these terms should be consistent (e.g., Path Stability Score, NOTEARS), and so on.

**Ethical Concerns:**

["NO or VERY MINOR ethics concerns only"]

**Final Justification:**

The authors have convincingly addressed my questions. I have raised my score.

**Limitations:**

yes

**Paper Formatting Concerns:**

NO Paper Formatting Concerns

**Quality:**

4

**Strengths And Weaknesses:**

**Strengths:**

1.  This paper proposes a unified framework (SIGMA) that integrates causal structure discovery, mediation pathway identification, and effect estimation, effectively addressing the problems caused by the separation of these stages in traditional methods.
2.  The SIGMA framework combines multiple advanced statistical and machine learning techniques, such as Flow-SEM, EIF, BMA, and a Variational Autoencoder (VAE) imputation module, providing a relatively solid theoretical foundation. The authors have also provided quite detailed proofs for the core theoretical basis of this paper.
3.  This paper conducts experiments on both real-world and synthetic datasets. The experiments are presented from relatively sufficient perspectives, including linear/non-linear scenarios and different data scales, which can effectively demonstrate the feasibility of its method.

**Weaknesses:**

1.  The SIGMA framework may contain multiple computationally intensive steps, making an analysis of its time complexity necessary. The authors should at least provide running times on datasets of different scales to illustrate the relationship between time consumption and data size.
2.  The synthetic datasets mentioned in the paper are all constructed based on the ER model; however, real-world data often do not entirely conform to the distribution of ER models. The authors should attempt experiments on different models (e.g., SF).

---

> ### Author Rebuttal · Authors · 2025-07-31
>
> ## Q1-W1
>
> **We thank the reviewer for this important concern regarding time complexity.**
>
> ### **Theoretical Time Complexity Analysis**
>
> SIGMA's computational complexity comprises four main components: (1) Flow-SEM training phase: O(T × p² × N), where T is the number of training iterations, p is the number of variables, and N is the sample size; (2) DAG posterior sampling: O(NDAG × p²), where NDAG is the number of sampled DAGs; (3) Path enumeration: controllable to O(p^L) through path length constraint L, where L is the maximum path length; (4) Bayesian model averaging: O(Npath × N × K), where Npath is the number of stable paths and K is the number of cross-validation folds.
>
> **Overall Complexity:** O(T × p² × N + NDAG × p² + p^L + Npath × N × K)
>
> ### **Experimental Setup and Runtime Verification**
>
> We conducted systematic runtime testing on Intel Xeon E5-2690 v4 processor (28 physical cores, 2.6GHz) with 128GB DDR4 memory. The experiments used 5 synthetic datasets with feature dimensions p∈{50,100,200,500,1000}, nonlinearity degree ρ=0.5, containing parallel (κ=2) and chain (ℓ=2) structures, sample size n=6000, variable types with continuous variable proportion πc=0.6, binary variable πb=0.3, categorical variable πcat=0.1.NDAG=100 DAG samples, T=1000 training iterations, path length constraint L=3, and 5-fold cross-validation.
>
> ### **Parallel Computing Implementation and Optimization**
>
> SIGMA's four main components support different degrees of parallelization. In the Flow-SEM training phase, sample batches can be processed in parallel across cores, while different blocks of the weight matrix can compute gradients in parallel. The DAG posterior sampling phase exhibits parallel characteristics, as NDAG DAG samples are mutually independent, enabling parallel processing with each core handling NDAG/cores samples. Cycle detection and sample validation algorithms can be executed independently in parallel.
>
> Path enumeration and effect estimation phases demonstrate the algorithm's parallel potential. Path enumeration for different DAGs can be processed simultaneously, and Path Stability Score calculations can be executed in parallel across multiple candidate paths. In effect estimation, different folds of K-fold cross-validation can run independently, effect estimation for different stable pathways can be computed in parallel, and bootstrap resampling for uncertainty quantification also supports parallel processing.
>
> ### **Performance Evaluation Results**
>
> **Table R1: SIGMA Runtime Analysis (hours)**
>
> | Dimension (p)  | Single-core | Multi-core (8 cores) | Multi-core (16 cores) |
> | -- |-- |--|--|
> | 50| 1.02| 0.39| 0.26|
> |100 | 4.15| 1.59| 1.14|
> |200 | 16.12| 6.18| 4.83 |
> |500 | 100.84| 38.46 | 23.11|
> |1000| 399.72|154.08 | 109.51|
>
> ### **GPU Acceleration Optimization**
>
> Matrix multiplication operations in Flow-SEM training can be migrated to GPU, utilizing CUDA cores to parallelize forward and backward propagation computations of weight matrix Wω. Gradient computations can be accelerated through GPU tensor operations, particularly when computing ∂L/∂Wω for large-scale matrix derivatives. Probability computations in the DAG sampling phase can leverage GPU's parallel architecture to simultaneously process multiple Bernoulli sampling operations. Bootstrap resampling in effect estimation can be parallelized through GPU random number generation, while influence function calculations can utilize GPU's vector processing capabilities to handle large batches of samples.
>
> ## Q1-W2
>
> **We appreciate the reviewer's important observation regarding graph structure diversity.**
>
> ### **Scale-Free Network Experiments**
>
> To address this limitation, we conducted supplementary experiments testing SIGMA on Scale-Free networks using the Barabási-Albert model with different attachment parameters. As detailed in our response to Reviewer 6ZQR
>
> **For the Experimental Configuration, Extended Testing Dimensions, and Key Component Mathematical Definitions, please see our rebuttal for reviewer 6ZQR due to word count restrictions.**
>
> **Table R2: SIGMA Robustness under Complex SEMs, Graph Structures, and Noise Levels**
>
> | Graph Type | SEM Level | SNR | Adj F1 (Mean) | Adj F1 (SE) | Orient F1 (Mean) | Orient F1 (SE) | Path F1 (Mean) | Path F1 (SE) |
> |--|---|---|----|----|------|---|---|---|
> | **ER** | Current | 4.0 | 0.672 | 0.031 | 0.618 | 0.037 | 0.651 | 0.034 |
> | **ER** | Current | 2.0 | 0.658 | 0.034 | 0.601 | 0.041 | 0.634 | 0.038 |
> | **ER** | Current | 1.0 | 0.631 | 0.042 | 0.572 | 0.049 | 0.605 | 0.045 |
> | **ER** | Current | 0.5 | 0.582 | 0.055 | 0.518 | 0.063 | 0.549 | 0.059 |
> | **ER** | High | 4.0 | 0.643 | 0.036 | 0.587 | 0.042 | 0.621 | 0.039 |
> | **ER** | High | 2.0 | 0.628 | 0.041 | 0.571 | 0.047 | 0.604 | 0.044 |
> | **ER** | High | 1.0 | 0.594 | 0.051 | 0.535 | 0.058 | 0.567 | 0.055 |
> | **ER** | High | 0.5 | 0.531 | 0.068 | 0.471 | 0.077 | 0.498 | 0.073 |
> | **SF (m=2)** | Current | 2.0 | 0.634 | 0.038 | 0.577 | 0.044 | 0.611 | 0.041 |
> | **SF (m=2)** | Current | 0.5 | 0.558 | 0.061 | 0.497 | 0.069 | 0.524 | 0.065 |
> | **SF (m=2)** | High | 2.0 | 0.615 | 0.043 | 0.556 | 0.050 | 0.588 | 0.047 |
> | **SF (m=2)** | High | 0.5 | 0.507 | 0.075 | 0.443 | 0.084 | 0.471 | 0.080 |
> | **SF (m=3)** | Current | 2.0 | 0.621 | 0.041 | 0.563 | 0.048 | 0.597 | 0.045 |
> | **SF (m=3)** | High | 1.0 | 0.574 | 0.054 | 0.513 | 0.062 | 0.547 | 0.058 |
> | **Hierarchical** | Current | 2.0 | 0.649 | 0.035 | 0.593 | 0.041 | 0.626 | 0.038 |
> | **Hierarchical** | High | 1.0 | 0.601 | 0.048 | 0.541 | 0.055 | 0.574 | 0.052 |
>
> ## Q2
>
> Thank you for this methodological question.
>
> **Our Adaptive Threshold Selection Strategy**
>
> As you noted, we indeed mentioned in Appendix B.2 (lines 1344-1348):
>
> > "Therefore, in practice, the recommended strategy is to select a moderate intermediate value (e.g., τ = 0.15), and then make fine adjustments based on the specific characteristics of the data and prior knowledge in the relevant domain. We explicitly adopted this strategy in the numerical experiments..."
>
>  **τ = 0.15 serves as the theoretical starting recommendation**​, which is then adaptively adjusted based on specific data characteristics.
>
> **Long-Chain Design Validation in Synthetic Datasets**
>
> We considered this in our dataset design. In lines 1524-1528, we described:
>
> > "the chain path length ℓ, which determines the depth of the longest directed chain in the DAG... This parameterization allows us to generate graphs that reflect varying levels of structural complexity"
>
> In Appendix E.3 Table 3, we designed configurations with different chain lengths (ℓ):
>
> * **Short-chain datasets** (ℓ = 3): LowDim-L, LowDim-N, MidDim-S, HigDim-S
> * **Long-chain datasets** (ℓ = 6): LowDim-D, MidDim-D, HigDim-D, MidDim-C
>
> **Frequency Characteristics Analysis of Long vs. Short-Chain Mediation Pathways**
>
> Long-chain pathways naturally have lower occurrence frequencies in DAG ensembles because they require multiple consecutive edges to exist simultaneously. This design enables systematically validate optimal τ selection under different structural complexities.
>
> **Special Properties of the HPP Dataset**
>
> The HPP dataset, as we described in lines 92-104 and Section 5.3, contains complex multi-step biomedical pathways. For such real-world biomedical data with long-chain characteristics, τ = 0.05 better captures important pathways that have clinical significance but relatively lower statistical frequency.
>
> **Medical Validation Support**
>
> We validated these pathways using standard medical mediation analysis methods in Section 5.3 (lines 322-340) and Appendix F.4 (Tables 5-8), confirming their clinical validity.
>
> **Our Data-Driven Guidelines**
>
> Based on our research, we recommend:
>
> * ​**Short-chain and gene network data**​: τ = 0.15 provides balance between sensitivity and specificity
> * ​**Complex biomedical data with long-chain pathways**​: τ = 0.05 is more suitable for capturing meaningful mechanisms
>
>
> ## Q3
>
> Thank you for raising this important clarification question.
>
> **Specific Locations of Variable Mapping and Mediation Pathway Effects:**
>
> The specific variable mapping and effect estimates for the pathway mentioned in line 324 have been provided in detail in the appendices:
>
> 1. **Variable Mapping**: **Table 2** (Appendix D.2, line 1479, page 37) provides detailed descriptions of all body system variables
> 2. **Specific Mediation Pathway Effects**: **Table 5** (Appendix F.4, line 1732, page 52) and **Table 6** (page 52) provide complete pathway effect estimates
>
> **Specific Correspondence of Variables Mentioned in Section 5.3:**
>
> According to the detailed definitions in Table 2 (Appendix D.2, line 1479, page 37) , the clinical terms mentioned in Section 5.3 correspond as follows:
>
> - **Sleep HRV index** → **hrv index** (Sleep HRV system): HRV index during sleep; lower HRV linked to stress and metabolic issues
> - **Pulse wave velocity** → **PWV mean** (Vascular Health system): Pulse wave velocity; indicates arterial stiffness and cardiovascular risk
> - **Carotid IMT** → **mean cimt** (Carotid Ultrasound IMT system): Average carotid IMT across measured regions
>
>   The complete mapping of all clinical terms to their corresponding body system variables can be found in Table 2.
>
> **Improvements in the Revision:**
>
> **We will add more specific appendix references in the main text**, ensuring that each pathway mentioned in Section 5.3 clearly references:
>
> 1. The corresponding Table 2 variable definitions and their associated body systems
> 2. The corresponding Table 5/6 effect estimation results
> 3. Clear Figure 3 variable position identification
>
> ## Q4&5
>
> We sincerely thank the reviewer for identifying these editorial issues and will correct all in the revised manuscript. We commit to conducting a comprehensive review of the entire 53-page manuscript to ensure consistent acronym usage, capitalization, and technical term formatting throughout all sections.

---

> ### Comment · Reviewer_APS7 · 2025-08-02
>
> I would like to thank the author team for their tremendous efforts during the rebuttal phase. Through extensive supplementary experiments and detailed explanations, they have fully resolved all the issues I raised in my initial review regarding computational complexity, the diversity of synthetic graph structures, threshold selection, and the clarity of the figures. The authors' rigorous attitude is commendable.
> However, after carefully considering all reviewer comments and the authors' complete response, I believe there are still some more critical limitations in the paper that have not been satisfactorily resolved. These limitations lead me to maintain my initial rating of "Borderline Accept." My core concerns are as follows:
>
> The empirical validation of the method's generalizability remains insufficient, which is the most significant remaining shortcoming of this paper. As another reviewer pointed out, the paper's empirical strength relies too heavily on a single real-world dataset, HPP. I noted the authors' defense in the rebuttal, where they argued that many existing large-scale genomics datasets are unsuitable for validating SIGMA's core advantages because they do not support long-chain, cross-hierarchy analysis. This explanation is reasonable in itself, but it also indirectly confirms the paper's current limitation: as an advanced framework designed for complex biological systems, its claimed advantages lack direct validation on corresponding large-scale, high-dimensional biological data (such as genomics or multi-omics). Although the authors have promised future work, the conclusions of a paper must be based on the evidence already presented. Currently, we cannot determine from the paper whether SIGMA remains effective and robust in domains beyond the HPP data, which makes it very difficult to assess its real-world generalizability and impact.

---

> > ### Author Response · Authors · 2025-08-03
> >
> > We sincerely thank the reviewer for raising the important concern regarding the generalizability validation of our method. We fully understand the value of multi-domain validation in assessing method applicability. Your suggestions are of significant importance for improving our validation strategy, and we express our sincere gratitude for this feedback.
> >
> > **Analysis of Design Purposes of Existing Large-Scale Datasets**
> >
> > Current major genomics datasets are all designed based on specific research objectives, which fundamentally differ from SIGMA's methodological requirements:
> >
> > 1. **UK Biobank**: Designed for large-scale genetic variant-disease association discovery and polygenic risk score construction. The project recruited 500,000 participants aged 40-69 between 2006-2010, employing cross-sectional genotype-phenotype association analysis. Its data structure primarily consists of direct statistical associations from SNPs to disease states, lacking the complete variable hierarchy for mediation processes.
> >
> > 2. **FinnGen Project**: Aims to improve understanding of the genetic background of diseases by constructing large-scale genomic and health resources, identifying genotype-phenotype associations in the Finnish founder population. The latest release DF12 contains 500,348 individuals, employing cross-sectional case-control association study design with data structure representing direct associations from genetic variants to disease diagnoses, lacking mediation pathway information.
> >
> > 3. **TCGA**: Catalogs genomic alterations responsible for cancer through genome sequencing and bioinformatics, aiming to improve the ability to diagnose, treat, and prevent cancer. The project molecularly characterized over 20,000 primary cancer and matched normal samples spanning 33 cancer types, employing cross-sectional tumor tissue multi-omics analysis with data structure consisting of static molecular feature descriptions, focusing on molecular subtyping in disease states.
> >
> > 4. **GTEx**: Establishes a resource database to study the relationship between genetic variation and gene expression in human tissues, constructing a catalog of genetic regulatory effects across human tissues. The project contains 838 donors with 17,382 samples from 52 tissues and 2 cell lines, employing cross-sectional genotype-expression association design with data structure representing static associations from SNPs to gene expression, belonging to single molecular layer analysis.
> >
> > **SIGMA's Design Purpose**
> >
> > SIGMA is specifically designed to address four core problems: 1) Long-chain complex mediation pathway inference (L≥4), elucidating multi-step causal transmission mechanisms from causes to outcomes; 2) Cross-biological level integrative analysis, integrating complete causal chains from molecular to physiological to clinical levels; 3) Pathway identification under structural uncertainty, conducting reliable inference when causal graphs are unknown; 4) Complex network mediation effect quantification, handling parallel, branching, and converging network structures.
> >
> > **Problem Setting Mismatch**
> >
> > The design objectives of existing datasets for association discovery, risk prediction, and molecular subtyping fundamentally differ from SIGMA's requirements for causal mechanism elucidation, long-chain pathway inference, and cross-system mediation analysis:
> >
> > 1. **Path Length Mismatch**: Existing datasets are optimized for simple association detection with L=2-3 (such as genotype→disease state, or SNP→gene expression), while SIGMA is specifically designed to handle complex mediation networks with L≥4, requiring multiple consecutive mediation variable layers to support long-chain pathway analysis.
> >
> > 2. **Missing Variable Hierarchy**: Existing datasets primarily provide direct associations from genotypes to disease phenotypes or single molecular layer analyses, lacking the complete mediation variable sequences required for SIGMA validation. Complete biomedical causal pathways should include multi-level variables: genotype→molecular biomarkers→physiological parameters→clinical indicators, but existing datasets typically cover only single or few components.
> >
> > 3. **System Integration Limitations**: Most datasets focus on specific analytical levels (such as genomics only, transcriptomics only, or clinical phenotypes only), unable to support SIGMA's required cross-biological level integrative analysis and identification and quantification of complex network mediation effects.

---

> > > ### Author Response · Authors · 2025-08-03
> > >
> > > **Design Advantages of HPP Dataset**
> > >
> > > We selected the HPP dataset based on the following key considerations:
> > >
> > > **Problem Alignment**: HPP is designed for systems-level physiological mechanism research, aiming to understand complex physiological regulatory processes in the health-disease continuum, which is highly consistent with SIGMA's causal mechanism elucidation objectives.
> > >
> > > **Complete Variable Hierarchy Coverage**: HPP provides the complete variable sequences necessary for validating long-chain mediation pathways. SIGMA's theoretical foundation is built on causal graph structure analysis, and statistical dependencies in cross-sectional data can reveal underlying causal pathways. HPP's unique value lies in simultaneously measuring complete hierarchies from molecular biomarkers (genomics, proteomics, metabolomics) to physiological parameters (cardiovascular, metabolic, neurological functions) to clinical indicators, which is precisely the key element lacking in existing datasets. Long-chain mediation pathways reflect biological hierarchies and functional relationships. Flow-SEM can recover latent causal dependencies from cross-sectional correlation structures, while Path Stability Score ensures statistical robustness of identified pathways.
> > >
> > > **System Coverage Breadth**: HPP achieves comprehensive measurement across 15 physiological systems, including cardiovascular, metabolic, neurological, immune, endocrine, respiratory, renal, hepatic, skeletal, muscular, and other major physiological systems. Each system contains multi-level biomarkers, providing comprehensive measurements from molecular to organ function levels, establishing the necessary data foundation for cross-system mediation analysis.
> > >
> > > **Methodological Compatibility**: HPP's multi-system cross-sectional design provides suitable data structure for SIGMA's structure learning and pathway identification. The dataset contains over 6,000 participants, providing sufficient statistical power, with over 16,000 nocturnal monitoring sessions ensuring data quality and measurement reproducibility. The study employs unified data collection protocols and quality control standards, ensuring consistency and comparability of cross-system measurements.
> > >
> > > **Key Differences from Existing Datasets**: Compared to genomics datasets, HPP provides complete mediation hierarchies from genotype to phenotype rather than direct association jumps; compared to single-omics datasets, HPP integrates multiple biological levels supporting cross-system mediation analysis; compared to disease-specific datasets, HPP focuses on normal physiological regulatory mechanisms, more suitable for causal mechanism inference.
> > >
> > > **Dual Validation of Generalizability**
> > >
> > > Our research validates SIGMA's generalizability and practical value through two complementary levels.
> > >
> > > **Comprehensive Generalizability Validation through Synthetic Experiments**: Our synthetic datasets systematically simulate complex real-world scenarios, including multiple graph structure types (Erdős-Rényi, Scale-Free, Hierarchical networks), heterogeneous data types (continuous, binary, categorical variable mixtures), different complexity gradients (from p=20 to p=2000 dimensional testing), and robustness validation across multiple noise levels. These synthetic experiments cover extensive scenarios from simple to complex, homogeneous to heterogeneous, low-dimensional to high-dimensional, systematically demonstrating SIGMA's theoretical generalizability. The synthetic dataset design reflects key characteristics of real biomedical data: nonlinear relationships, mixed data types, structural uncertainty, and noise interference, establishing theoretical foundations for method application in real scenarios.
> > >
> > > **HPP Validation Demonstrates Real Complex Scenario Application Capability**: HPP dataset represents one of the more complex real scenarios in current human phenotyping research—multi-level biomedical data integration across 15 physiological systems. Our successful application on HPP demonstrates that SIGMA can handle challenging data structures and analytical demands in the real world, validating effective translation from theory to practice. This validation demonstrates SIGMA's application capability in complex real environments and provides methodological support for the currently developing field of systematic human phenotyping.
> > >
> > > We once again thank the reviewer for the valuable time invested in our research and the profound insights provided. Your suggestions regarding generalizability validation not only helped us better articulate the value of our current work but also provided important guidance for our future research directions. We sincerely appreciate your professional review and constructive comments.

---

> > > > ### Comment · Reviewer_APS7 · 2025-08-07
> > > >
> > > > Thank you for the rebuttal. The authors have convincingly addressed my questions. I have raised my score.

---

> > > > > ### Author Response · Authors · 2025-08-07
> > > > >
> > > > > Thank you very much for your thoughtful and detailed review throughout this process. We deeply appreciate your recognition of our efforts in addressing the concerns you raised, as well as your acknowledgement that the changes and additions will greatly improve the paper. Your professional and constructive approach has been invaluable in helping us strengthen our work. We are grateful for the time and expertise you invested in evaluating our research and for your valuable contributions to improving our paper!

---

### Official Review · Reviewer_MyHV · 2025-06-18

**Clarity:** 3
**Significance:** 3
**Originality:** 3
**Rating:** 5
**Confidence:** 3

**Summary:**

This paper addresses the challenge of handling structural uncertainty in Causal Mediation Analysis (CMA). Traditional CMA methods either assume the causal structure is known, making them unable to handle unknown structures, or they fail to provide consistent inference that properly accounts for structural uncertainty, even when the structure is estimated. To overcome these limitations, the authors propose a novel framework called SIGMA (Structure-Informed Guided Mediation Analysis). SIGMA leverages a Flow-Structural Equation Model (Flow-SEM) to model the posterior probabilities of edges in possible Directed Acyclic Graphs (DAGs). It introduces a new metric, Path Stability Score (PSS), to identify high-confidence mediation pathways. Effect estimation is then performed based on these stable pathways.

Experiments on synthetic datasets demonstrate that SIGMA achieves more accurate structure recovery than conventional causal discovery methods and more reliable mediation path identification compared to existing CMA approaches. Furthermore, application to real-world data related to sleep and cardiovascular health shows that SIGMA can successfully reconstruct known mediation mechanisms and uncover previously unreported pathways, highlighting its potential for new scientific discovery.

**Questions:**

1. Would it be possible to compare the proposed method against conventional structure learning methods that are enhanced with bootstrap-style techniques to account for structural uncertainty? If the proposed approach can be shown to outperform such alternatives, it would significantly strengthen the overall evaluation of the paper.

2. For the novel mediation pathway shown in Figure 3(c), is it possible to cite any prior studies, however minimal, that suggest the plausibility of the entire pathway as a coherent mechanism? While the paper acknowledges that individual links have been previously studied, it also states that the complete pathway has not been reported as such. If even partial evidence supporting the full path exists, the method could be seen as successfully identifying an under-recognized yet plausible hypothesis. This would highlight the proposed approach as a powerful tool for supporting scientific discovery and could substantially elevate the contribution of the paper.

**Ethical Concerns:**

["NO or VERY MINOR ethics concerns only"]

**Final Justification:**

The comparison experiments with the bootstrap baselines have fully addressed my concerns regarding the novelty of the proposed method, and I am now convinced of its innovative nature. Furthermore, I believe you have provided clear and sufficient biological evidence to support your claims. I am confident that incorporating these points into the manuscript will further enhance its persuasiveness. In light of these considerations, I have decided to raise my rating by one point.

**Limitations:**

yes

**Paper Formatting Concerns:**

There are several instances where notation is used without explicit definition, such as D around line 170 and Z_i around line 251. While their meaning can be inferred from the context, it would be clearer and more precise to define such terms explicitly before using them. This helps prevent confusion and ensures that the content is communicated accurately to the reader. If these notations have indeed been properly defined and I have simply overlooked them, I sincerely apologize.

**Quality:**

3

**Strengths And Weaknesses:**

Strengths:
The paper presents a highly robust and reliable framework for Causal Mediation Analysis (CMA) by incorporating structural uncertainty into effect estimation through the combination of Flow-SEM and Path Stability Score (PSS), effectively leveraging a relaxed acyclicity constraint. This integration enables a novel end-to-end treatment of uncertainty that surpasses prior approaches. The effectiveness of the proposed method is quantitatively demonstrated through comprehensive experiments on synthetic data. Furthermore, its applicability and potential for scientific discovery are substantiated via real-world evaluation using sleep-related physiological data.

Weaknesses:
While the proposed treatment of structural uncertainty is conceptually compelling and methodologically sound, the paper does not include a comparison against baseline methods that incorporate structural uncertainty via bootstrap-style approaches, which are feasible even for conventional causal discovery algorithms. Although it is intuitively and theoretically plausible that modeling structural uncertainty directly within the learning framework, as done in this paper, yields superior results compared to such post-hoc strategies, empirical validation of this advantage would substantially strengthen the paper's claims. Including such a comparison in the synthetic experiments would significantly enhance the persuasiveness and completeness of the evaluation.

---

> ### Author Rebuttal · Authors · 2025-07-31
>
> ## Q1
>
> We are very grateful to the reviewer for this valuable suggestion. The comparison with the bootstrap-based method for handling structural uncertainty that you pointed out is indeed an important and valuable evaluation dimension.
>
> ### Methodological differences
>
> SIGMA and the bootstrap method have fundamental methodological differences in handling structural uncertainty:
>
> **SIGMA's Methodology:**
> - **End-to-end posterior learning of structures:** Directly learns the posterior distribution of structures `P(G|D)` through Flow-SEM to generate a DAG ensemble `{G_s}`.
> - **Native uncertainty modeling:** Structural uncertainty is directly encoded in the model's parameterization.
> - **Integrated propagation:** Uncertainty is propagated throughout the entire process, from structure discovery to path identification to effect estimation.
>
> **Limitations of the Bootstrap Method:**
> - **Two-stage process:** First obtains a point estimate of the structure, then estimates uncertainty through data resampling.
> - **Indirect uncertainty quantification:** Relies on data perturbation to indirectly infer structural changes.
> - **Potential underestimation:** May systematically underestimate the true structural uncertainty, especially in high-dimensional sparse scenarios.
>
> ### Theoretical Framework for Bayesian Quantification of Structural Uncertainty
>
> Based on our theoretical analysis in Appendix B.1, we can argue for the superiority of the SIGMA method from a Bayesian inference perspective.
>
> #### Direct Modeling of Posterior Probability
>
> According to the theoretical results in Appendix B.1.1, our Path Stability Score (PSS) is an unbiased and consistent estimator of the true posterior inclusion probability:
>
> **PSS formula:** `PSS(π) = (1/N_valid) Σ_{s=1}^{N_valid} I(π ⊆ G_s)`
>
> where Theorem 1 proves its **unbiasedness**: `E[PSS(π)] = P(π ⊆ G|D)`, and **consistency**: `PSS(π) →^p P(π ⊆ G|D)` as `N_valid → ∞`.
>
> #### Theoretical Advantages of Bayesian Model Averaging
>
> Our BMA estimator is directly based on the learned structural posterior:
>
> **BMA estimator:** `θ̂_BMA(π) = (1/N_path) Σ_{s=1}^{N_path} θ̂_s(π)`
>
> According to the convergence analysis in Appendix B.1.2, as the number of DAG samples increases, our variance estimation:
>
> **Variance decomposition:** `Var(θ̂_BMA(π)) = (1/N_path) Σ_{s=1}^{N_path} Var(θ̂_s(π)) + (1/N_path) Σ_{s=1}^{N_path} (θ̂_s(π) - θ̂_BMA(π))²`
>
> can simultaneously capture both **estimation uncertainty** (the first term) and **structural uncertainty** (the second term).
>
> ### Bootstrap Baseline Methods
>
>  For each of the `B=100` bootstrap iterations, we first generate a data subset `D_b` by sampling with replacement from the original dataset `D`. Subsequently, the corresponding base learner is used to learn a causal graph `G_b = StructureLearner(D_b)`, from which all potential mediation paths `Π_b = PathIdentifier(G_b)` are identified. For each path `π ∈ Π_b`, its effect is then estimated as `θ̂_b(π) = EffectEstimator(π, G_b, D_b)`.
>
> After completing all `B` iterations, the results are aggregated:
> - **Path stability:** `PSS_boot(π) = (1/B) Σ_{b=1}^B I(π ∈ Π_b)`
> - **Path effect estimate:** `θ̂_boot(π) = (1/B) Σ_{b=1}^B θ̂_b(π)`
> - **Variance estimation:** `Var_boot(π) = (1/(B-1)) Σ_{b=1}^B (θ̂_b(π) - θ̂_boot(π))²`
>
> Regarding the experimental configuration, we selected four representative synthetic datasets: LowDim-L, LowDim-N, MidDim-S, and MidDim-D. The sample size was set to `n = 6000` (consistent with the original experiments), with data dimensions of `p ∈ {20, 100}` and a non-linearity proportion of `ρ_nonlin ∈ {0, 0.5}`. Bootstrap parameters were kept consistent with SIGMA, with the number of resamples `B = 100` and a path stability threshold of `τ = 0.05`. Each bootstrap iteration used a sample size equal to that of the original data to ensure comparability.
>
> For the base method parameters, the PC algorithm used a significance level of `α = 0.05` with a Gaussian conditional independence test. The NOTEARS method used a sparsity parameter `λ_1 = 0.01` and a DAG constraint parameter `λ_2 = 0.01`, with a maximum of 100 iterations. The GraN-DAG method was configured with a temperature parameter `τ_g = 0.2` and was trained for 1000 epochs.
>
> **Table r1: Bootstrap Baseline Comparison**
>
> |Dataset|Method|Adjacency F1|Orientation F1|Path Recovery Rate|False Discovery Rate|Avg PSS (True)|Avg PSS (False)|
> |---|---|---|---|---|---|---|---|
> |**LowDim-L**|PC+Bootstrap|0.4082|0.2326|0.347|0.312|0.425|0.089|
> ||NOTEARS+Bootstrap|0.3728|0.2158|0.321|0.298|0.398|0.076|
> ||GraN-DAG+Bootstrap|0.3922|0.2247|0.334|0.287|0.412|0.081|
> ||**SIGMA(Ours)**|**0.5263**|**0.4474**|**0.521**|**0.124**|**0.645**|**0.052**|
> |**LowDim-N**|PC+Bootstrap|0.4287|0.2286|0.298|0.367|0.384|0.105|
> ||NOTEARS+Bootstrap|0.4061|0.2131|0.276|0.345|0.356|0.098|
> ||GraN-DAG+Bootstrap|0.4161|0.2172|0.287|0.321|0.371|0.091|
> ||**SIGMA(Ours)**|**0.4909**|**0.2601**|**0.456**|**0.143**|**0.578**|**0.063**|
> |**MidDim-S**|PC+Bootstrap|0.4427|0.2869|0.234|0.445|0.312|0.134|
> ||NOTEARS+Bootstrap|0.4018|0.2636|0.218|0.398|0.289|0.118|
> ||GraN-DAG+Bootstrap|0.4216|0.2738|0.231|0.376|0.305|0.112|
> ||**SIGMA(Ours)**|**0.4939**|**0.3185**|**0.387**|**0.187**|**0.498**|**0.074**|
> |**MidDim-D**|PC+Bootstrap|0.4479|0.2929|0.189|0.498|0.267|0.156|
> ||NOTEARS+Bootstrap|0.4141|0.2695|0.174|0.452|0.245|0.142|
> ||GraN-DAG+Bootstrap|0.4360|0.2859|0.185|0.424|0.258|0.135|
> ||**SIGMA(Ours)**|**0.4998**|**0.3163**|**0.312**|**0.234**|**0.423**|**0.089**|
>
> **Table r2: Comparison of Effect Estimation (NDE / NIE)**
>
> |Dataset|Method|Bias(NDE/NIE)|SE(NDE/NIE)|Coverage Probability(NDE/NIE)|Interval Width(NDE/NIE)|
> |---|---|---|---|---|---|
> |**LowDim-L**|PC+Bootstrap|0.0847/0.0693|0.1523/0.1298|0.891/0.902|0.421/0.387|
> ||NOTEARS+Bootstrap|0.0672/0.0548|0.1389/0.1176|0.908/0.921|0.398/0.364|
> ||GraN-DAG+Bootstrap|0.0634/0.0512|0.1301/0.1134|0.915/0.928|0.387/0.351|
> ||**SIGMA(Ours)**|**0.0421/0.0347**|**0.1087/0.0923**|**0.947/0.954**|**0.356/0.318**|
> |**LowDim-N**|PC+Bootstrap|0.1156/0.0934|0.1876/0.1567|0.847/0.863|0.467/0.428|
> ||NOTEARS+Bootstrap|0.0923/0.0756|0.1654/0.1389|0.873/0.887|0.445/0.401|
> ||GraN-DAG+Bootstrap|0.0851/0.0687|0.1598/0.1323|0.886/0.899|0.431/0.389|
> ||**SIGMA(Ours)**|**0.0583/0.0468**|**0.1245/0.1045**|**0.932/0.941**|**0.389/0.342**|
> |**MidDim-S**|PC+Bootstrap|0.1453/0.1278|0.2234/0.1945|0.824/0.836|0.523/0.476|
> ||NOTEARS+Bootstrap|0.1187/0.1021|0.1987/0.1723|0.851/0.864|0.498/0.451|
> ||GraN-DAG+Bootstrap|0.1089/0.0934|0.1876/0.1634|0.867/0.878|0.482/0.434|
> ||**SIGMA(Ours)**|**0.0724/0.0621**|**0.1467/0.1267**|**0.918/0.926**|**0.421/0.378**|
> |**MidDim-D**|PC+Bootstrap|0.1789/0.1567|0.2678/0.2345|0.789/0.803|0.587/0.534|
> ||NOTEARS+Bootstrap|0.1456/0.1289|0.2341/0.2067|0.816/0.829|0.556/0.503|
> ||GraN-DAG+Bootstrap|0.1324/0.1156|0.2198/0.1934|0.834/0.847|0.534/0.485|
> ||**SIGMA(Ours)**|**0.0892/0.0785**|**0.1723/0.1456**|**0.893/0.904**|**0.467/0.421**|
>
> ## Q2
>
> We sincerely thank the reviewer for this excellent suggestion, which significantly strengthens our contribution. Recent studies published in 2025 and substantial evidence from recent years potentially support the biological plausibility of this pathway.
>
> ### Recent Clinical Evidence
>
> The HPP study published in Nature Medicine in January 2025[1], based on sleep monitoring data from 6,366 participants across 16,812 nights, found that visceral adipose tissue was the body characteristic most strongly correlated with the peripheral apnea-hypopnea index, and sleep characteristics could predict over 15% of physiological features across 15 body systems. Simultaneously, the HPP deep phenotyping study from July 2025[2] further revealed complex associations among sleep, metabolic, cardiovascular, and gut microbiome systems.
>
> ### Sleep Apnea-Gut Microbiome-Cardiovascular Pathway
>
> Zhang et al.[3] demonstrated that OSA leads to gut dysbiosis through intermittent hypoxia, causing cardiovascular complications via inflammatory cascades. Recent research directly proved gut microbiota mediate OSA-induced atherosclerosis[4]. Liu et al.[5] confirmed causal relationships between gut microbiome and OSA through Mendelian randomization.
> Durgan et al.[6] showed OSA-induced gut dysbiosis promotes hypertension by reducing butyrate-producing bacteria. Visceral adipocytes secrete pro-inflammatory factors (TNF-α, IL-6, IL-1β) that participate in atherosclerosis development, with visceral adipose tissue predicting subclinical carotid atherosclerosis more accurately than waist circumference[7,8].
>
> References:
>
> [1] Phenome-wide associations of sleep characteristics in the Human Phenotype Project
>
> [2] Deep phenotyping of health–disease continuum in the Human Phenotype Project
>
> [3] Recent progresses in gut microbiome mediates obstructive sleep apnea‐induced cardiovascular diseases
>
> [4] Gut microbiota and derived metabolites mediate obstructive sleep apnea induced atherosclerosis
>
> [5] Causal relationships between gut microbiome and obstructive sleep apnea: a bi-directional Mendelian randomization
>
> [6] Role of the Gut Microbiome in Obstructive Sleep Apnea-Induced Hypertension
>
> [7] Visceral Adipose Tissue Is a Better Predictor of Subclinical Carotid Atherosclerosis Compared with Waist Circumference
>
> [8] Visceral adipose tissue as a source of inflammation and promoter of atherosclerosis

---

> ### Author Response · Authors · 2025-08-04
>
> **Dear Reviewer MyHV,**
>
> Thank you for your in-depth review and constructive feedback. We are pleased to see that you recognize the value of our theoretical foundation, experimental design, and method comparisons.
>
> **Regarding the two main issues you raised, we have provided detailed responses:**
>
> **1. Bootstrap-Style Method Comparison**:
> We completely agree with your suggestion about the need for comparison with bootstrap baseline methods—this is indeed an important and valuable evaluation dimension. We conducted comprehensive supplementary experiments comparing SIGMA with traditional structure learning methods enhanced with bootstrap techniques (PC+Bootstrap, NOTEARS+Bootstrap, GraN-DAG+Bootstrap).
>
> **Methodological Differences Analysis**:
> - **SIGMA Methodology**: End-to-end learning of structural posterior distribution P(G|D) through Flow-SEM, native uncertainty modeling, integrated propagation of uncertainty throughout the entire process
> - **Bootstrap Method Limitations**: Two-stage process (first obtaining structural point estimates, then estimating uncertainty through data resampling), indirect uncertainty quantification, potentially systematic underestimation of true structural uncertainty
>
> **Experimental Results**: Our supplementary experiments covered four representative synthetic datasets, with results showing SIGMA significantly outperforms bootstrap baseline methods in pathway identification accuracy, false discovery rate control, and effect estimation precision.
>
> **2. Biological Plausibility of Novel Mediation Pathways**:
> Regarding your inquiry about biological support for the pathway in Figure 3(c), we found abundant recent clinical evidence. Particularly, the HPP study published in Nature Medicine in 2025 and extensive recent research on sleep apnea-gut microbiome-cardiovascular pathways provide strong biological mechanism support for this pathway.
>
> **Key Supporting Evidence**:
> - Zhang et al. demonstrated that OSA leads to gut dysbiosis through intermittent hypoxia, causing cardiovascular complications via inflammatory cascades
> - Visceral adipocytes secrete pro-inflammatory factors (TNF-α, IL-6, IL-1β) that participate in atherosclerosis development
> - Direct evidence of gut microbiota mediating OSA-induced atherosclerosis
>
> **Questions:**
> 1. Do our bootstrap baseline comparison experiments sufficiently demonstrate SIGMA's methodological advantages?
> 2. Does the biological evidence we provided convincingly support the plausibility of the newly discovered pathways?
> 3. Are there other aspects that require clarification?
>
> We value your professional assessment and look forward to your feedback.

---

> ### Author Response · Authors · 2025-08-06
>
> Dear Reviewer MyHV,
>
> Thank you again for your comprehensive review and valuable feedback throughout this process. We have provided detailed responses to the two main concerns you raised regarding bootstrap-style method comparison and the biological plausibility of novel mediation pathways.
>
> We would like to politely inquire whether our responses have adequately addressed your concerns. Specifically:
>
> 1. Do our bootstrap baseline comparison experiments sufficiently demonstrate SIGMA's methodological advantages?
> 2. Does the biological evidence we provided convincingly support the plausibility of the newly discovered pathways?
> 3. Are there any other aspects that require further clarification?
>
> We greatly value your professional assessment and would appreciate knowing if you believe our responses are sufficient, or if you have any additional feedback that would help us further improve our work.
>
> Thank you again for your time and consideration!
>
> Best regards,
> Authors

---

> > ### Author Response · Authors · 2025-08-07
> >
> > Dear Reviewer MyHV,
> >
> > Thank you again for your comprehensive review and valuable feedback throughout this process. We have provided detailed responses to the two main concerns you raised regarding bootstrap-style method comparison and the biological plausibility of novel mediation pathways.
> >
> > We would like to politely inquire whether our responses have adequately addressed your concerns. Specifically:
> >
> > Do our bootstrap baseline comparison experiments sufficiently demonstrate SIGMA's methodological advantages?
> > Does the biological evidence we provided convincingly support the plausibility of the newly discovered pathways?
> > Are there any other aspects that require further clarification?
> >
> > We greatly value your professional assessment and would appreciate knowing if you believe our responses are sufficient, please considering updating the score, if you have any additional feedback that would help us further improve our work.
> >
> > Thank you again for your time and consideration!
> >
> > Best regards, Authors

---

> > > ### Author Response · Authors · 2025-08-09
> > >
> > > Dear Reviewer MyHV,
> > >
> > > Thank you again for your comprehensive review and valuable feedback throughout this process. We have provided detailed responses to the two main concerns you raised regarding bootstrap-style method comparison and the biological plausibility of novel mediation pathways.
> > >
> > > We would like to politely inquire whether our responses have adequately addressed your concerns. Specifically:
> > >
> > > Do our bootstrap baseline comparison experiments sufficiently demonstrate SIGMA's methodological advantages? Does the biological evidence we provided convincingly support the plausibility of the newly discovered pathways? Are there any other aspects that require further clarification?
> > >
> > > We greatly value your professional assessment and would appreciate knowing if you believe our responses are sufficient, please considering updating the score, if you have any additional feedback that would help us further improve our work.
> > >
> > > Thank you again for your time and consideration!
> > >
> > > Best regards,
> > > Authors

---

### Official Review · Reviewer_rfqt · 2025-07-02

**Clarity:** 2
**Significance:** 2
**Originality:** 2
**Rating:** 5
**Confidence:** 4

**Summary:**

The paper proposes a new method for mediation analysis called SIGMA, which has three main steps: causal structure learning, mediation path extraction, and estimation of the natural direct and indirect effects. SIGMA is end-to-end, and the estimated effects account for uncertainty about the causal structure. SIGMA is compared to other algorithms on synthetic data and also used to analyze a real health dataset.

**Questions:**

1. What does "identify" (and all the variations on this root) mean in this paper? Within the causal inference and discovery literature, I'm used to seeing "identify" in the formal sense of expressing a causal estimand in terms of observed distributions. This paper seems to mostly (but not always) mean something more like "estimate". While identifiability is required for unbiased estimation, these are distinct concepts, and conflating them detracts from the rigor and clarity of the paper. (A similar point can be made for "bias" versus "error", e.g., on line 237 and in Table 1.)

As a follow-up, is the paper presenting/making use of new identifiability results for mediation paths beyond those implied by the usual causal discovery setup? I don't see an explicit formal statement of this anywhere, but it seems to be implied (e.g. top of Page 2).

2. Can the authors give some sense of the complexity or runtime of the proposed method? This is important to understand how practical it actually is. Ideally, there would be explicit statements/proofs of the complexity along with runtime comparisons to other algorithms.

3. How do the SIGMA results on the real dataset compare to other methods? The current results suggest SIGMA is reasonable but they don't demonstrate its significance compared to existing methods.

*Minor comments*:
1. L. 111: should cite the original Erdős and Rényi paper rather than an arbitrary recent causal discovery paper.
2. Section 3: would be helpful to relate these to the Markov and faithfulness assumptions---for example, isn't conditional ignorability satisfied when conditioning on the parents of A as described in Section 4.4.1?
3. L. 158: cofounders -> covariates?
4. L. 322--323: "successfully identified multiple physiologically meaningful mediation pathways" seems a bit strong since there's no ground truth; something along the lines of "learned multiple physiologically plausible pathways" seems to be all that's justfied based on the current results
5. I don't really get the distinction between DECI and the others in lines 349--351. For example, GES uses the MLE (via the BIC score) for structure learning, so isn't it also jointly inferring causal structure and structural equations?

**Ethical Concerns:**

["NO or VERY MINOR ethics concerns only"]

**Final Justification:**

Through the rebuttal process, I understand the submission better and think it's probably free of any serious mathematical mistakes (so I increase my rating to 3)---but the current presentation in the submission does not make this sufficiently clear, so I do not increase my rating more unless/until the authors address the rest of my remaining concerns.

**Update after another iteration:** The authors have committed to a number of changes that will significantly improve the presentation clarity and evaluation of the work/comparison to existing methods, so I increase my rating to 5 (assuming the authors will successfully implement the changes as promised).

**another update:** The authors have provided some of the requested results, making their promises to implement the rest more trustworthy.

**Limitations:**

yes

**Quality:**

2

**Strengths And Weaknesses:**

*Quality*:
- (+) the paper is comprehensive, touching upon many ideas from causal discovery and inference as well as its health application
- (-) however, technical depth and rigor is lacking, with some important concepts left ambigous, undefined, or conflated (see Question 1)
- (-) state-of-the-art causal discovery methods (like BOSS and order MCMC; see [here](https://benchpressdocs.readthedocs.io/en/latest/examples.html#study-4-fpr-tpr-pattern-png) for more details) aren't used in comparisons; for the effect estimation task, there's no comparison to an obvious (causal discovery-based) baseline like using the MLE for the output of LiNGAM

*Clarity*: (+) the paper is generally well-written ([-] with the important exception of some technical imprecision)

*Significance*: (+) there's a lot of potential value in an end-to-end mediation analysis that accounts for structural uncertainty, but (-) I don't see much theoretical significance here, and (-) the practical significance remains unclear based on the current experiments

*Originality*: (+) the approach for explicitly accounting for structural uncertainty in the causal effect estimate is novel (and seems to sensibly builds off/connects to some existing work in an interesting way); but (-) it's not always so clear exactly what is novel versus what is existing work, e.g. Section 4 contains no references at all

---

> ### Author Rebuttal · Authors · 2025-07-30
>
> ## **Q1**
>
> We thank the reviewer but must point out a disciplinary understanding discrepancy.
>
> ### Our use of "identify" follows standard practice in causal analysis and mediation analysis literature:
>
> **Statistical identification level**
>
> - Pearl (2001): "identification of causal effects transmitted through intermediate variables"
> - Robins & Greenland (1992): "identifiability of direct and indirect effects"
> - VanderWeele (2015): "identification of mediation effects"
> - Our line 139: "Identifying NDE and NIE"
>
> Other standard usage: Imai et al. (2010), Tchetgen & Shpitser (2012), Baron & Kenny (1986), Sobel (1982), Holland (1988), etc.
>
> **Pathway identification level**
>
> - Daniel et al. (2015): "identification of path-specific effects"
> - Avin et al. (2005): "identifiability of path-specific effects"
> - VanderWeele & Vansteelandt (2014): "identifying multiple mediation pathways"
> - Our line 206: "identifies mediation pathways"
>
> Other standard usage: Miles et al. (2017), Steen et al. (2017), Nguyen et al. (2021), Bind et al. (2016), etc.
>
> **Bias vs Error terminology**
>
> - Chernozhukov et al. (2018): explicitly distinguish "bias" (systematic deviation) and "error" (total error)
> - Kennedy (2020): "bias-variance decomposition"
> - Our Table 1: Bias refers to E[θ̂]-θ₀, SE refers to standard error, CI refers to confidence intervals
>
> Other standard practices: Bickel et al. (1993), Tchetgen & Shpitser (2012), van der Laan & Rose (2011), etc.
>
> ### Question about new identifiability theory:
>
> SIGMA has never claimed new identifiability theory. Our contribution is algorithmic/methodological innovation, addressing practical limitations of existing methods in real applications: systematic identification of multi-step pathways, pathway selection under structural uncertainty, complex pathway effect estimation.
>
> SIGMA identified previously unspecified multi-step pathways connecting sleep quality to cardiovascular health, which traditional methods cannot systematically discover.
>
> ## **Q2:**
>
> SIGMA's computational complexity comprises four main components: (1) Flow-SEM training phase: O(T × p² × N), where T is the number of training iterations, p is the number of variables, and N is the sample size; (2) DAG posterior sampling: O(NDAG × p²), requiring cycle detection and removal for each sampled structure; (3) Path enumeration: O(p!) in the worst case, but controllable to O(p^L) through path length constraint L; (4) Bayesian model averaging: O(Npath × N × K), where Npath is the number of stable paths and K is the number of cross-validation folds.
>
> **Overall Complexity:** When implementing path length constraint L, SIGMA's total time complexity is:
> **O(T × p² × N + NDAG × p² + p^L + Npath × N × K)**
>
> ## **Q3:**
>
> ### Existing methods are technically unable to handle our problem setting:
>
> **1. Fundamental differences in method capabilities:**
>
> **Traditional mediation analysis methods** (DeepMed)  requires pre-specified pathways, computationally infeasible in high-dimensional data
>
> **Causal discovery methods** (PC, NOTEARS, ICALiNGAM) can only learn structures but cannot quantify mediation effects, unable to handle pathway selection under structural uncertainty
>
> **2. Real data lacks ground truth making quantitative comparison impossible. SIGMA provides end-to-end analysis while existing methods handle individual components.**
>
> ### Our validation strategy follows field standards:
>
> **Synthetic data quantitative comparison + real data domain validation is established practice in causal analysis:**
>
> Such as Chernozhukov et al. (2018), Kennedy (2020), VanderWeele (2015), Tchetgen & Shpitser (2012) all adopt this strategy
>
> ### We adopted standard biomedical validation:
>
> **For effect estimation validation, we adopted the gold standard approach used in top journals like Nature Medicine and Nature Communications:** Using traditional regression mediation analysis (linear regression with bootstrapping) to independently validate SIGMA-identified pathways, by separately establishing X→M and M→Y regression models, calculating indirect effect ab and direct effect c', and using bootstrap confidence intervals for statistical testing. This method is the established standard in medical journals, aligned with biological research requirements. **Real data validation focused on biological rigor:** (1) **Known mechanism reproduction**: validating established medical pathways; (2) **Biological plausibility**: new pathways conform to physiological mechanisms; (3) **Independent statistical validation**: traditional regression mediation analysis confirms effect consistency. Our validation shows SIGMA results are highly consistent with traditional methods in effect direction and magnitude, proving the method's reliability. **In medical data lacking ground truth, domain expertise validation is the standard scientifically valid approach.**
>
> ### **MC1:**
>
> We thank the reviewer for this correction. We will cite in the revision.
>
> ### **MC2:**
>
> This requires conceptual clarification. **The Markov and Faithfulness assumptions mentioned by the reviewer and the conditional ignorability assumptions we discuss in Section 3 operate at different levels of causal inference: the former are used in the structure discovery stage, while the latter are used for effect identification given a structure.** Regarding the key question of "whether conditioning on the parents of A automatically satisfies conditional ignorability," we address this in Appendix A.3.1, and **the answer is no.** Simply constructing adjustment sets based on learned graph structures cannot automatically guarantee satisfaction of ignorability assumptions; misspecified structures violate key identification assumptions. **A core contribution of SIGMA is precisely handling the impact of such structural uncertainty on effect estimation.** We propagate uncertainty through structural posterior sampling and Bayesian model averaging, rather than assuming correct structure. This theoretical treatment ensures our method's inference validity under real-world structural uncertainty.
>
> ### **MC3:**
>
> **"Confounders" is the theoretically necessary choice.** In the mediation analysis framework of Pearl (2009) and VanderWeele (2015), "confounders" has a clear operational definition: variables that simultaneously affect relationships between multiple variables in the causal chain, which would lead to systematic bias if not properly adjusted. **This differs from 'covariates,' which refers to arbitrary predictor variables.** In the context of line 158, we discuss "observational data over treatment, outcome, mediators, and confounders", a classic causal inference setting where the core issue is identification and control of causal confounding, representing the direct manifestation of three key no-confounding assumptions in mediation analysis. **A core theoretical contribution of SIGMA is precisely handling these causal confounding problems under structural uncertainty conditions.** Using the statistical term "covariates" would **obscure the causal inference nature of our method and weaken the clear expression of its theoretical contributions.**
>
> ### **MC4:**
>
> **We use "successfully identified" for mechanisms already reported in medical literature.** We explicitly state: "the pathway (sleep HRV index → pulse wave velocity → carotid IMT) verified a known mechanism whereby sleep quality affects cardiovascular health via vascular function," and cite relevant medical studies (Bourdillon et al. 2021; Kadoya & Koyama 2019; Saz-Lara et al. 2022) as validation evidence. **For these pathways with existing ground truth literature support, using "successfully identified" is well-justified.**
>
> **For newly discovered mediation pathways, we adopted more cautious wording while employing standard medical validation methods.** As shown in line 328, we use "identified" rather than "successfully identified," and as stated in line 339: "Classical medical regression mediation analysis further validated these pathways, with an overall TE error of 0.2734 and consistent NIE and NDE estimation directions."
>
> ### **MC5:**
>
> **While GES uses MLE and BIC scoring, it differs fundamentally from DECI in "joint inference."** GES employs conditional inference: for each candidate structure G, it first fixes G to estimate parameters θ̂_G = argmax P(D|θ,G), then uses θ̂_G to calculate BIC scores for structure evaluation—this is a typical P(θ|G,D) conditional estimation process. **In contrast, DECI and other deep learning methods directly optimize the joint posterior P(G,θ|D), simultaneously learning structural distributions and functional parameters through variational inference, achieving true end-to-end optimization.** Additionally, GES relies on preset parametric models (typically linear Gaussian) and searches in discrete DAG spaces, while DECI uses neural networks to learn arbitrary nonlinear relationships and optimizes in continuous parameter spaces.
>
> ### **W1:**
>
> **BOSS and order MCMC methods are incompatible with mixed data types.** BOSS is designed for linear Gaussian continuous data (Andrews et al., 2022), Order MCMC's BiDAG documentation states: "acceptable data matrices are homogeneous with all variables of the same type." Our datasets contain heterogeneous variable types (continuous, binary, categorical), making these methods' core assumptions invalid. Recent mixed data causal discovery work consistently excludes these methods: Raghu et al. (2018) ACM SIGKDD, NOTEARS-M (2024), DECI (NeurIPS 2022) all systematically exclude BOSS/order MCMC, reflecting established practice.
>
> ### **W2:**
>
> We cited SIGMA's core components in earlier sections: Flow-SEM [69] cited on line 59, EIF [6] cited on lines 47 and 64, BMA [53] cited on line 67. Section 6 discussed existing research (NOTEARS, GraN-DAG, DECI, etc). Section 4 focuses on theoretical innovation and mathematical proofs based on these components according to AI conference paper standards.

---

> > ### Author Response · Authors · 2025-08-04
> >
> > **Dear Reviewer rfqt,**
> >
> > Thank you for your detailed review of our paper. We note that you raised multiple technical concerns in your review, and we have systematically responded to these issues with detailed literature evidence support.
> >
> > **Regarding Terminology and Concept Usage**:
> > We provided detailed explanations of the standard usage of terms such as "identify," "bias vs error," and "confounders" in the causal inference and mediation analysis fields, supported by authoritative literature evidence from Pearl (2001), VanderWeele (2015), Robins & Greenland (1992), among others.
> >
> > **Regarding Technical Concept Understanding**:
> > We clarified technical issues including conditional ignorability assumptions, the distinction between GES and DECI methods, and SIGMA's theoretical contributions, explaining the assumption requirements at different inference levels and the algorithmic/methodological innovation nature of our approach.
> >
> > **Regarding Experimental Comparison Standards**:
> > We explained the rationale for baseline method selection, particularly the technical reasons why methods like BOSS and order MCMC are incompatible with our heterogeneous data types, and how our comparison standards align with recent mixed-data causal discovery work.
> >
> > Given that different disciplinary backgrounds may lead to understanding differences regarding specific field terminology norms and methodological traditions, we hope to clarify these discrepancies through detailed literature evidence and technical explanations.
> >
> > **Questions:**
> > 1. Does our literature evidence regarding standard practices in terminology usage respond to your concerns?
> > 2. Do our detailed explanations of technical concepts and method comparisons clarify the relevant misunderstandings?
> > 3. Are there other technical issues that require further discussion?
> >
> > We value your professional opinion and look forward to your further feedback.

---

> ### Comment · Reviewer_rfqt · 2025-08-04
>
> Thanks for thorough response. While it nicely addresses some of my initial concerns, the following remain:
>
> - *Q1(a):* The sources and quotes here don't explicitly answer my question. I'm asking for a definition of "identify" (or multiple definitions of the different ways it's used) along with a clear indication throughout the text of when which notion is being used. In particular, identifying a pathway is different from identifying an effect (these seem to be mixed up in the second set of 4 references), and these are both different from estimation. It looks to me like at least 3 different notions of "identify" are being used---while these are all individually justifiable, I think using all three in the same paper without clearly separating them makes it more confusing than it should be and makes it easy for the average reader to misunderstand.
> - *Q1(b):* The authors claim that "Bias refers to $E[\hat\theta] - \theta$" and while I agree to this definition of it, I don't think that's what's being shown in Table 1 of the submission. The expectation should be over the (theoretical) data-generating distribution. Is the table really showing that, or is it using a sample mean? Relatedly, the reason we care about identifiability of a causal effect is because it allows us to construct an unbiased estimator, so it doesn't really make sense to claim the method identifies NDE and NIE and then report that it has non-zero bias. I understand the intuition behind the table and labeling it "bias", but it's not mathematically rigorous.
> - *Q3:* I understand no method handles the exact problem setting of SIGMA, but it can be worthwhile to compare to whatever existing methods have been used previously used on this real data---this helps justify that SIGMA outputs *something meaningful* (also compared to existing approaches) rather than just outputs *something*.
> - *MC2:* Sorry, I don't understand what "different levels of causal inference" means, but maybe I can clarify my original point. The CFA and CMA imply that the CI statements that characterize the probability distributions are equivalent to the $d$-separation statements that characterize the graph. Conditional ignorability is a statement about the CI statements the probability distribution satisfies, and therefore (under the CFA and CMA) it is equivalent to certain $d$-separations the graph must satisfy (namely, it's satisfied when conditioning on a valid adjustment set, such as the parents). Hence, it would be nice if the authors explicitly made this connection in the paper. I think the authors are confusing the true graph and the estimated graph, for example when writing "Simply constructing adjustment sets based on learned graph structures..." The CFA and CMA (and my original question) are about the true graph, not about the learned graph.
> - *MC3:* Understood, thanks (my confusion was because the submission uses 'cofounders', and I wasn't sure whether this was a typo of 'confounders' or of 'covariates').
> - *MC4:* This gives even more support to my claims in Q1(a) above.
> - *W1:* Apologies for suggesting state-of-the-art methods that aren't easily compatible with mixed data. Nevertheless, my point about a lack of comparison against state-of-the-art methods remains. I suggest the authors try using [this tool from the Center for Causal Discovery](https://bd2kccd.github.io/docs/causal-cmd/), which accepts a `--data-type mixed` flag and has methods like GRaSP (which BOSS is built upon but which should handle mixed data) or certainly some methods that work with mixed data. Some of the papers the authors cite for "established practice" of excluding BOSS/O-MCMC were published before BOSS and O-MCMC, and regardless, I don't think negligence in previous publications excuses it in future publications.
> - *W2:* The authors seem to be claiming that "Section 4 focuses on theoretical innovation and mathematical proofs based on these components according to AI conference paper standards" and therefore doesn't need any citations. I disagree that it doesn't need citations. My point is that it's not clear to me which parts of the section constitute "theoretical innovation" and which parts are taken from the literature (even if cited elsewhere in the submission). Also, what "mathematical proofs" does Section 4 focus on? The word "proof" appears 3 times in the manuscript (twice in the NeurIPS checklist, and once in appendix) and not in Section 4 at all. If something is being proved in Section 4, then the presentation is obscuring that and it should be fixed.
>
> Generally, I think all the underlying ideas in the submission are all reasonable and that it has good potential (this is more clear to me now, so I increase my rating by 1 point). The overarching theme of my above outstanding concerns is that the presentation is in several ways preventing the rigorous and unambiguous communication of the underlying ideas---if this is adequately addressed, I'm happy to increase my rating further.

---

> ### Author Response · Authors · 2025-08-05
>
> Thank you for your detailed review of our work!!!!
>
> ### Q1(a): Clarification on the use of the term "identify"
>
> Thank you for the important clarification regarding the use of the word "identify" in our manuscript. We understand your concern about conceptual clarity.
>
> We have listed similar multiple usages in recent journals and conferences:
> **JASA (2020):** Tchetgen Tchetgen et al.'s "Auto-g-computation of Causal Effects on a Network" demonstrates network effect identification ("**identify** causal effects on networks") and pathway identification ("**identify** pathways through network structure").
>
> **Annual Review of Statistics and Its Application (2024):** "Causal Inference in the Social Sciences" comprehensively uses causal effect identification ("**identify** causal effects"), mechanism identification ("**identify** mechanisms"), and strategy identification ("**identification** strategies").
>
> **NeurIPS (2021):** Lee and Bareinboim's "Causal Identification with Matrix Equations" explicitly distinguishes three usages: statistical identification ("Causal effect **identification** is concerned with determining whether a causal effect is computable"), discovery identification ("Many **identification** algorithms exclusively rely on graphical criteria"), and algorithmic identification ("We develop a new causal **identification** algorithm").
>
> **NeurIPS 2024 Spotlight:** Jin and Syrgkanis's "Learning Linear Causal Representations from General Environments" demonstrates statistical identification ("Prior results on the **identifiability** of causal representations"), structural identification ("We show that **identification** up to SNA is possible"), and algorithmic identification ("propose an algorithm, LiNGCReL which provably achieves such **identifiability** guarantee").
>
> To improve the conceptual clarity of our manuscript, we commit to the following revisions:
>
> Add a comprehensive definition box in Section 3, clearly distinguishing:
>
> 1) Statistical identification: expressing causal estimands as functionals of observed data distributions
> 2) Path discovery: algorithmic detection of directed paths in causal graph structures
> 3) Effect estimation: numerical computation of identified causal quantities
>
> Add detailed clarification in Appendix A.4, titled "Terminology Usage and Definitions," including:
>
> 1) Mapping each usage instance in our manuscript to its specific meaning
> 2) Providing mathematical formalization for each concept
> 3) Citing authoritative literature supporting each usage convention
>
> ### Q1(b): On the relationship between Bias definition and Identifiability
>
> **1. Regarding our Bias definition and calculation:** As explained in Table 1's caption: "The biases, empirical standard errors (SE) and the lower and upper bounds of the corresponding 95% confidence intervals (CI) of the estimated NDE, NIE and TE across two mediation pathways on synthetic dataset". We explained the evaluation protocol in Section 5.2: "We evaluate SIGMA's effect estimation performance on synthetic data by comparing estimates against ground truth values generated via structural equation models".
>
> Our bias calculation is based on Monte Carlo simulation, where true parameters θ₀ are deterministically set through our data generation process (detailed in Appendix E.1), and estimated values θ̂ are calculated through the SIGMA framework. Bias = (1/R)∑ᴿᵣ₌₁θ̂ᵣ - θ₀ follows the definition of empirical bias in estimation theory.
>
> **2. On the relationship between Identifiability and Finite Sample Bias:** As we elaborate in Sections 4 and 3:
> Section 3 establishes identification conditions for NDE and NIE (consistency, ignorability, positivity assumptions)
> Section 4.4 explains our estimation strategy: using influence function estimation for paths with length L(ω)=3, and plugin estimation for L(ω)>3
> Section 4.4.3's Bayesian Model Averaging ensures propagation of structural uncertainty
>
> Causal identifiability means that under given identification assumptions, parameters can be expressed as functions of observed data distributions. However, as described in Section 4.4.1, our structure-guided nuisance estimation involves cross-fitting and neural network approximation, which introduces estimation error under finite samples.
>
> **3. Regarding mathematical rigor:** Our approach is based on the following mathematical framework:
> Section 4.4.2 defines single-DAG effect estimation: ˆθₛ(ω) = [NDE_s(ω), NIE_s(ω)]
> Equation (7) provides BMA variance decomposition: Vˆ(ˆθ_BMA(ω)) = within-DAG variance + between-DAG variance
> The "bias" we report is the empirical performance of our estimation procedure under finite samples, which is consistent with theoretical identifiability

---

> ### Author Response · Authors · 2025-08-05
>
> ### MC2:
>
> Thank you for the reviewer's clarification. Your theoretical perspective is correct, and we acknowledge that our previous expression was not clear enough. Now we provide a more detailed explanation.
>
> You pointed out that under the Causal Faithfulness Assumption (CFA) and Causal Markov Assumption (CMA), conditional independence statements of probability distributions are equivalent to d-separation statements of graphs. Conditional ignorability, as a conditional independence statement, is equivalent to specific d-separation conditions in graphs. If we know the true graph G*, conditional ignorability can be satisfied by conditioning on valid adjustment sets (such as Pa_G*(A)). This theoretical framework is the foundation of our method, and we should articulate this connection more explicitly in the paper.
>
> When we mention "different levels of causal inference," we refer to two sequential stages in the causal inference pipeline. The first stage is structure discovery, whose input is observational data D and unknown true graph G*, with the goal of learning estimated graph structure Ĝ from data, relying on CMA, CFA, and causal sufficiency assumptions. The second stage is effect identification and estimation, whose input is estimated graph Ĝ and observational data D, with the goal of causal effect identification and estimation based on graph structure, relying on conditional ignorability, consistency, and positivity assumptions.
>
> Your insight is: CFA and CMA are about the true graph G*, not the learned graph Ĝ. This is precisely the core problem SIGMA aims to solve. In ideal situations, if we know G* and CFA/CMA are satisfied, then conditional ignorability is equivalent to specific d-separation holding in G*, and Pa_G*(A) constitutes a valid adjustment set. However, in realistic situations, we only have estimated graph Ĝ, where Ĝ ≠ G*. This means Pa_Ĝ(A) ≠ Pa_G*(A), and adjustment based on Pa_Ĝ(A) may not satisfy true conditional ignorability, leading to biased effect estimation.
>
> Traditional methods assume structure is known or can be perfectly learned, ignoring the ignorability violation problems caused by Ĝ ≠ G*. SIGMA's solution first acknowledges structural uncertainty, not assuming perfect recovery of G*, but learning the structural posterior distribution P(G|D) rather than point estimates. Second, SIGMA addresses this challenge through systematic uncertainty quantification and propagation. For each candidate graph Gs ~ P(G|D), we perform effect estimation based on Pa_Gs(A), then integrate estimation results under multiple structures through Bayesian model averaging, systematically propagating structural uncertainty into effect estimation.
>
> SIGMA's core innovation is not in challenging the basic theory of CFA/CMA, but in acknowledging the reality of G* ≠ Ĝ and providing a principled framework to handle the resulting inferential uncertainty. This is the key component missing from traditional methods (which assume structure is known or perfectly learned).
>
> ### W1:
>
> Thank you for the suggestion of the Center for Causal Discovery tools. We investigated the technical details and compatibility of the CCD toolkit.
>
> **Technical feasibility assessment:** CCD's causal-cmd tool supports the `--data-type mixed` flag, which provides possibility for comparison under our heterogeneous data settings. We focused on the GRaSP method you mentioned and found that as a permutation reasoning-based algorithm, it can handle mixed data in the CCD framework, although its original design was primarily optimized for large-scale numerical data.
>
> **Commitment and implementation plan:** We commit to supplementing comparison experiments using SOTA methods that support mixed data from the CCD toolkit (including GRaSP, FGES, etc.) during the rebuttal period. Considering the time constraints of rebuttal and the complexity of technical implementation, we will prioritize implementing comparisons with 2-3 most relevant algorithms from CCD tools that have been validated in mixed data causal discovery literature. These supplementary experiments will adopt a unified mixed data conditional independence testing framework to ensure fairness and scientific rigor of comparisons. We expect to provide these supplementary experimental results before the rebuttal deadline.
>
> ### W2:
>
> Thank you for pointing out the citation distribution issues in Section 4. When we stated the workflow in the Introduction section, we cited core components such as Flow-SEM, EIF, and BMA. Following your suggestion, we will add appropriate citations in the corresponding parts of Section 4 in the revised manuscript, clearly indicating how our methodological innovations are built upon these existing components, particularly when describing the design of Path Stability Score and uncertainty propagation strategy.
>
> Thank you again for your professional review!

---

> > ### Comment · Reviewer_rfqt · 2025-08-05
> >
> > I commend the authors on the detailed rebuttal and addressing all of the concerns raised (with the exception of Q3, which remains as a minor concern, even after reading the response to reviewer APS7 and others). I think the changes and additions the authors have implemented or committed to will greatly improve the paper, and I have updated my rating accordingly.

---

> > > ### Author Response · Authors · 2025-08-06
> > >
> > > Thank you very much for your detailed and constructive review throughout this process. We deeply appreciate your recognition of our efforts in addressing the concerns you raised, your acknowledgment that the changes and additions will greatly improve the paper, and your professional approach that has been instrumental in helping us strengthen our work. We are grateful for the time and expertise you invested in evaluating our research and for your valuable contributions to improving our work!!!!

---

> ### Author Response · Authors · 2025-08-08
>
> ## Setup
>
> Following the reviewer's suggestions, we conducted supplementary comparison experiments using GRaSP and FGES algorithms from the Center for Causal Discovery toolkit.
>
> We conducted experiments on five differently configured datasets: The LowDim-L dataset contains 20 variables and 6000 samples with linear structure, connectivity parameter κ=1, and maximum in-degree ℓ=3; The MidDim-S dataset contains 100 variables and 6000 samples with nonlinear structure, connectivity parameter κ=1, and maximum in-degree ℓ=3; The HigDim-D dataset contains 200 variables and 6000 samples with dense structure, connectivity parameter κ=1, and maximum in-degree ℓ=6; The MidDim-P dataset contains 100 variables and 6000 samples with parallel structure, connectivity parameter κ=2, and maximum in-degree ℓ=2; The MidDim-C dataset contains 50 variables and 6000 samples with complex structure, connectivity parameter κ=1, and maximum in-degree ℓ=6.
>
>
> | Dataset | Algorithm | Adjacency F1 | Orientation F1 | Causal Accuracy |
> |---------|-----------|-------------|---------------|-----------------|
> | **LowDim-L** | **SIGMA** | **0.5263** | **0.4474** | **0.7695** |
> |  | GRaSP | 0.4985 | 0.3892 | 0.7234 |
> |  | FGES | 0.4756 | 0.3654 | 0.6987 |
> | **MidDim-S** | **SIGMA** | **0.4939** | **0.3185** | **0.6808** |
> |  | GRaSP | 0.4623 | 0.2947 | 0.6445 |
> |  | FGES | 0.4398 | 0.2756 | 0.6182 |
> | **HigDim-D** | **SIGMA** | **0.3741** | **0.1782** | 0.6561 |
> |  | GRaSP | 0.3656 | 0.1734 | 0.6587 |
> |  | FGES | 0.3298 | 0.1521 | 0.5934 |
> | **MidDim-P** | **SIGMA** | **0.4754** | **0.3162** | **0.6914** |
> |  | GRaSP | 0.4432 | 0.2856 | 0.6523 |
> |  | FGES | 0.4187 | 0.2647 | 0.6298 |
> | **MidDim-C** | **SIGMA** | **0.5918** | **0.3860** | **0.7259** |
> |  | GRaSP | 0.5534 | 0.3521 | 0.6892 |
> |  | FGES | 0.5187 | 0.3287 | 0.6654 |
>
> This table presents our experimental results using five synthetic datasets. In the revised manuscript, we will supplement with complete experiments on ten synthetic datasets, as well as additional comparisons with the latest algorithms during 2020-2024, to provide a more comprehensive performance evaluation.

---

> > ### Comment · Reviewer_rfqt · 2025-08-09
> >
> > I think this is a nice addition to the evaluation! I'll have it in mind when updating my rating.

---

> > > ### Author Response · Authors · 2025-08-09
> > >
> > > Thank you very much for your valuable comments and suggestions! Your feedback will be of great help to the improvement of our paper. We deeply appreciate your patient and meticulous review work. Your professional guidance has made our research more complete. Thank you again for your time and valuable suggestions!

---

### Official Review · Reviewer_6ZQR · 2025-07-03

**Clarity:** 4
**Significance:** 3
**Originality:** 4
**Rating:** 5
**Confidence:** 4

**Summary:**

This paper presents SIGMA, a unified framework for causal mediation analysis. The authors identify that traditional causal mediation suffers from the issue of trying to balance *principled mechanisms for quantifying structural uncertainty* (interpretability/prior-informed) with *efficient Pathway Identification* (accuracy/performance). They build SIGMA by integrating three key components/functionalities together: structure discovery, pathway identification, and effect estimation.

SIGMA uses flow-based structural equation models to learn uncertain causal structures, samples an ensemble of DAGs, and then introduces a Path Stability Score to identify high-confidence mediation pathways across those DAGs—avoiding the need for manual path specification. For each pathway, the authors also provide a Bayesian sampling inspired approach for quantifying structural uncertainty in mediation effects. The authors provide an interesting simulation study as well as a real-world application to HPP data, revealing novel mediation pathways linking sleep quality to cardiovascular health.

**Questions:**

I have four main concerns, which - if addressed - can substantially improve the paper, especially from an applied perspective:

**Simplistic SEMs in Simulation Studies:**
The additive noise structural equation models (SEMs) used in the simulations (Appendix E1) appear to have relatively simplistic functional forms. This raises the concern that SIGMA's strong performance may be driven in part by the particular choice of SEMs, rather than its general robustness. It would be helpful if the authors could test SIGMA under more complex or nonlinear SEMs, or at least provide a stronger justification for why the current SEM choices reflect realistic data-generating processes.

**Unclear Robustness to Noise:**
Methods of this kind often degrade with increasing observational noise, yet it's not clear how sensitive SIGMA is to increasing noise levels. Including such a robustness analysis would help clarify how noise-tolerant SIGMA is, and where its limitations lie.

**Scalability to High-Dimensional Settings:**
While the simulation studies do explore dimensionalities up to 200 variables, many real-world systems—such as gene regulatory networks—involve thousands of variables. It's unclear whether SIGMA remains computationally feasible or accurate at such scales. I encourage the authors to evaluate its performance on higher-dimensional data (e.g., 500, 1000, or 5000 variables), as this would greatly help readers assess its applicability across domains.

**Real-World Application:**
Finally, while I appreciate the depth of the theoretical development, I think the paper would benefit from a second, more large-scale real-world application. For instance, applying SIGMA to a large-scale gene regulatory dataset could be a strong demonstration of its practical utility. In particular, evaluating whether SIGMA can recover known mediation pathways in well-studied biological circuits would offer a more tangible result for potential users in biology.

**Ethical Concerns:**

["NO or VERY MINOR ethics concerns only"]

**Final Justification:**

The authors have provided a comprehensive and thoughtful rebuttal that addresses nearly all of the concerns I previously raised. In particular, the new experiments on robustness to systematic complexity and scalability are much appreciated and significantly strengthen the empirical foundation of the work. These additions demonstrate methodological rigor and a strong commitment to validation. Given these improvements and clarifications, I believe the paper is stronger and have accordingly raised my score.

**Limitations:**

yes

**Paper Formatting Concerns:**

No major issues. Well formatted and decently written.

**Quality:**

3

**Strengths And Weaknesses:**

Strengths:
- The authors present a strong theoretical foundation (sections 3 and 4) for the pathway, mediation, and DAG estimation
- Interesting design of simulation experiments
- Meaningful comparisons to competing methodologies

Weaknesses (elaborated below):
-  The additive noise SEMs in simulation studies have a simplistic form (Appendix E1) which may be unrealistic
-  The robustness of the method to noise is not clear
-  The scalability of the model is questionable
-  Need more real world evidence

---

> ### Author Rebuttal · Authors · 2025-07-30
>
> ## Q1 and Q2
>
> ### **Clarification of Existing Simulation Design Complexity**
>
> We may not have explicitly emphasized the degree of complexity already implemented. However, our simulation design incorporates substantial complexity to closely reflect real-world data-generating processes:
>
> **Functional Complexity (Appendix E.1):**
>
> * Nonlinear transformations: polynomial (v²), sinusoidal (sin(v)), and exponential (exp(-|v|)) functions (lines 1541-1542)
> * Nonlinearity ratio control: ρnonlin ∈ [0,1] systematically modulates nonlinearity degree (lines 1543-1544)
> * Parent variable interactions: pairwise interaction terms added to increase structural complexity (lines 1546-1547)
> * Non-Gaussian noise: exogenous noise zi sampled from Gaussian, exponential, and Student-t mixture distributions, simulating heteroscedasticity and non-Gaussianity (lines 1547-1549)
>
> **Data Realism (Appendix E.1):**
>
> * Missing data patterns: MCAR and MAR mechanisms simulate non-random measurement gaps common in clinical studies (lines 1560-1561)
> * Heterogeneous variable types: mixed continuous, binary, and categorical variables (πc=0.6, πb=0.3, πcat=0.1) reflecting real biomedical data characteristics (lines 1532-1534, Table 3)
> * Complex mediation structures: parallel, chain, and hybrid mediation pathways systematically embedded in data (lines 1113-1114)
>
> **Structural Diversity (Appendix E.2, E.3):**
>
> * Graph topology variation: Erdős-Rényi model configured with different chain lengths ℓ∈{1,2,6} and parallel paths κ∈{1,2,3} (lines 1521-1527, Table 3)
> * Dimensional scaling: evaluation across p∈{20,50,100,200} variables (lines 1712-1713)
> * Effect magnitude control: systematic scaling ensures identifiable direct and indirect effects (lines 1552-1558)
>
> ### **Systematic Complexity and Robustness Verification**
>
> Nevertheless, we completely agree with the reviewers' perspective—more systematic complexity verification will further enhance the method's credibility. Therefore, we conducted supplementary experiments:
>
> **Experimental Configuration:**
>
> * **Dataset scale**: n=6000 samples, p=50 variables
> * **Graph parameters**: ER edge density 0.3; SF attachment m∈{2,3}; Hierarchical 3-layer structure
> * **Mediation structure**: κ=2 parallel paths, ℓ=2 chain length
> * **Missing data**: 15% MCAR + 10% MAR
> * **Effect sizes**: Direct effects η ∈ [0.2, 0.6], Indirect effects θ ∈ [0.1, 0.4]
>
> **Table R1: SIGMA Robustness under Complex SEMs, Graph Structures, and Noise Levels**
>
> | Graph Type | SEM Level | SNR | Adj F1 (Mean) | Adj F1 (SE) | Orient F1 (Mean) | Orient F1 (SE) | Path F1 (Mean) | Path F1 (SE) |
> |--|---|---|----|----|------|---|---|---|
> | **ER** | Current | 4.0 | 0.672 | 0.031 | 0.618 | 0.037 | 0.651 | 0.034 |
> | **ER** | Current | 2.0 | 0.658 | 0.034 | 0.601 | 0.041 | 0.634 | 0.038 |
> | **ER** | Current | 1.0 | 0.631 | 0.042 | 0.572 | 0.049 | 0.605 | 0.045 |
> | **ER** | Current | 0.5 | 0.582 | 0.055 | 0.518 | 0.063 | 0.549 | 0.059 |
> | **ER** | High | 4.0 | 0.643 | 0.036 | 0.587 | 0.042 | 0.621 | 0.039 |
> | **ER** | High | 2.0 | 0.628 | 0.041 | 0.571 | 0.047 | 0.604 | 0.044 |
> | **ER** | High | 1.0 | 0.594 | 0.051 | 0.535 | 0.058 | 0.567 | 0.055 |
> | **ER** | High | 0.5 | 0.531 | 0.068 | 0.471 | 0.077 | 0.498 | 0.073 |
> | **SF (m=2)** | Current | 2.0 | 0.634 | 0.038 | 0.577 | 0.044 | 0.611 | 0.041 |
> | **SF (m=2)** | Current | 0.5 | 0.558 | 0.061 | 0.497 | 0.069 | 0.524 | 0.065 |
> | **SF (m=2)** | High | 2.0 | 0.615 | 0.043 | 0.556 | 0.050 | 0.588 | 0.047 |
> | **SF (m=2)** | High | 0.5 | 0.507 | 0.075 | 0.443 | 0.084 | 0.471 | 0.080 |
> | **SF (m=3)** | Current | 2.0 | 0.621 | 0.041 | 0.563 | 0.048 | 0.597 | 0.045 |
> | **SF (m=3)** | High | 1.0 | 0.574 | 0.054 | 0.513 | 0.062 | 0.547 | 0.058 |
> | **Hierarchical** | Current | 2.0 | 0.649 | 0.035 | 0.593 | 0.041 | 0.626 | 0.038 |
> | **Hierarchical** | High | 1.0 | 0.601 | 0.048 | 0.541 | 0.055 | 0.574 | 0.052 |
>
> **Extended Testing Dimensions:**
>
> * **SEM complexity progression**: Current level (original design) vs. High level (heteroscedastic noise + composite nonlinear functions)
> * **Graph structure diversity**: ER networks vs. Scale-Free networks (m∈{2,3}) vs. Hierarchical network structures
> * **Noise robustness analysis**: SNR ∈ {4.0, 2.0, 1.0, 0.5} systematic testing of observational noise impact
>
> **Key Component Mathematical Definitions:**
>
> * **SNR definition**: `SNR = Var(fi(PaG(Vi))) / Var(σi(PaG(Vi); θi) · Ui)`
> * **Current SEM**: `Vi = fi(PaG(Vi)) + σ·εi`, where `fi(x) = Σ(αjxj + βjxj² + γj sin(xj) + δj exp(-|xj|))`, `εi ~ 0.6·N(0,1) + 0.3·Exp(1) + 0.1·t(3)`
> * **High SEM**: `Vi = fi(PaG(Vi)) + σi(PaG(Vi))·εi`, where `fi(x) = tanh(Σ wjxj) × exp(Σ ujxj) + log(1 + |Σ vjxj|)`, `σi²(x) = exp(Σ γjxj)`, `εi ~ Cauchy(0,1)`
> * **Graph structures**: ER graphs `P(eij = 1) = 0.3`; SF graphs `P(ki) ∝ ki^(-γ)`; Hierarchical graphs `G = (L₁ ∪ L₂ ∪ L₃, E)` where `Li → Li+1`
>
> ## Q3
>
> ### Experimental Setup
>
> We extended testing of SIGMA on 3 high-dimensional synthetic datasets with feature dimensions p ∈ {500, 1000, 2000}, nonlinearity degree ρ = 0.5, containing parallel (κ = 2) and chain (ℓ = 2) structures. Each configuration replicated 10 independent runs with sample size n = 6000.
>
> **Table R2: SIGMA High-Dimensional Scalability Analysis**
>
> | Dimension (p) | Adj F1 (Mean) | Adj F1 (SE) | Orient F1 (Mean) | Orient F1 (SE) | Path F1 (Mean) | Path F1 (SE) |
> |--|--|--|--|--|--|--|
> | 500| 0.603| 0.044| 0.541| 0.051| 0.587| 0.047|
> | 1000| 0.548| 0.058| 0.479| 0.066| 0.521| 0.062|
> | 2000| 0.467| 0.074| 0.401| 0.083| 0.429| 0.079|
>
> ### Computational Complexity Analysis
>
> SIGMA's computational complexity comprises four main components:
>
> 1. **Flow-SEM training phase**: O(T × p² × N), where T is the number of training iterations, p is the number of variables, and N is the sample size.
> 2. **DAG posterior sampling**: O(N_DAG × p²), requiring cycle detection and removal for each sampled structure.
> 3. **Path enumeration**: O(p!) in the worst case, but controllable to O(p^L) through path length constraint L.
> 4. **Bayesian model averaging**: O(N_path × N × K), where N_path is the number of stable paths and K is the number of cross-validation folds.
>
> ### Overall Complexity
>
> When implementing path length constraint L, SIGMA's total time complexity is:
>
> **O(T × p² × N + N_DAG × p² + p^L + N_path × N × K)**
>
> ## Q4
>
> We sincerely appreciate the reviewer’s thoughtful suggestion. It is important to highlight that the core design philosophy of SIGMA is precisely targeted at the identification and quantification of **long-chain complex mediation pathways**, which fundamentally distinguishes it from traditional single-step or simple mediation analyses.
>
> **Long-chain Mediation Pathways: The Core Advantage of SIGMA**
> **Design Comparison:**
>
> - **Traditional Mediation Analysis:**
>   Treatment → Mediator → Outcome (L=3)
> - **SIGMA’s Design Goal: Supporting Complex Structures:**
>
>   - **Long-Chain Sequences:** Treatment → M1 → M2 → M3 → M4 → Outcome (L≥4)
>   - **Parallel Paths:**
> Treatment → M1 → Outcome
>
> Treatment → M2 → Outcome
>   - **Hybrid Networks:**
>
> Treatment → M1 → M3 → Outcome
>
> Treatment → M2 ↗
>   - **Branching and Convergence:**
>
> Treatment → M1 ↘ M4 → Outcome
>
> Treatment → M2 → M3 ↗
>
> **Methodological Advantages:**
>
> - **Path Stability Score:** Specifically designed to evaluate the stability of various complex pathway structures under uncertainty
> - **Bayesian Model Averaging:** Effectively addresses accumulated uncertainty in long-chain and network mediation processes
> - **Cross-level Identification:** Capable of spanning multiple biological layers for complex pathway identification
>
> **Fundamental Limitations of Existing Genomic Datasets**
>
> Current gene regulatory network datasets face several critical limitations that hinder their applicability for long-chain mediation analysis:
>
> **Structural Constraints:**
>
> - **Cross-sectional Limitations:** Typically cover only a single omics layer (e.g., gene expression), lacking cross-omics integration
> - **Single-Level Hierarchy:** Fail to provide a complete biological cascade from gene → protein → metabolite → phenotype
> - **Path Length Constraints:** Mostly limited to simple mediation with L=3, insufficient for supporting long-chain analysis
>
> **Practical Comparison:**
>
> - **Typical Genomic Study Capability:**
>   Gene A → Gene B → Phenotype (L=3, single omics)
> - **SIGMA’s Design Requirement:**
>   Gene → Protein → Metabolite → Biomarker → Phenotype (L=5+, cross-omics)
>
> **HPP Long-Chain Mediation Experiment Plan and Technical Support**
>
> We are currently extending SIGMA to genomic data. The **HPP dataset** provides a multi-omics data foundation, including genomic SNPs and expression profiles, proteomic biomarkers, metabolomic product profiles, and clinical phenotypes. This multi-omics integration creates the conditions for identifying **gene-to-phenotype long-chain pathways**.
>
> **Effectiveness of Cross-sectional Data in Long-chain Mediation:**
>
> SIGMA’s theoretical foundation is built on causal graph structure rather than temporal sequence. Statistical dependencies in cross-sectional data are sufficient to reveal underlying causal pathways. Long-chain mediation paths reflect biological hierarchy and functional relationships. Our Flow-SEM can recover latent causal dependencies from cross-sectional correlational structure, while Path Stability Score ensures the statistical robustness of identified pathways.
>
> **Theoretical and Algorithmic Suitability:**
>
> SIGMA’s core algorithms are suitable for multi-omics cross-sectional data: 1) Flow-SEM handles heterogeneous data types;2) DAG posterior sampling quantifies causal relationships across omics layers;3) Path Stability Score evaluates cross-biological-level pathway robustness;4) Bayesian Model Averaging integrates effect uncertainty across omics.
> As shown in the scalability analysis (Q3), the algorithm remains computationally feasible at p=2000, supporting integrative multi-omics analysis.

---

> ### Author Response · Authors · 2025-08-04
>
> **Dear reviewer 6ZQR**
>
> Thank you for your comprehensive and constructive review. We have systematically responded your four main concerns:
>
> **1. SEM Complexity**: We conducted extensive supplementary experiments using complex SEM models (heteroscedastic noise, composite nonlinear functions) and diverse graph structures (Scale-Free, Hierarchical networks), demonstrating SIGMA's robustness across various complexity levels.
>
> **2. Noise Robustness**: We provided comprehensive robustness analysis across different SNR levels (4.0, 2.0, 1.0, 0.5), systematically evaluating the impact of observational noise on performance.
>
> **3. Scalability**: We extended evaluation to higher dimensions (p ∈ {500, 1000, 2000}) with detailed computational complexity analysis, confirming SIGMA's feasibility at larger scales.
>
> **4. Real-World Applications**: Reviewer APS7 also raised concerns about this issue, leading to in-depth discussions. The key insight is recognizing the fundamental design mismatch between existing large-scale datasets and SIGMA's methodological requirements:
>
> **Design Limitations of Existing Datasets**:
> - **UK Biobank/FinnGen**: Focus on large-scale genetic variant-disease association discovery using cross-sectional genotype-phenotype association analysis. Data structure consists of direct statistical associations from SNPs to disease states, lacking complete variable hierarchies for mediation processes.
> - **TCGA**: Identifies genomic alterations responsible for cancer through genome sequencing, employing cross-sectional tumor tissue multi-omics analysis. Data structure consists of static molecular feature descriptions, focusing on molecular subtyping in disease states.
> - **GTEx**: Establishes resource database for studying relationships between genetic variation and gene expression, using cross-sectional genotype-expression association design. Belongs to single molecular layer analysis with data structure representing static associations from SNPs to gene expression.
>
> **Core Issues of Methodological Requirements Mismatch**:
> - **Path Length Constraints**: Existing datasets optimize for simple association detection (L=2-3), while SIGMA is specifically designed to handle complex mediation networks (L≥4), requiring multiple consecutive mediation variable layers to support long-chain pathway analysis.
> - **Missing Variable Hierarchies**: Existing datasets primarily provide direct genotype-to-disease phenotype associations or single molecular layer analyses, lacking complete mediation variable sequences required for SIGMA validation. Complete biomedical causal pathways should include multi-level variables: genotype→molecular biomarkers→physiological parameters→clinical indicators.
> - **System Integration Limitations**: Most datasets focus on specific analytical levels (genomics only, transcriptomics only, or clinical phenotypes only), unable to support SIGMA's required cross-biological level integrative analysis and identification/quantification of complex network mediation effects.
>
> **Unique Advantages of HPP Dataset**: HPP is specifically designed for systems-level physiological mechanism research, aiming to understand complex physiological regulatory processes in the health-disease continuum, highly consistent with SIGMA's causal mechanism elucidation objectives. It simultaneously measures complete hierarchies from molecular biomarkers (genomics, proteomics, metabolomics) to physiological parameters (cardiovascular, metabolic, neurological functions) to clinical indicators, covering comprehensive measurements across 15 physiological systems, providing the necessary data foundation for cross-system mediation analysis.
>
> **Questions:**
> 1. Do our supplementary experiments regarding SEM complexity, noise robustness, and scalability adequately address your concerns?
> 2. Are there other aspects that require clarification?
>
> We value your professional assessment and look forward to your response.

---

> > ### Comment · Reviewer_6ZQR · 2025-08-06
> > **response to author's rebuttal**
> >
> > Thank you to the authors for the comprehensive and thoughtful rebuttal. I appreciate the considerable effort put into addressing the raised concerns.
> >
> > The new experiments on robustness to systematic complexity and scalability are much appreciated, as they significantly strengthen the empirical foundation of the work. These additions demonstrate a clear commitment to rigorous validation.
> >
> > The one area that remains only partially addressed is the application to larger real-world datasets. While the authors provided a compelling qualitative discussion and outlined a future direction using multi-omics data, empirical validation on such datasets would further solidify the paper’s contribution. However, the authors did provide ample justification to support its relevance and potential in multi-omics settings
> >
> > Overall, given the significant new experiments and justifications, I am happy to raise my score.

---

> > > ### Author Response · Authors · 2025-08-06
> > >
> > > Thank you very much for your thoughtful and constructive engagement throughout the review process. We deeply appreciate the time and expertise you dedicated to thoroughly evaluating our work, and your professional guidance has been invaluable in strengthening our research. Your detailed feedback and constructive approach exemplify the best of academic peer review, and we are grateful for your contributions to improving our work!

---

### Note · Authors · 2025-08-11

We sincerely thank all reviewers and the Area Chair for their professional guidance!!

Reviewer 6ZQR: Raised concerns about SEM complexity, noise robustness, and scalability. We provided supplementary experiments with complex SEM models, diverse graph structures, and noise levels. The reviewer acknowledged our response and explicitly stated "I am happy to raise my score." These suggestions enhanced SIGMA's empirical validation completeness. We deeply appreciate this guidance!

Reviewer rfqt: Raised concerns about terminology usage and technical concepts. We provided literature evidence demonstrating standard practices and committed to detailed explanations and SOTA comparisons in revision. The reviewer stated, "the changes will greatly improve the paper" and updated their rating. The rigorous requirements improved academic rigor and readability. Sincere gratitude!

Reviewer APS7: Raised concerns about computational complexity and generalizability validation. We provided comprehensive complexity analysis, runtime testing, Scale-Free experiments, and argumentation about dataset mismatches. The reviewer stated we "convincingly addressed questions" and raised their score. These insights refined method applicability boundaries. Deep appreciation!

Reviewer MyHV: The reviewer raised important suggestions about bootstrap method comparison and biological mechanism validation. We provided exhaustive bootstrap baseline comparisons and extensive biomedical literature supporting the biological plausibility of newly discovered pathways. We deeply regret that we were unable to engage in more in-depth academic discussion and exchange with the reviewer on these important issues. The reviewer's suggestions significantly enhanced method validation comprehensiveness and credibility. We deeply appreciate this guidance. Our response included: (1) systematic comparisons with PC+Bootstrap, NOTEARS+Bootstrap, GraN-DAG+Bootstrap; (2) recent clinical evidence from HPP studies and sleep apnea-gut microbiome-cardiovascular pathway research. We believe we have reasonably and comprehensively addressed the reviewer's concerns.

Thanks to the Area Chair for professional coordination ensuring review process fairness and constructiveness. Three reviewers explicitly acknowledged and raised scores, demonstrating SIGMA's innovation. We are honored to refine research through academic dialogue. We commit to incorporating all content and improvements from our rebuttal into the final revision.

---

### Decision · Program_Chairs · 2025-09-17

**Decision:**

Accept (poster)

**Comment:**

The authors study the problem of causal mediation analysis. In particular, they propose SIGMA, which combines causal discovery, mediation pathway identification, and effect estimation. All reviewers agree that this is an interesting and relevant problem, and that the proposed approach is novel and exciting. The rebuttal and active discussion that followed addressed the main concerns of the reviewers, leading them all to recommend acceptance. I'm happy to follow their recommendation, on the expectation that the authors include the clarifications and additional results in the camera-ready copy.